

# Effects of shrub cover increase on the near surface atmosphere in northern Fennoscandia

Johanne H. Rydsaa[1], Frode Stordal[1], Anders Bryn, [2], Lena M. Tallaksen[1]

[1] Department of Geosciences, University of Oslo, Oslo, Norway
[2] Natural History Museum, University of Oslo, Oslo, Norway

*Correspondence to*: Johanne H. Rydsaa (j.h.rydsaa@geo.uio.no)

**Abstract.** Shrub expansion in high latitudes is a widely observed response to climate change. Extensive evidence has shown that shrub expansion can lead to positive feedbacks to the regional climate. In this study we evaluate the sensitivity to a potential expansion in shrub and tree cover in the northern Fennoscandia region. Two perturbation experiments are performed in which we prescribe a gradual increase of vegetation height in the alpine shrub and tree cover according to empirically established climatic zones within the study region. The first experiment is based on present day climate, and the second is based on a future 1 K increase in temperature. To evaluate the sensitivity of the atmospheric response to inter-annual variations, simulations were conducted for two different years, one with warmer and one with colder spring and summer conditions. We have applied the Weather Research and Forecasting model (WRF) with the Noah-UA land surface module in evaluating biophysical effects of increased shrub cover on the near surface atmosphere on a fine resolution (5.4 km x 5.4 km). We find that shrub cover increase leads to a general increase in near surface temperatures with the peak influence occurring during the snow melting season. It has the largest effect in spring, by advancing the onset of the melting season, and more moderate effect on summer temperature. We find that the net SW absorbed by the surface is sensitive to the shrub and tree heights, which act to strengthen the albedo decrease. Counteracting effects include increased snow cover and enhanced evapotranspiration causing increased cloud cover and precipitation. We find that the strength of the feedback effects resulting from increased shrub cover is more sensitive to snow cover variations than summer temperatures. Taller vegetation has a stronger influence on both spring and summer temperatures. Our results show that the positive feedback to high latitudes warming induced by increased shrub and tree cover is a robust feature across inter-annual differences in meteorological conditions, and will likely play an important role in the future.

**Keywords.** Shrubs, shrub expansion, Arctic greening, Fennoscandia, WRF, land-atmosphere feedback

## 1    Introduction

Arctic warming is occurring at about twice the rate as the global mean (IPCC, 2013;Pithan and Mauritsen, 2014), partly due to feedback mechanisms in the high latitude ecosystems (Beringer et al., 2001;Chapin et al., 2005;Serreze and Barry, 2011;Pearson et al., 2013). Some of these feedbacks are related to the observed arctic greening (Myneni et al., 1997;Piao et al., 2011;Snyder, 2013). This refers to observed increase in high latitude biomass mainly resulting from increased temperatures (Walker et al., 2006;Forbes et al., 2010;Elmendorf et al.,





2012). While present across arctic ecosystems, the observed increase in biomass is largely related to the extensive expansion in shrub cover in previously tundra covered areas (Tape et al., 2006;Sturm et al., 2001b;Forbes et al., 2010) and northward migrating tree lines (Soja et al., 2007;Tommervik et al., 2009;Hofgaard et al., 2013;Chapin et al., 2005).

Increased tree and shrub cover alters the biophysical as well as the biochemical properties of the surface. This changes land-atmosphere feedbacks. With increasing canopy height and complexity more incoming radiation is absorbed, and the surface albedo is decreased. This is known as the albedo effect of increased vegetation. Sturm et al. (2005) observed the importance of shrub cover on wintertime albedo in snow covered regions and its

implications for the winter surface energy balance. They concluded that increased shrub cover due to higher temperatures caused a positive feedback through lowered albedo. The increase of shrub canopies protruding the snow cover shaded the snow beneath the canopy from radiation and influenced the melt and sublimation. Also, absorbed radiation heated the canopy itself and increased the sensible heat flux to the atmosphere.

Increase in leaf area index (LAI) associated with increase in shrub and tree cover can cause enhanced evapotranspiration (ET) This leads to more latent heat (LH) transfer into the atmosphere, which may also act to increase the temperature (Chapin et al., 2005). The enhanced LH might also lead to enhanced cloudiness and precipitation (Bonfils et al., 2012;Liess et al., 2011). Increased cloud cover may in turn act to limit the effect of the albedo decrease through lowering of the short wave (SW) radiation reaching the surface. In addition,

increased shrub and tree cover has been observed to increase the soil temperatures in winter, enhance the winter snow cover and speed the melting season through its effect on surface albedo (McFadden et al., 2001;Sturm et al., 2001a).

The shrub and tree height is also important for the atmospheric response, and this was studied specifically by

Bonfils et al. (2012). By modelling an increase in the shrub cover by 20% in areas north of 60°N, they found that the regional temperature increased more for taller shrubs as compared to lower ones. They explained the temperature increase by the additional lowering of albedo and increase in ET corresponding to taller and more complex canopies. Increased shrub cover may also act to shade the soil beneath, thereby lowering the temperature of the soil. This acted to decrease summer permafrost thaw as observed by Blok et al. (2010). This

effect was modelled in a study by Lawrence and Swenson (2011) who applied an increase in shrub cover by ~20% in the Arctic region. They, however, found that increased temperatures due to albedo decrease more than offset the cooling of the soil by the shading effect, resulting in a net increase in soil temperatures. In both of the mentioned modelling studies, a 20% increase in shrub cover was prescribed, based on a procedure of expanding existing shrub cover into areas of tundra or bare ground. Based on circumpolar dendroecological data and several

future emission scenarios, Pearson et al. (2013) concluded that the warming effects of increased shrub cover found in the studies by Bonfils et al. (2012) and Lawrence and Swenson (2011) were realistic, but  that the applied shrub expansion of a 20% increase in land cover might be substantially underestimated. They predicted that about half of the regions defined as tundra could be covered by shrubs by 2050, by applying various climate scenarios.




The actual extent of shrub expansion into tundra regions, as well as the predicted increase in shrub density and height in coming decades, is determined by numerous and complex mechanisms and environmental forcers. As highlighted by Myers-Smith et al. (2011), climatic forcers (e.g. air temperature, incoming solar radiation, precipitation), and soil properties (e.g. soil moisture, soil temperature and active layer depth), coupled to

biochemical factors such as the availability of soil nutrients and atmospheric $CO_2$ concentration, all influence the rate of shrub growth. In addition, disturbances, such as fires, heavy snow pack and biotic interactions including herbivory, makes accurate estimates of future shrub distribution challenging (Milbau et al., 2013). Tape et al. (2012) also highlights the importance of soil properties in estimating likely areas of shrub expansion and shrub-climate sensitivity. This factor increases the geographic heterogeneity of shrub expansion. In addition to these

determining mechanisms, increased shrub cover has also been suggested to trigger feedback loops that further induce shrub growth by e.g. shrub-snow interactions. Positive feedbacks include lowering of winter albedo which causes earlier snowmelt, longer growing seasons and increased soil temperatures favorable for growth. Also, thicker wintertime snow packs in shrub areas acts to insulate the ground during winter and increase the soil temperatures (Sturm et al., 2001a).

Several of the controlling factors regulating shrub growth and expansion have been further investigated using dynamic vegetation models. Miller and Smith (2012) found simulated increase in shrub cover caused by warmer temperatures and longer growing seasons. They explained that the shrub cover increase was in part enhanced by feedbacks related to the albedo effect of covering tundra with shrubs, related to increase in canopy heights

protruding the snow cover. In agreement with observations, several other modelling studies have also reported increased biomass production and LAI in response to increased temperatures in tundra regions, resulting partly from shrub invasion and replacement of low shrubs by taller shrubs and trees (e.g. Zhang et al., 2013;Miller and Smith, 2012;Wolf et al., 2008).

Several recent studies have aimed at isolating the dominating environmental drivers of shrub expansion. Myers-Smith et al. (2015b) investigated climate-shrub growth relationships and found that mean summer temperatures and soil moisture content are particularly important forcers. By examining circumpolar dendroecological data from Arctic and alpine sites, they demonstrated that the sensitivity of shrub growth to increased summer temperatures was higher at European than American sites. Furthermore, they found that the sensitivity to climate

forcing was higher for taller shrubs at the upper or northern edges of their present domain and at sites with greater soil moisture. Also based on dendroecological observations in northern Scandinavia, Hallinger et al. (2010) concluded that the mean summer temperatures and winter snow cover are the main climatic drivers correlated with shrub growth in sub alpine areas. Elmendorf et al. (2012) also demonstrated a biome-wide link between high latitude vegetation increase and local summer warming, based on tundra vegetation surveys

covering 30 years in 158 plant communities spread across 46 high latitude locations.

Since the change in biophysical properties related to increased shrub cover in tundra areas is more moderate as compared to changes resulting from e.g. a comparable expansion of forest ecosystems, a comparatively modest effect on the overlying atmosphere is to be expected (Beringer et al., 2005;Chapin et al., 2005;Rydsaa et al.,

2015). Considerable uncertainties still exist within estimates of both the extent of shrub and tree advance in



relation to climatic forcing, and to the corresponding feedback of these (Myers-Smith et al., 2015b;Pearson et al., 2013).

In this study we investigate the regional atmospheric response related to biophysical changes resulting from enhanced vegetation cover in high latitudes. We focus on both expansion of the areal extension of shrubs and low deciduous trees and effects of increased height of shrubs and trees. For this purpose, simulations are conducted for a limited region on high temporal and spatial resolution (5.4 km x 5.4 km), using a state-of-the-art regional atmospheric model. This enables us to investigate fine scale features of vegetation changes and the corresponding atmospheric response. The shrub and tree vegetation is redistributed across the study domain by applying three types of shrub and tree classes according to their climatic envelope. These are derived from empirically determined vegetation-climate relationships for the region.

We focus on the response of the atmosphere during the spring and summer seasons. In addition, the sensitivity to inter-annual variation in mean environmental conditions is investigated. Finally, to explore the potential future atmospheric feedbacks to increased shrub and tree cover on high latitudes, an additional experiment representing a simplified future scenario is conducted. In this experiment, we apply a re-distribution of the shrub and tree cover corresponding to a theoretical 1 K increase in summer temperatures. The resulting feedbacks to the atmosphere are assessed and compared to the effect of vegetation changes corresponding to present day climatic envelopes.

## 2 Methodology and model

### 2.1 Methodology

In order to investigate the effects of increased shrub and tree cover (referring to both areal expansion and increased height), we have conducted six simulations; control simulations for two different years, and two corresponding sets of simulations in which the vegetation cover has been manually altered to represent increased shrub and tree cover. By comparing the reference and experiment simulations we can determine the effect of vegetation cover changes, as the simulations are otherwise identical. Our investigations are carried out on a domain covering northern Fennoscandia and north-west Russia (Fig. 1). This is a sensitive region for shrub expansion in response to climate forcing according to (Myers-Smith et al., 2015a). Extensive increase in the shrub covered area, in addition to shifts in tree lines towards higher latitudes and altitudes, has been observed in this region over the past decades (Tommervik et al., 2004;Hallinger et al., 2010;Tommervik et al., 2009;Rannow, 2013).

In this study, subjects of particular interest include differences in atmospheric feedbacks between seasons. The spring season has been identified as the season with the strongest feedback to temperatures from increased shrub cover in previous studies (Bonfils et al., 2012;Lawrence and Swenson, 2011). This is due to the effects on snowmelt and corresponding surface albedo changes. As the mean summer temperature has been identified as one of the main environmental drivers for future shrub expansion by several studies, a large potential for growth feedbacks lies with the warming response of the atmosphere during this season. For these reasons we have chosen to focus on the atmospheric response during spring and summer seasons.





The atmospheric response to shrub expansion may also be sensitive to inter-annual variation in mean environmental conditions. Our investigations are conducted for two spring seasons and two summer seasons which represent a wide range in two key environmental conditions; the spring snow cover and the summer temperature. This setup enables us first to investigate the importance of shrub cover increase on the land-atmosphere interactions in both summer and spring seasons. Secondly, we are able to examine how the atmospheric response varies between snow-rich and snow-poor melting seasons, and the sensitivity to temperature on the summer atmospheric response. The two years were selected based on a ten-year (2001-2010) long simulation by Rydsaa et al., (2015), which is a dynamical downscaling of ERA Interim, using the Weather Research and Forecasting (WRF) model. The reason for using this dataset instead of a global dataset was the ability to search through relevant variables, and the consistency in modelling tool and boundary conditions with this study. The year of 2003 represents a low snow cover spring season and a warm summer season in this region (hereafter referred to as the warm spring and summer season). The year 2008 represents a snow-rich spring season and a cold summer season in this region (hereafter referred to as the cold spring and summer season).

## 2.2    Model

For the purpose of this study, we have used WRF  V3.7.1 (Skamarock et al., 2008). WRF is a non-hydrostatic weather prediction system with a wide variety of applications ranging from local scale domains of a few hundred meters in resolution to global simulations. With a wide range of physical parameterization schemes, the setup may be adjusted to simulate case-specific short-term weather events, or decadal long climate simulations. The current setup is based on available literature about the NCAR choices for physical parametrizations for high latitude domains, and a consideration of the polar WRF setup and validation studies (Hines and Bromwich, 2008;Hines et al., 2011). A summary of key physical schemes applied in this study is presented in Table 1.

As initial and boundary conditions we used the ERA Interim 6-hour reanalysis. The model was run for two domains, where the outer domain with a resolution 27 km x 27 km (90 x 49 grid cells) serves purely as a bridge between the coarse resolution boundary conditions and the finer inner domain with resolution 5.4 km x 5.4 km used for analysis. The model was run with 42 vertical layers and output written every 3 hours. Each simulation spans the snow accumulation season (starting in November), spring and summer, however only the spring (MAM) and summer (JJA) seasons are included in the analyses.

The model was run with the Noah-UA land surface model (LSM), which is the widely used Noah LSM (Tewari et al., 2004), with added parameterization for snow-vegetation interactions by Wang et al. (2010). The added parameterizations include consideration of the vegetation shading effect on snow sublimation and snowmelt, under-canopy resistance, improvements to the ground heat flux computation when snow is deep, and revision of the momentum roughness length computation when snow is present. In the model, the soil is divided into four layers of varying thickness, summing up to a total of 2 m. The LSM controls the soil and surface energy and water budgets, and computes the water and energy fluxes to the atmosphere, depending on air temperature and moisture, wind speed and surface properties. The dominant vegetation category in a given grid cell determines a range of biophysical parameters related to its interaction with the atmosphere. These parameters include the



height and density of the canopy, the number of soil layers available to the plants' roots, minimum canopy resistance, snow depth water equivalent required for total snow cover, and ranges for values of leaf area index, albedo, emissivity and surface roughness length. A list of parameter values used to represent the relevant vegetation categories in our simulation is presented in Table S1, supplementary material. The value within each range is scaled according to the vegetation greenness factor, which is updated here based on a prescribed monthly dataset provided with the WRF model. As such, this setup is able to capture changes in surface properties related to redistribution of vegetation classes and atmospheric effects resulting from these. It will not simulate the vegetation's response to environmental forcing, such as changes in surface temperature or soil moisture. Only prescribed changes to the vegetation as described in the next section, and corresponding effects on the overlying atmosphere, differ in reference versus perturbed simulations. Alterations in the atmosphere results from the biophysical changes related to the applied vegetation perturbations alone.

### 2.2.1 Land cover and re-distribution

For the land cover input data, we use the newly available 20 class MODIS 15 sec resolution dataset (Broxton et al., 2014). In this dataset most of the Arctic and alpine part of our study area is covered by the dominant vegetation category of "open shrubland", consisting of low shrubs of less than 0.5 m height. This category was further used as a basis to implement empirically based adjustments aiming to distinguish shrubs and low deciduous trees of various heights. For this purpose we divided our study domain into climatic vegetation zones based on mean JJA temperatures, and we defined shrub and low tree categories according to their empirically derived climatic vegetation zones (Bakkestuen et al., 2008). In order to prevent shrubs from being distributed in areas that are unsuitable despite favorable climatic conditions, the extent of areas with other vegetation categories were kept unaltered. In this way the vegetation was adjusted in accordance with the derived climatic vegetation zones, while keeping the heterogeneity in the vegetation distribution as observed in the original dataset.

The results for the dominant vegetation categories as distributed across the domain for the different simulations, along with mean JJA 2 m temperature limits used to determine the climatic envelopes, are shown in Fig. 1.

In order to distribute the climatic vegetation zones geographically across the study domain, we utilized some general features of vegetation distribution among the alpine zones that have been determined for this area. The various alpine zones are defined as altitudinal dependent belts of vegetation above the forest line (Gottfried et al., 2012). Although the altitudinal extent of each alpine zone is determined by the local mean decline in temperature with elevation (i.e. local mean lapse rate) (Bakkestuen et al., 2008), in addition to various geographical and climatic features, the relative altitudinal extent of alpine zones remains rather constant throughout the domain in focus here. This altitudinal extent is used here to determine temperature based climatic envelopes for each alpine zone. The altitudinal extent of each alpine zone used in this study is based on (Moen et al., 1999), but also confirmed by a new dataset from the region (Bjørklund et al., 2015).

As illustrated in Fig. 2, and following the vegetation categorization of Moen et al. (1999) and Bakkestuen et al. (2008), we defined tall shrubs and boreal deciduous trees reported to be characterized by a height from 2 to 5 m (Aune et al., 2011) as belonging to the sub-alpine zone. We defined tall shrubs with height from 0.5-2 m as





belonging to the low-alpine zone, and low shrubs with height up to 0.5 meters belonging to the mid-alpine zone. The high-alpine zone contains no shrubs and is characterized by barren ground, boulder fields or scattered vegetation (Moen et al., 1999). High mountain tops were regarded as high-alpine (and largely fit with the defined climatic limits), and vegetation cover in these areas were adjusted accordingly (see e.g. Karlsen et al. (2005)).

The climatic forest line was found and applied to separate the boreal forest from the sub-alpine region characterized by scattered mountain birch (Aas and Faarlund, 2000). The last mountain birches stretching towards higher elevations are approximately 2 m tall, and define the so-called boreal-tundra or tree line ecotone (Hofgaard, 1997;Bryn et al., 2013;de Wit et al., 2014). This ecotone was determined here to be above the line where the fraction of boreal tree cover exceeds 25% in each grid cell. This line furthermore defines the base line

temperature used to derive the alpine vegetation zones at higher elevations. The climatic boreal forest line was found to correspond well with the mean summer 12 $^o$C isotherm in our domain (Fig. 1), which is slightly higher than what is found in southern parts of mountainous Scandinavia (Aas and Faarlund, 2000;Bryn, 2008). The sub-alpine zone was then determined based on an average altitudinal extent of 100 m (Aas and Faarlund, 2000), the low-alpine and mid-alpine zones were estimated to be on average 300 m each, and vegetation cover at higher

elevations defined as high-alpine zone (Moen et al., 1999) (Fig. 2).

Based on temperatures from previous simulations covering the area (Rydsaa et al., 2015), the mean tropospheric JJA lapse rate for the area was found to be 6.0 K km$^{-1}$. This value was used together with the average zone-heights to find the summer temperature ranges for each climatic vegetation zone. The interpolated mean JJA 2 m

temperature was then used to distribute each shrub category across the domain in accordance with their climatic envelope. In order to simulate a shrub expansion and re-distribution of tall vs. low shrubs in accordance with a theoretical future increase in summer temperatures, revised climatic zones with the same relative altitudinal extent were calculated based on increased JJA 2 m temperatures, and vegetation categories were re-distributed accordingly. For simplicity, the mean lapse rate is assumed to be the same in the new temperatures. The

procedure is illustrated in Fig. 3.

To represent each alpine vegetation type in the model, we chose suitable vegetation categories (and corresponding parameter values) from the ones already defined within the model system. The categories were chosen based on categories already present in the domain, with special emphasis on decreasing LAI and canopy

height for vegetation applied towards higher altitudes and latitudes. The choices were based on a recent mapping of vegetation types in the region (Bjørklund et al., 2015). A list of shrub categories and their corresponding parameter values is presented in Table S1, supplementary material. With two exceptions (see supplementary material, Table S1, bold), parameter values were left unaltered to keep consistency between and within each vegetation category. The original usage in the model system is given along with the minor alterations.

**3   Results**

The only change between the reference simulations ("RefVeg") and perturbed simulations ("Veg0K", "Veg1K") is the land cover. Any other differences in other variable values result from the land cover changes, as simulations are otherwise identical with respect to setup and meteorological forcing data. This methodology aims to isolate the signal in the atmospheric response to land cover changes alone. The results are therefore presented



largely as anomalies between reference and perturbed simulations when the effects of land cover are of main interest. The difference between Veg0K and RefVeg shows the effects of an increase in shrub and tree cover as having reached the present day climatic potential (as defined here). The difference between Veg1K and RefVeg on the other hand, shows the enhanced effect of a potential vegetation shift corresponding to a 1 K shift in surface temperatures and is presented afterwards.

In the following section the mean effects on the overlying atmosphere of increased shrub and tree cover (Veg0K-RefVeg) for each season (MAM and JJA) are presented. Results are given as averages over the two spring seasons and two summer seasons. This gives a mean estimate of the response of the atmosphere across a wide range in meteorological conditions and thus a robust estimate of shrub induced effects across inter-annual variations. Results comparing the anomalies between warm and cold spring and summer seasons are presented next, as it will indicate the sensitivity of the atmospheric response to specific meteorological variations. Here these are limited to variations in snow cover (in Section 3.1.1) and near surface temperatures (in Section 3.1.2). Finally, in Section 3.2 a brief analysis of the future scenario with vegetation distributed according to a 1 K increase in temperature is presented. Special emphasis is on how the increased shrub and tree cover alters the feedback to atmospheric temperature. Mean values for the reference simulations, along with mean response as averaged over all areas with vegetation changes for surface fluxes and near surface atmospheric variables, are presented in Table 2.

### 3.1 Atmospheric effects of shrub and tree cover increase

Effects of shrub and tree cover increase by both expansion into areas with lower vegetation, and by increased height of existing shrubs (Veg0K-RefVeg) as averaged over the two spring seasons, are presented in Fig. 4. Top panels show the change in 2 m temperature averaged over the spring seasons (left) and mean values for each separate area with vegetation changes (corresponding areas are indicated in Fig.1, bottom panel) (right).

In spring, an overall increase in near surface temperatures is seen for all areas with shrub cover increases. The higher anomaly values are seen in areas with increase in taller vegetation (as also indicated in the bar plots). Average increase in 2 m temperature over the spring season is 0.1 K (Table 2); however large spatial differences appear (Fig. 4, bar plot). The mean response reaches up to 0.6 K in some areas with taller vegetation (Fig 4). There are also large temporal differences within the seasonal response, and the increase as averaged over all areas with vegetation changes peaks during the melting season in mid-May with 0.8 K (not shown).

The second row of panels in Fig. 4 shows changes in fluxes of net short wave (SW) radiation (towards the surface). The main increase in net SW fluxes is seen during the spring season, mainly due to decreased surface albedo caused by both the increased shrub and tree cover, and particularly its effect on earlier snowmelt (Section 3.1.1). The spring season's net increase in SW radiation occurs despite a slight decrease in downwelling SW (not shown) caused by enhanced cloud cover (Fig. 6 and Table 2). The reduction in downwelling SW is more than compensated by the albedo decrease in areas with sub alpine vegetation (taller vegetation), but the net value is close to zero in areas with low alpine shrub increase (lower vegetation) due to smaller albedo changes (second row, and bar plot). The long wave (LW) radiation is slightly increased in response to enhanced cloud cover and





atmospheric humidity (Fig. 7 and Table 2).The increase is more evenly distributed across the region, as it is not as directly linked to the vegetation changes.

The heating associated with the increase in SW is partly balanced by an increase in evapotranspiration (ET), shown as the latent heat flux (LH) (Fig 3, row 4). The increased LAI due to increased shrub and tree cover

(Table S1, and Fig. S4, supplementary material) causes increased ET, and correspondingly increased LH. The effect is larger in areas with larger LAI increase, i.e. in areas with taller vegetation. The increase is largest towards the end of the spring season (not shown), much owing to larger above-snow canopy fraction due to the canopy height increase associated with more shrubs and trees and reduced snow cover (Fig. S2, S3, supplementary material). An increase in sensible heat flux (SH) (Fig. 4, row 5) from the surface and from

canopies protruding the snow cover is seen in areas with taller vegetation, where net SW is positive. This adds to the effect of increased LH in balancing the surplus of SW energy at the surface.

In the summer season (Fig. 5) the 2 m temperature is increased in areas with taller vegetation, and decreased in the areas with low alpine shrub increase (lower vegetation). These areas are characterized by a lowering of net

SW radiation in this season, which results in a decreased sensible heat flux and less warming of the lower atmosphere. The summer season SW downwelling is decreased due to increased cloud cover (Table 2, Fig.7 and as confirmed by the increased LW to the surface). In addition, the negative net SW is caused by a slight albedo increase in the early summer season (early to mid-June, not shown) due to enhanced snow cover in these areas (Fig. S3 and S4, supplementary material). The latter is due to the increased precipitation (also snowfall) (Table

2) and enhanced shading effect by more shrubs, combined with the small albedo decrease associated with the low-alpine shrub increase. The areas with taller shrubs and trees on the other hand, are characterized by a decrease in snow cover throughout the spring and summer seasons due to a stronger albedo decrease (Fig. S4, supplementary material).

The LH does not affect the 2 m temperature to the same degree as the SH, as the heat is released higher up in the atmosphere as the water condenses (Fig. 6). The vertical structure of the lower atmosphere heating along a cross section across the domain is shown in Fig. 6, along with changes in planetary boundary layer PBL) height and turbulent fluxes of SH and LH. The values are mean monthly May values from the warm spring season, and as such they represent one of the months with the largest heating anomaly. The increased SH mainly acts to heat the

lower atmosphere within the boundary layer, while the LH is also released above the PBL height. The heating anomaly has a westward tilt with altitude, and the signal extends well beyond the PBL, however the main heating is occurring within the lower 200 meters. The changes in LH and SH fluxes are local to the areas with vegetation changes, as is the increase in PBL height.

The spatial distribution of mean changes to low clouds (here defined as below 3 km) and precipitation for the two seasons is shown in Fig. 7. The atmospheric humidity increase associated with increased shrub cover results in more clouds and accumulated precipitation in both seasons (Table 2). The increased cloud cover acts to decrease the SW radiation reaching the surface in both seasons (shown only as net SW, Fig. 4 and 5) and increase the amount of LW radiation towards the surface (shown only as net LW, Fig. 4 and 5). The effect is



largest in areas where the humidity is increased the most through enhanced LH, i.e. in areas with increase in taller vegetation.

Top panels in Fig. 7 show the relative change in low cloud cover resulting from increased shrub cover. Here the change in cloud cover is shown as the fractional cloud cover as averaged over the lower 3 km of the atmosphere (further detail about this variable in the supplementary material). The most prominent increase in low cloud cover is occurring in spring (Fig. 7, upper left panel) and is largely covering areas with vegetation changes. The pattern for the summer season's response is patchy, although a tendency of increased cloud cover in areas with vegetation change is recognizable and statistically significant (see Table 2). The second row shows the relative increase in precipitation, as accumulated over the season. Only areas with significant changes are shown and the relative change in precipitation is based on daily accumulated values. The spatial distribution of significant values of precipitation changes is also somewhat patchy, particularly for the summer season. However, the significance is higher in areas with vegetation changes, as compared to the total area (the relative amount of cells with significant results in areas with vegetation changes is 8.3%, versus 5.7% in the total domain). This is indicating that most of the significant increase in precipitation is related to the vegetation changes. The increase in accumulated precipitation is most prominent in summer, during which the precipitation increase summarized over all areas with vegetation changes is 186 mm, corresponding to a 2.2% increase ($p$-value based on the Mann-Whitney significance test is $1.2\ 10^{-5}$). For spring, the increase in precipitation is 1.1% ($p= 2.8\ 10^{-9}$), and for precipitation in the form of snow and ice, it is 1.4% ($p= 3.19\ 10^{-9}$).

### 3.1.1 Sensitivity to snow cover differences

The spring seasons are characterized by large differences in snow cover, albedo and near surface temperatures between the two years. As compared to the cold season, the warm spring season has 16% less snow cover, resulting in a decreased albedo of 12% and an on average 3.1 K warmer 2 m temperature. There are small differences in precipitation among the two years, although the rain-to-snow ratio is larger in the warm season due to higher temperatures. The difference in mean spring snow cover between the warm and cold spring seasons is presented in Fig. 8.

It is clear that the onset and speed of the melting season differ between the two years, and the main difference in snow cover is in May. The melting starts more than two weeks earlier in the warm spring season. The dominating effects of increased shrub cover differ between snow covered and snow free conditions and the differences in the shrubs' influence on the atmosphere are largest during the melting season in May-June.

The difference in 2 m temperature response to increased shrub cover between the warm and cold seasons is shown in Fig. 9. The warm spring season experiences up to 0.38 K higher increases in temperature in response to shrub cover increase as compared to the cold spring season. As seen in the right panel of Fig. 9, the anomaly distribution is shifted towards higher values in general in the warm season. The shrubs act to enhance warming in the warm spring season more than in the cold spring season. This represents a positive feedback to warm conditions and early snowmelt.

The increased shrub and tree cover decreases the snow depth in the spring season as averaged over all areas with vegetation changes, as seen in Fig. 10, top panel (spatial pattern shown in Fig. S3, supplementary material). An





exception is seen in the late spring (and early cold season summer, not shown). This is owing to the above mentioned late spring and early summer increase in snow cover in areas with low alpine shrub increase. In these areas the mean increase is due to increased snow fall in the cold season and possibly the increased shading effect of the shrubs. These effects dominate over the effect of the weak albedo decrease. This increase in late

spring/early summer snow cover leads to a shortening of the snow free season in these areas (here defined as the fraction of ground covered by snow less than 0.1). In the cold season the shortening is only about half a day averaged over the areas with vegetation changes. The warming effect of shrub cover in the warm season on the other hand, acts to prolong the snow free season by just over one day, but it speeds the onset of melting by several days.

Increased shrub and tree cover also influences the soil moisture and temperature, as shown in Fig. 10. The shrubs act to decrease surface runoff during the peak of both the cold and warm melting seasons. However, during the warm spring season before the main snowmelt starts, runoff is slightly increased due to shrubs, because of their effect on increased snow melt earlier in spring. The soil moisture is increased in areas with increased shrub cover

throughout the warm spring, and due to the shrubs' ability to decrease surface runoff during the main snowmelt, the mid-May soil moisture is particularly increased. Also, increased shrub and tree cover acts to enhance the soil temperatures (Fig. 10, third panel), with maximum impact in the upper layers of the soil (not shown). The increased precipitation throughout both spring and summer seasons also influences the soil moisture.

**3.1.2    Sensitivity to summer temperatures**

The two summer seasons represent a large range in inter-annual variability. The mean JJA 2 m temperature over land for the warm summer season was 11.7 $^{\circ}$C, while the cold summer represents a lower than usual mean temperature of 9.7 $^{\circ}$C. Comparing the two summers, the temperature difference over land is on average 2 $^{\circ}$C, and in some places the average difference reaches 3.3 $^{\circ}$C. The corresponding increase in atmospheric absolute

humidity at 2 m is 6.9%. The warm summer also represents drier conditions with lower precipitation (Table 2). The difference in atmospheric temperature response to increased shrub cover is shown in Fig. 11.

The response of the atmosphere to increased shrub cover shows more similarity across the two summer seasons as compared to the two spring seasons. For the summer seasons, the mean difference in near surface temperature

response between the warm and cold season is smaller and more evenly distributed around zero (Fig.11, right panel). The positive values over areas with low alpine shrub expansion largely indicate less cooling in the warm summer season as compared with the cold season, during which these areas were partially covered by snow. The tall vegetation contributes to similar warming in the summer seasons. The temperature response in the warm season is slightly shifted towards warmer anomalies (Fig 11, right panel), indicating a slightly positive feedback

to warmer summer temperatures in the warm summer season when compared with the cold.
As the difference in atmospheric response is larger between the warm and cold spring season when compared with the warm and cold summer season, it seems that the strength of the temperature feedback to increased shrub cover is more sensitive to snow conditions than summer temperatures.





### 3.2 Sensitivity to a 1 K shift in vegetation cover: "future scenario"

The shift in shrub and tree distribution according to the theoretical 1 K increase in summer temperature results largely in a northward migration of the boreal tree line ecotone, replacing low alpine shrubs with small trees across most of the shrub covered areas. It also acts to increase the low alpine shrub cover in higher latitudes and altitudes (Fig. 2). The increased cover of trees, with corresponding strong decrease in albedo and increase in LAI, enhances the net SW absorbed by the surface. This is balanced by strong increases in SH and LH (Table 2, and Fig. S5, supplementary material). In addition, the vegetation changes result in increasing precipitation and cloud cover (Table 2). The mean seasonal response in 2 m temperature to the increase in shrub and tree cover corresponding to the 1 K increase is shown in Figure 12.

The atmospheric heating effect at 2 m resulting from the "future" vegetation shift is on average more than doubled as compared to that of the shrub and tree cover increase according to present day climate (Veg0K) in both seasons (Table 2). This is due to the more drastic effect on the biophysical properties related to the extensive shift towards taller vegetation. The warming is most prominent in late spring as the increased vegetation cover affects the snow melt and corresponding albedo and surface heat fluxes. The average spring season heating is therefore strongest in areas with the tallest vegetation. The highest peak values, up to 0.71 K, are however found in summer (Fig. 12). The potential enhancement of the warming effect in Veg1K (as compared to the warming resulting from Veg0K) is thus largest in the summer season. Increased LH also leads to enhanced atmospheric moisture and more summer precipitation in this experiment (Table 2) and corresponding greenhouse effect of up to 5 W m$^{-2}$ (not shown).

The response of this vegetation change also differs between the warm and cold summer. In contrast to the previous experiment (Veg0K), the strongest warming is taking place in the cold summer in most areas. However, due to the influence of early summer cooling in areas with low alpine shrubs caused by increased snow cover in the cold summer (as in the previous experiment), the average response across the entire area with vegetation changes is a slightly stronger warming in the warm summer (0.16 K versus 0.15 K in the cold summer, Table 2).

### 4 Discussion

The vegetation perturbations applied to represent shrub cover increase in this study are moderate in both areal extent and in vegetation property changes, as compared to other studies with similar purpose (e.g. Bonfils et al., 2012;Lawrence and Swenson, 2011). We have altered shrub properties only in areas already covered by tundra and low shrubs, and only within empirically based suitable climatic zones (Fig. 1 and 2). Shrub properties were selected from predefined vegetation categories within the modelling system employed to represent high latitude vegetation. Only minimal alterations were made to the existing categories in order to keep consistency within and between the vegetation categories applied in the modelling domain. This approach does inherit some uncertainty regarding the suitability of single parameter values. However, we judged that changing them might lead to unintended biases within the modelling system. A complete review of the parameter values applied within the modelling system is beyond the scope of this study.



As we have chosen to focus on biophysical aspects of the effects of shrub expansion, there has been no atmospheric or soil chemistry changes included, nor effects of aerosols. These factors may substantially alter atmospheric composition and possibly impact on the response to vegetation changes. However, other studies have concluded that the main impact of changes in the high latitudes ecosystems results from biophysical effects
(Pearson et al., 2013;Bonan, 2008).

Due to its limited size and proximity to warm waters off the coast of Norway, our domain is largely influenced by the incoming marine air from the west. This advection of weather into the domain acts to diffuse the shrubs' and trees' effects on the atmosphere. As such, our results for impacts on upper atmospheric features, such as
cloud cover and precipitation, are more heavily influenced by the meteorological boundary conditions than the near surface variables. This effect could influence our results for atmospheric response to be more modest as compared to results of similar studies on circumpolar domains (Bonfils et al., 2012;Liess et al., 2011).

No vegetation dynamics were included in this study to account for the vegetation's response to the changing
environmental conditions. This represents a limitation in our simulations, particularly with regard to differing responses among the cold and warm seasons. However, it is hard to predict whether this represents an over or underestimation of our results. The seasonal variations of vegetation in these simulations are results of the daily interpolated greenness factor (based on monthly values), acting to scale between maximum and minimum parameter values representing each vegetation category (such as the LAI and vegetation albedo etc.). As we have
made no assumptions about changes in the vegetation density distribution within each grid cell besides that of the vegetation specific properties, the greenness fraction was left unaltered in our perturbations. This might influence the results and in particular influence the partitioning between latent and sensible heat flux.

The spring albedo effect on absorbed radiation is often regarded as the most important effect of increased
vegetation cover in high latitudes (Arora and Montenegro, 2011;Bonan, 2008), and our results confirm this as the main cause of warming during the spring season. Our findings show that the net SW is highly sensitive to the vegetation properties such as the height of the vegetation. We find that competing effects of increased ET resulting in more cloud cover and precipitation, versus the effect of albedo decrease and enhanced melting, determine the net SW and influence the near surface temperatures. Albedo changes related to more complex
canopies and enhanced snowmelt dominate the spring effects in areas with increase in tall vegetation. The net effect related to increase in lower shrub cover is more dependent on the balance between the albedo decrease and added snowfall and snow cover due to more moisture in the atmosphere.

Taller vegetation has a stronger impact on the summer surface fluxes and temperature due to the large increase in
LAI and decrease in albedo in summer for the boreal and sub-alpine deciduous trees. The surface albedo decrease is largest in summer, despite the snow masking effect in winter, mostly owing to the deciduous nature of the northward expanding shrubs and trees in this study, which is based on what is observed in the study region (Hofgaard et al., 2013;Aune et al., 2011). This would be different if we allowed for expansion of evergreen needle leaved trees (Rydsaa et al., 2015;Arora and Montenegro, 2011;Betts and Ball, 1997), which would more
strongly affect the albedo across all seasons.



Bonfils et al. (2012) applied a 20% increase in shrub cover in bare ground areas north of 60°N in order to study the influence of shrubs on climate. They found a regional annual mean temperature increase of 0.66 K for shrubs with height 0.5 m, which was most prominent during the spring melting season. To investigate the sensitivity of height and stature of shrubs, they performed a second experiment, increasing the shrub heights to 2 m. This
caused the regional annual warming to increase to 1.84 K by 2100. Furthermore, they found increases in both SH and LH, the latter mainly resulting from an increase in ET. They also found an increase in summer precipitation, particularly in the case of tall shrubs.

The seasonal mean spring temperature increase in our simulation for present day shrub distribution (Veg0K)
reached 0.59 K in in some areas with the tallest vegetation. The warming as averaged over the entire area with vegetation changes reached 1.0 K during the warm melting season, due to the strong impact of shrubs and trees under snow free conditions. These peak values represent the warming potential of the vegetation changes applied here. However, we find large differences in the response related to the varying heights of the shrub cover, as was also found by Bonfils et al. (2012). In the large areas with low alpine shrub cover increase, the average warming
was of only 0.1 K, reflecting the increased snow cover in late spring in these areas which was partly caused by increased snowfall and partly by the increased shading. Combined with the weak counteracting effect of small albedo decreases associated with the low alpine shrubs as compared to the low shrubs previously present in these areas, the net results for the SW and 2 m temperature were negative in early summer. The summer maximum increase in near surface temperature reached 0.39 K in areas with taller vegetation. The warming was mainly
associated with taller vegetation, confirming the strong dependence of the atmospheric response on vegetation height also in this season.

Lawrence and Swenson (2011) also applied a 20% increase in shrub cover north of 60°N. In their case this led to a moderate increase in mean annual temperatures of 0.49-0.59 K, with a peak during the melting season in May
of 1-2 K. They also found an increase in soil temperatures of 3-5 K in winter and spring following added shrub cover and re-distributed snow cover. Although not directly comparable, we note that their results were substantially larger than the soil temperature response in our results, with maximum values reaching up to 1.5 K in the top soil layer during the warm melting season. This difference is probably related to inter-model differences in soil and vegetation properties, and particularly to differences in simulation domain and extension
of shrub and snow cover increase. Their analyses did not include effects on cloud cover and precipitation.
Swann et al. (2010) applied a 20% increase in shrub cover north of 60°N and found an annual warming of 0.2 K and a decrease in low level clouds despite increased vapor content due to increased ET. Similarly to our study, they also found increase in summer precipitation, but not in spring.

The temperature increases in our results, both for the peak melting seasons and in seasonal means, are below the seasonal estimates of some of the similar studies. This was expected given the comparatively more moderate vegetation shifts (both on areal scale and partly in vegetation properties) in our simulations. Also, large variations in the atmospheric response with regard to cloud cover and precipitation were found among the mentioned modelling studies, despite qualitatively similar responses of enhanced ET and LH related to increased
shrub cover.



The response of shrub expansion in our simulations was larger in the warm than in the cold year, when averaged over the entire area with vegetation shifts, both in spring and summer. However, the difference in response between warm and cold summers was more moderate. Based on these results, we might expect that in a warmer climate, the effect of shrub expansion would enhance the positive feedback of early snow melt, and contribute to prolonged growing season and particularly increase spring surface temperatures. The areas with strongest feedback to the summer season warming were related to taller vegetation (sub alpine and boreal).

We assume that the vegetation distribution applied according to a theoretical 1 K shift in summer temperatures may serve as a simplified proxy for a future scenario with regard to vegetation trajectories and re-distribution. However, precautions should be made when interpreting these results, as the time delay related to such a vegetation shift could be substantial (Corlett and Westcott, 2013), and the fact that the actual vegetation re-distribution according to such a shift in summer temperatures would be limited by other environmental and ecological factors, as mentioned in the introduction and discussed by Svenning and Sandel (2013) and Myers-Smith et al. (2011). If this is kept in mind, a careful interpretation of these results is still useful. The increased vegetation cover resulting from the 1 K shift in JJA temperatures was largely represented in this study by extended areas of sub alpine and boreal deciduous vegetation cover, consisting of tall shrubs and low trees. The northward migration of taller trees and the sub-alpine ecotone enhanced the warming in both seasons, but to a larger degree in summer (on average 0.16 K in Veg1K-RefVeg, as compared to 0,04 in Veg0K-RefVeg, Table 2). Peak mean values of seasonal anomalies in this experiment were higher in the summer season as compared to the spring season.

These findings lead us to conclude that the main summer temperature feedbacks are related to taller vegetation. As the mean summer temperature is assumed here to be the main environmental driver of shrub expansion, we find that the warming effect on summer temperature and thus feedback to further shrub and tree growth is subject to a time lag corresponding to that of establishment of taller shrubs and sub alpine trees in tundra areas, as also found by de Wit et al. (2014).

Also, based on the differences in response between the warm and cold summer in these experiments, a positive feedback to summer warming seems to be a robust feature across inter-annual differences, as warming of the summer atmosphere occurs quite evenly in cold and warm summers. Given the strong impact of the northward migrating sub-alpine ecotone on the summer temperature shown here, we find the possibility for a future ecological "tipping point" in this area rather probable, and this would be an interesting topic to investigate further. The term refers here to the level of vegetation response, where the atmospheric warming resulting from increased shrub and tree cover feedbacks enhances the further growth to such a degree that the response becomes nonlinear in relation to the initial warming (Brook et al., 2013).

## 5    Summary and conclusions

In this study, we have applied the weather, research and forecasting model (WRF) coupled with the Noah-UA land surface model, in evaluating the biophysical effects of shrub expansion and increase in shrub height on the near surface atmosphere. We have applied an increase in shrub and deciduous tree cover with heights varying in





line with the present climate potential according to empirical temperature-vegetation limits for the region (Veg0K vs. RefVeg). To evaluate the sensitivity of the atmospheric response to inter-annual variations, simulations were conducted for two different years, one with warmer and one with colder spring and summer conditions. The response across the different years represents an atmospheric response across a broad range in
temperature and snow cover conditions. To evaluate the sensitivity to a potential further expansion in shrub and tree cover corresponding to a 1 K increase in mean summer temperature, we conducted additional simulations for each year, applying a new vegetation cover shifted according to new climatic envelopes (Veg1K).

Our results show that shrub and tree cover increase leads to a general increase in near surface temperatues,
enhanced surface fluxes of heat and moisture, increase in precipitation and cloud cover across years and seasons. A notable exception is areas with sub alpine shrubs early in the cold summer season, where increased snow cover resulting from shrub expansion leads to increased surface albedo dominating over the decrease in albedo resulting from the shrub increase. This highlights that the net SW absorbed by the surface strongly depends on the strength of the albedo decrease due to enhanced canopies, versus albedo changes related to enhanced snow
cover. The strength of the albedo effect is also influenced by the increased ET causing enhanced cloud cover and precipitation. The atmospheric responses in all variables strongly depend on the shrub and tree heights. However; increased LAI leads to a persistent increase in LH in all areas with shrub expansion, in all seasons investigated.

We find that the effects of increased shrub and tree cover are more sensitive towards snow cover variations than summer temperatures. Increased shrub cover has the largest effect in spring, leading to an earlier onset of the melting season particularly in the warm spring season, representing a positive feedback to warm spring temperatures. Taller vegetation influences summer temperatures more than spring temperatures in most areas, and the response is not affected by summer temperatures to any large degree but rather seems to be a robust
signal across inter-annual variations in summer temperatures.

Summer temperatures have been estimated to be one of the strongest drivers of vegetation expansion in high latitudes. Here, we find that the strongest impacts on the summer temperatures are related to the expansion of taller vegetation rather than shorter shrubs. Due to large areas with small elevation gradients within this domain
as well as the rest of the circumpolar tundra covered areas, the temperature zones as derived here are highly sensitive to increases in summer temperatures. Small increases in mean temperatures will as such make vast areas climatically available for shrubs and tree growth. Our results show that the positive feedback to summer temperatures induced by increased tall shrub and tree cover is a consistent feature across inter-annual differences in summer temperatures. In combination with the vast area that is made available for taller shrubs and trees by
relatively small increases in temperature, this represents a clear potential for a so-called vegetation-feedback tipping point, which we find to be an interesting subject for further research. s

**Competing interests**

The authors declare that they have no conflict of interest





**Acknowledgements**

This work is part of LATICE which is a strategic research area funded by the Faculty of Mathematics and Natural Sciences at the University of Oslo. Discussions and collaboration with members of LATICE has greatly improved this manuscript. In particular we thank Dr. James Stagge for his valuable advice regarding the statistical analysis.

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





**Table 1: Key parameterizations used in the model setup.**

| Parameterization scheme | Reference |
| --- | --- |
| Mellor–Yamada–Janjić planetary boundary | (Janjic, 1994) |
| Morrison two moment microphysics | (Morrison et al., 2009) |
| RRTMG short- and longwave radiation options | (Iacono et al., 2008) |
| Noah-UA land surface model | (Wang et al., 2010) |





**Table 2. Mean response in surface fluxes and near surface atmospheric variables as averaged over all areas with vegetation changes.**

| | RefVeg mean value | | Δ Veg0K-RefVeg | | Δ Veg1K-RefVeg | |
|---|---|---|---|---|---|---|
| | MAM | JJA | MAM | JJA | MAM | JJA |
| | *(Warm,Cold)* | *(Warm,Cold)* | *(Warm,Cold)* | *(Warm,Cold)* | *(Warm,Cold)* | *(Warm Cold)* |
| **Near surface temperature [K]** | -5.77 | 10.02 | 0.10 | 0.05 | 0.23 | 0.16 |
| | (-4.28 , -7.25) | (11.0, 9.06) | (0.13,0.07) | (0.06, 0.03) | (0.28, 0.18) | (0.16, 0.15) |
| **Upward sensible heat flux [W m⁻²]** | 0.3 | 52.3 | 0.8 | 1.8 | 1.9 | 4.2 |
| | (0.1, 0.5) | (59.2, 45.5) | (1.1, 0.6) | (2.2, 1.5) | (2.4, 1.3) | (4.5, 3.8) |
| **Upward latent heat flux [W m⁻²]** | 6.1 | 33.7 | 2.3 | 2.5 | 3.7 | 3.8 |
| | (7.7, 4.5`) | (34.7, 32.7) | (2.3, 2,3) | (2.8, 2.2) | (3.7, 3.7) | (4.2, 3.5) |
| **Net short wave down [W m⁻²]** | 54.2 | 153.2 | 2.45 | 3.6 | 4.93 | 7.22 |
| | (60.2, 48.3) | (165.4, 141.0) | (3.18, 1.73) | (4.26, 2.99) | (5.98, 3.88) | (7.86, 6.58) |
| **Net Long wave down [W m⁻²]** | -38.0 | -55.45 | 0.35 | 0.64 | 0.60 | 0.47 |
| | (-40.3, -35.7) | (-60.8, -50.1) | (0.09, 0.60) | (0.59, 0.69) | (0.16, 1.04) | (0.53, 0.42) |
| **Precipitation\* [mm day⁻¹]** | 5865 | 8446 | 1.07% | 2.2% | 2.5% | 4.3% |
| | (6496, 5234) | (8090, 8801) | (1.1%,1.01%) | (2.4%, 2.06%) | (2.7%, 1.6%) | (5.0, 3.7)% |
| **Snowfall\* [mm day⁻¹]** | 4477 | 274 | 1.3% | 2.3 %\*\* | 2.8% | 3.0%\*\* |
| | (4289, 4666) | (328, 220) | (1.5%, 1.3%) | (3.04%, 1.4%)\*\* | (3.0%, 2.4%) | (3.5%, 1.2%)\*\* |
| **Low cloud coverage (<3km) [fraction]†** | 0.31 | 0.16 | 1.92% | 0.81% | 3.2% | 0.71% |
| | (0.29,0.29) | (0.14, 0.19) | (2.06%, 1.85%) | (1.0%, 0.7%) | (3.3%, 3.4%) | (1.0%, 0.5%) |
| **Vegetation buried by snow [fraction]** | 0.87 | 0.01 | -0.42 | - | -0.52 | - |
| | (0.78, 0.95) | (0.00, 0.02) | (-0.43, -0.42) | | (-0.49, -0.55) | |

*\*accumulated values over areas with vegetation changes, \*\*not statistically significant ,†average fraction over model layers below 3km*





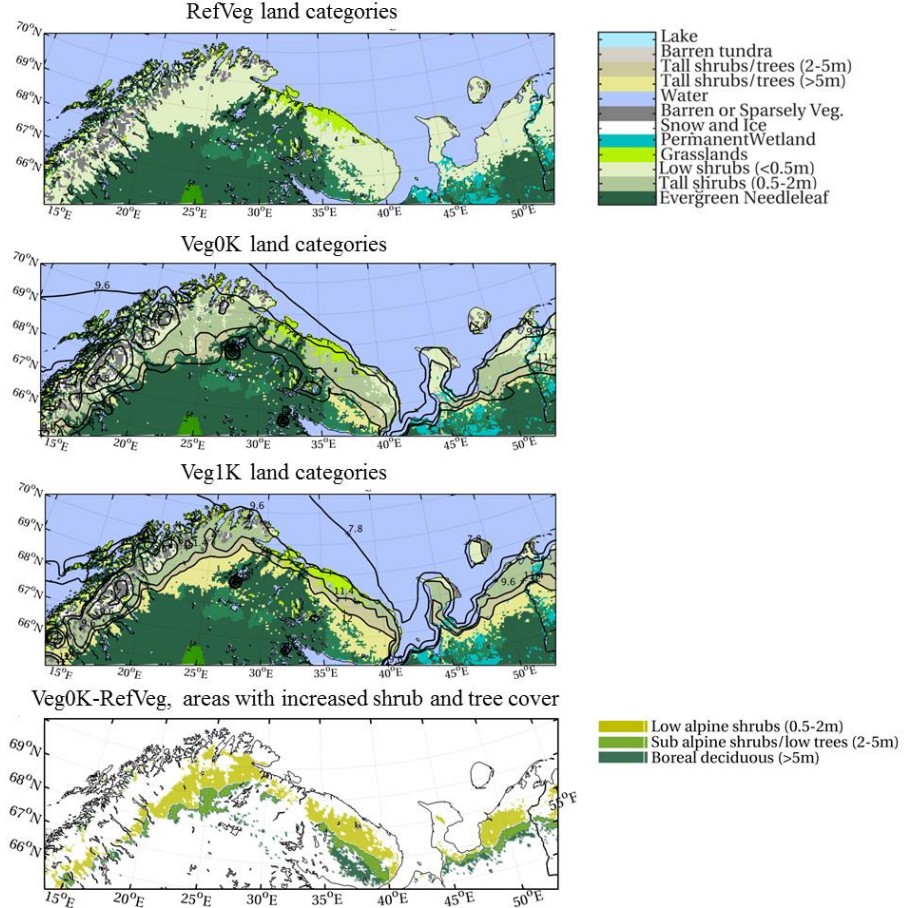

**Figure 1: Dominant land use categories (colors) in the reference simulations (RefVeg) (top panel), and as distributed according to 2 m temperature limits (contour lines) in each of the perturbed simulations. Re-distributed vegetation according to present day climatic envelopes (Veg0K) in the second panel, and future scenario (Veg1K) based on a 1 K temperature increase in the third panel. Only the temperature contour lines calculated to distinguish between the various alpine zones are shown. In the bottom panel, only areas with increased shrub and tree cover are colored, given as the difference in vegetation cover between the Veg0K and the reference simulation.**



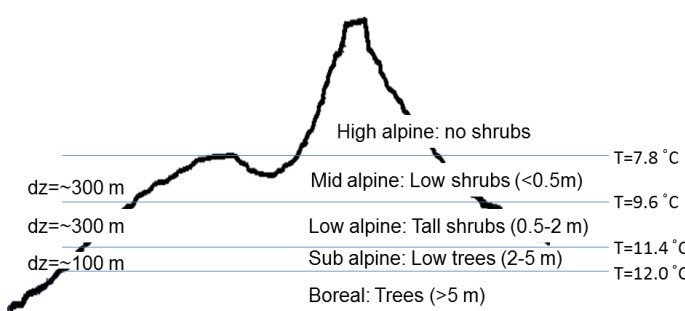

**Figure 2: Illustration of alpine zones and corresponding dominating shrub vegetation. The altitudinal extent of each alpine zone is indicated by the values of elevation differences (dz), and corresponding mean JJA temperatures dividing the zones based on mean summer lapse rates in the area.**

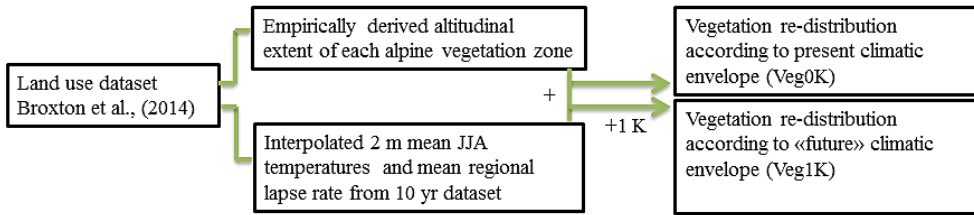

**Figure 3. Illustration of procedure applied to re-distribute the alpine vegetation across the study domain.**



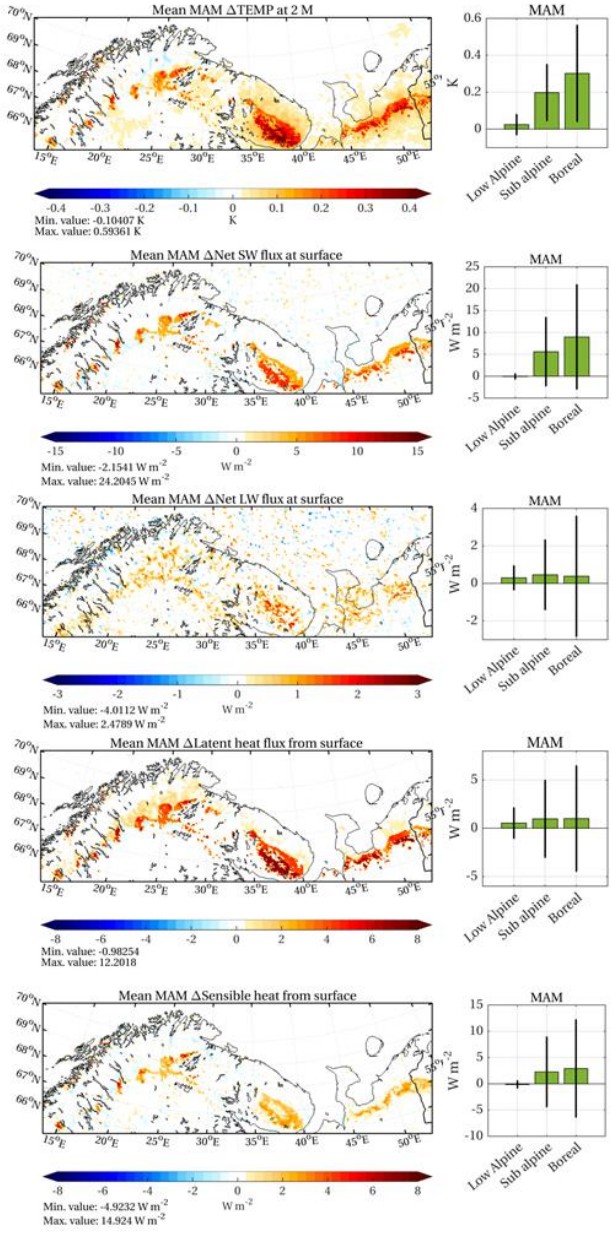

**Figure 4: Effects of increased shrub cover (Veg0K-RefVeg) on the MAM season 2 m temperature (top panel) and surface fluxes of net SW and LW radiation (direction downward) and fluxes of SH and LH (upward from surface). The minimum and maximum seasonal mean values are shown below each map, to present the full spatial variations in the average seasonal response. Colors only show significant results at the 95% confidence level based on a Mann-Whitney test of equal medians. Bar plots indicate the mean response as averaged over the separate areas with vegetation changes (black lines indicate one σ range about the mean). Note that scales differ among variables.**



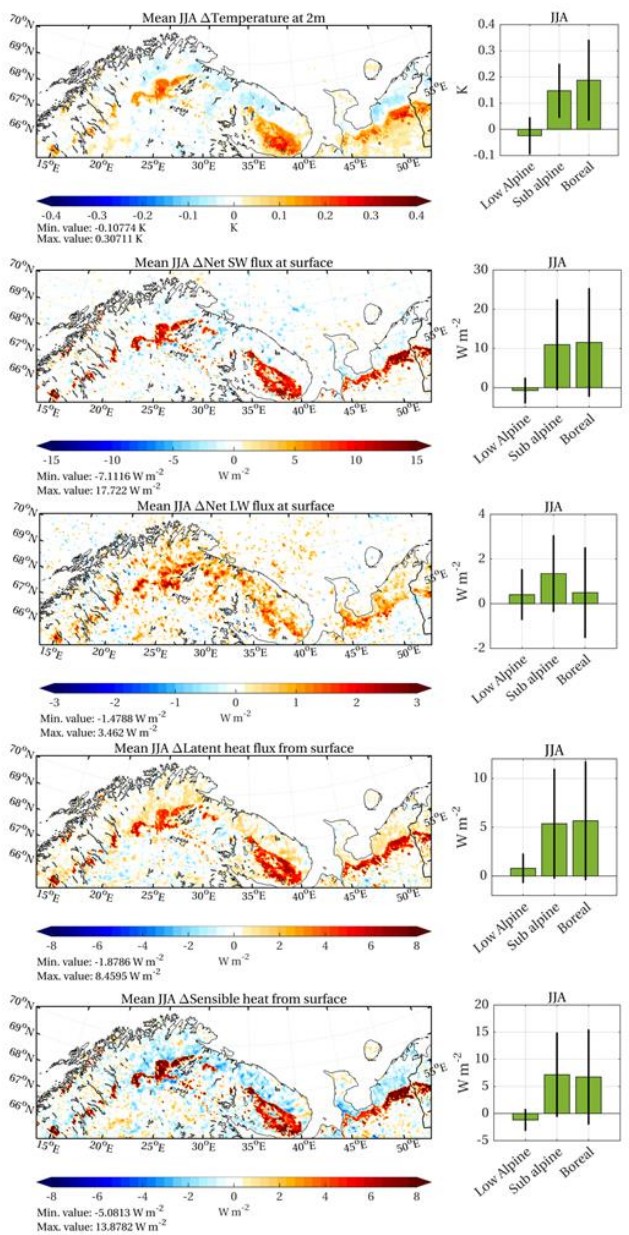

**Figure 5. Effects of increased shrub cover (Veg0K-RefVeg) on the JJA season 2 m temperature (top panel) and surface fluxes of net SW and LW radiation (direction downward) and fluxes of LH and SH (upward from surface) (Only showing significant results at the 95% confidence level, as in Fig 3). Bar plots indicate the mean response as averaged over the separate areas with vegetation changes (black lines indicate one σ range about the mean). Note that scales differ among variables.**





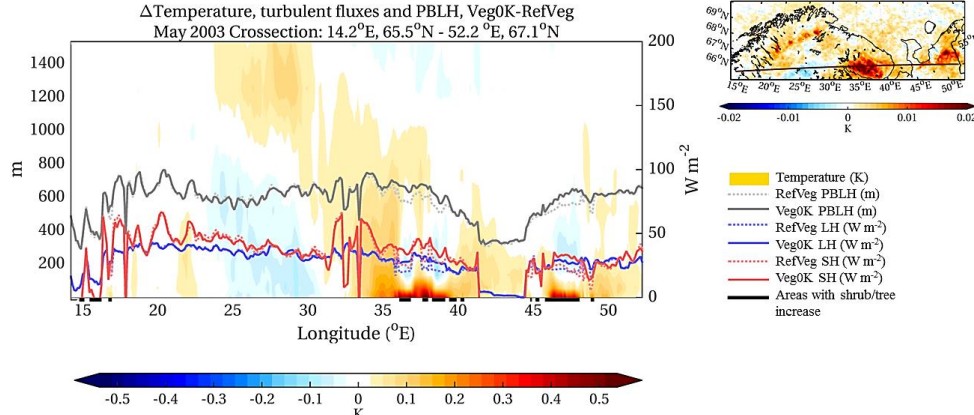

**Figure 6: Cross section showing anomalies resulting from increased shrub cover (Veg0K-RefVeg) as averaged over the warm May month. The air temperature anomaly ΔT is shown in colors, and PBL height in grey lines, left axis. Blue and red lines show LH and SH, respectively (in W m⁻², right axis). Stippled lines show RefVeg values, and solid lines show Veg0K values. Black lines along the bottom line indicate areas with shrub expansion along the cross section. The placement of the cross section line is shown in the inset, along with mean monthly bottom model layer air temperature anomalies.**





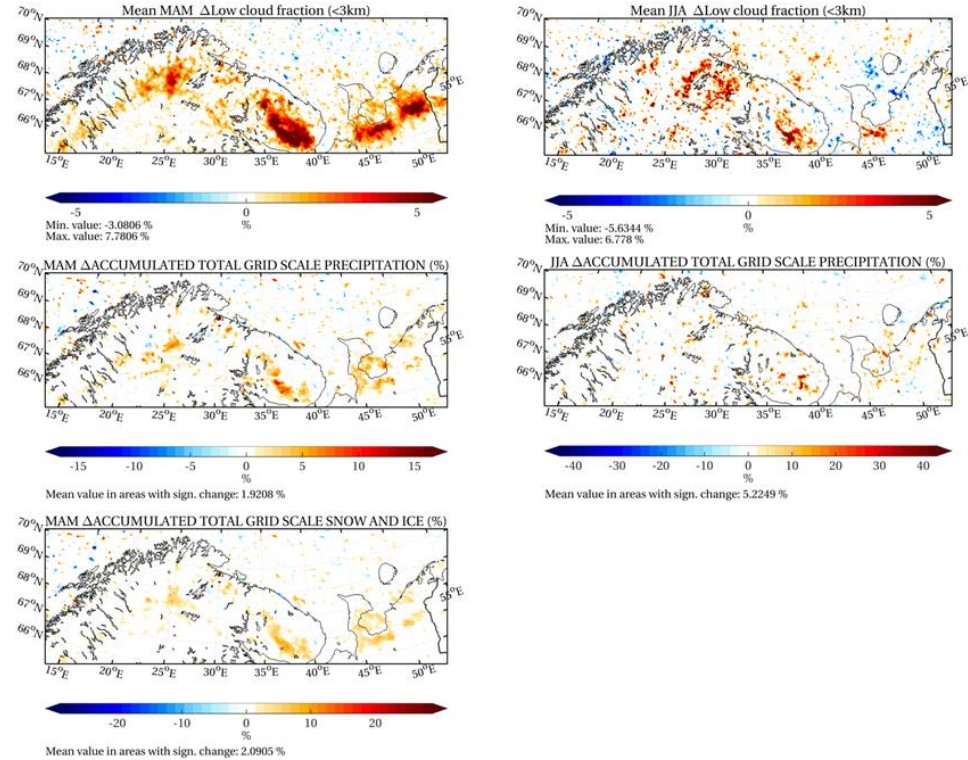

**Figure 7. Mean seasonal effects on low level (< 3 km) cloud cover fraction (top row), relative change in accumulated seasonal precipitation (middle row) and spring season snow and ice precipitation (bottom panel). Only showing significant changes at the 95% confidence level, as in Fig. 4. For precipitation, significance tests are conducted on daily values of accumulated precipitation, rather than three-hourly values. Mean over spring seasons in left column, and summer seasons in right column. Note that scales differ among panels.**

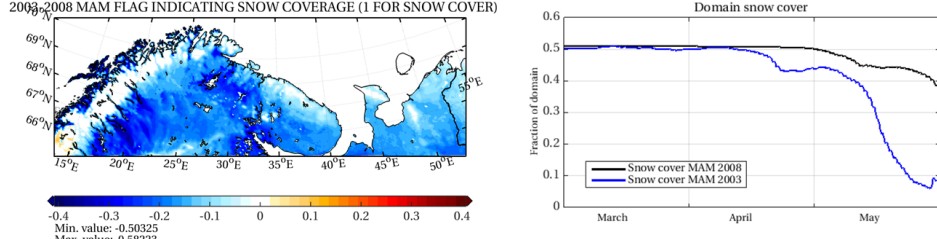

**Figure 8: Difference in mean seasonal snow cover between the warm and cold spring season (RefVeg$_{warm}$ −RefVeg$_{cold}$). Mean seasonal spatial differences are shown in the left panel, and the temporal development over the seasons in the right.**





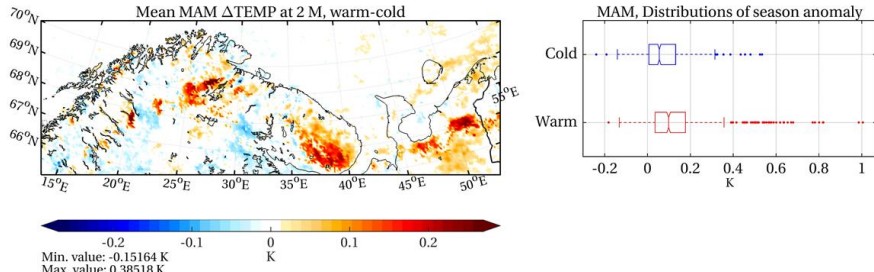

**Figure 9: Difference (warm-cold) in temperature response due to increased shrub cover (Veg0K-RefVeg) (only showing significant results at the 95% confidence level). The distributions of shrub induced anomaly values are shown in box plots, the red box shows warm season anomalies and blue box cold season anomalies in areas with vegetation changes.**





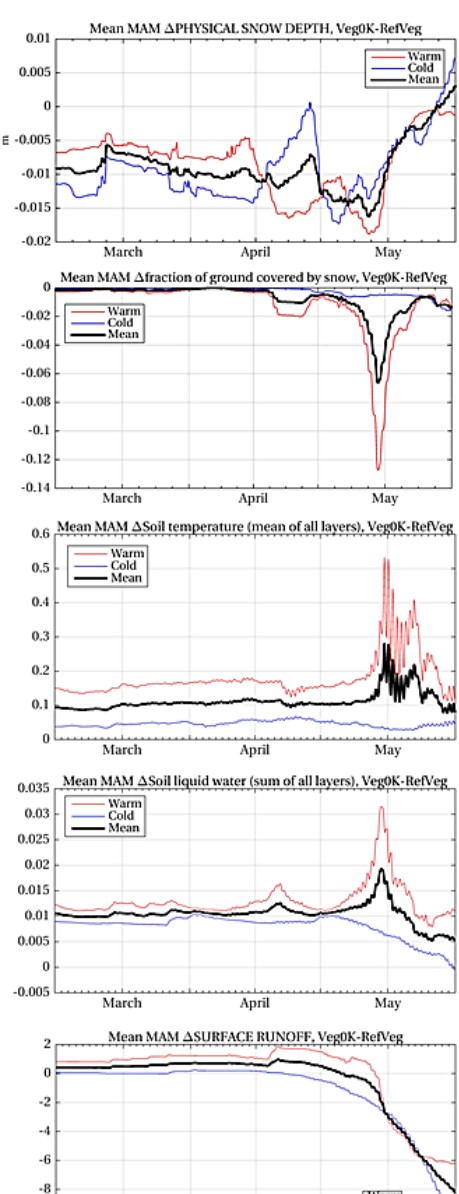

**Figure 10: Effect of increased shrub cover on spring snow depth and cover, soil temperatures and moisture content and surface runoff, as averaged over all areas with vegetation changes. Red and blue lines indicate warm and cold season response, respectively. Black lines indicate inter-seasonal means.**





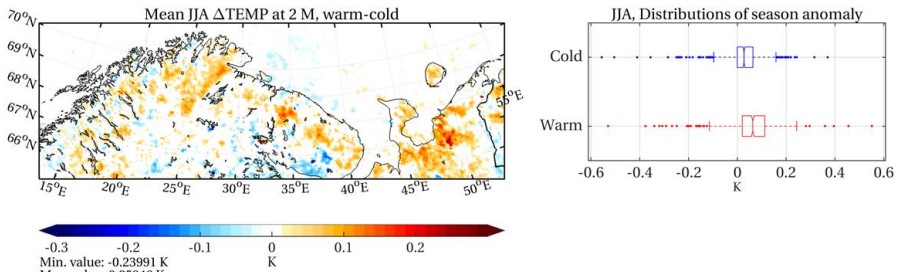

**Figure 11: Difference (warm-cold) in temperature response due to increased shrub cover (Veg0K-RefVeg). The anomaly distribution across the domain is shown (right panel), red box shows warm season anomalies and blue box cold season anomalies in areas with vegetation changes.**

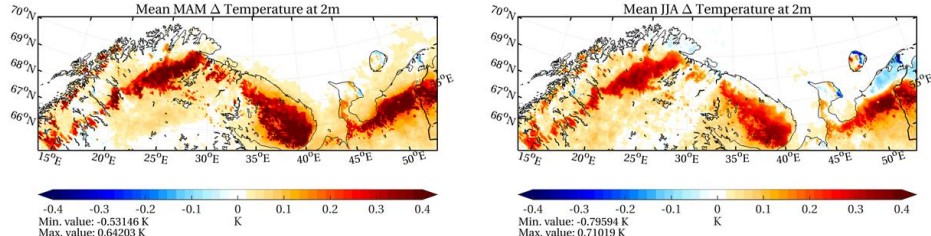

**Figure 12: Effect on the 2 m temperature resulting from a shrub and tree cover increase corresponding to a 1 K warming of JJA temperatures (Veg1K-RefVeg) (only showing significant results at the 95% confidence level, as in Fig. 4). Mean spring season is shown in the left panel and summer season in the right.**

