# Peer review of "Effects of shrub cover increase on the near surface atmosphere in northern Fennoscandia"

_Biogeosciences, 2016_

## Referee Comment (RC1) · Anonymous Referee #1 · 28 Nov 2016

Rydsaa et al. present an interesting investigation into changes to the near-surface atmosphere resulting from vegetation change, with a particular focus on near-surface temperatures. They use the WRF model to compare simulations where the vegetation extent is (i) based on present day climate distributions, and (ii) relating to a 1K increase in summer temperatures, relative to the current vegetation distribution. The authors also considered the sensitivity of this response to inter-annual variability. Based on the results presented in this manuscript, the authors suggest that tall shrubs are key to a summer warming feedback but that the main impact of shrub expansion is on advancing the onset of snowmelt in the spring, thus inducing a positive feedback to spring temperatures. In terms of inter-annual differences, the authors propose that

their results show that the warm summer-tall shrubs feedback is consistent across warm and cold seasons. They finish by proposing that their findings show "a clear potential for a so-called vegetation-feedback tipping point".

In order for this paper to be acceptable for publication in Biogeosciences, the following major revisions are required:

1. Ideas are not always introduced in a logical manner, and the text is frequently hard to follow. Moreover the writing style needs to be more concise to improve clarity and flow of ideas (See detailed line-by-line comments for examples). It would also be nice to see some hypothesis or specific research questions clearly stated in the introduction and these used to structure the subsequent results and discussion. This would greatly facilitate the overall readability and coherence of the work, but will require some major structural changes to the manuscript to be achieved.

2. Central to this work is the assumption that published climate envelopes for vegetation types in Norway are sufficient to predict changes in future vegetation when mean summer temperatures are 1K warmer. However, the authors show significant discrepancies between the present-day vegetation distribution and that expected from the climate envelopes based on present day climate. This suggests that either (i) the climate envelopes are inappropriate, or (ii) the present-day vegetation is out of equilibrium with the present climate – perhaps due to warming that has already occurred. The former case presents obvious difficulties for the use of these climate envelopes. In the second case, it seems somewhat unrealistic that the vegetation will have had time to adjust to the scenario indicated by the climatic envelopes under 1K warming, given the timeframe over which 1K warmer summer temperatures will be achieved. As it stands, this study is limited in how much can be drawn from the 'future' distribution based on the climatic envelopes associated with a 1K increase. There needs to be significantly more discussion of the limitations of using climatic envelopes and justification of their use.
3. It is stated that only the vegetation distributions are altered and that the simulations are identical in all other respects. This suggests that your model is being forced with present day climate for both the reference and 1K runs, i.e. you are measuring the strength of the vegetation feedback under today's climate. However, of interest is the strength of the vegetation feedback in a 1K warmer climate. To assess this, surely the driving meteorological data need to reflect this 1K warming (and associated changes in winter climate and snow cover)? I suggest that you run a new 1K reference simulation with the present-day vegetation distribution and met data reflecting the 1K warming. This would still allow you to isolate the vegetation feedback, under a more meaningful scenario.

4. The introduction highlights the importance of soil properties in determining the distribution of shrubs and their response to changes in climate, in particular drawing attention to soil moisture content, in addition to mean summer temperatures, as key drivers of shrub expansion. This study explores the effects of increased summer temperatures (by 1K). However, it is not clear that the influence of soil properties is taken into account regarding the perturbed vegetation simulations, although this will surely be an important constraint on future distributions.

Detailed comments

Abstract

P1L11 Specify that you are evaluating the sensitivity of near surface atmosphere / temperatures

P1L12 Specify that these are model experiments

P1L21 Shortwave radiation instead of SW would be clearer

P1L22 "shrub and tree heights, which lower the surface albedo"

P1L28 Be more specific –a role in what?

Introduction

P1 L34-6 Writing style needs to be more concise

P2 L6 You mention biochemical effects here but nowhere else. Can you provide some references for this opening sentence?

P2 L7-8 The increase in radiation absorption is due to the decrease in surface albedo –please structure the sentence to reflect this.

P2 L12 "influence the melt and sublimation" this sentence is quite vague –can you specify what this influence is (i.e. does it enhance or reduce melting/sublimation)

P2L15-16 Provide a reference for this.

P2 L16 Missing punctuation.

P2 L21 "speed the melting season" –this is unclear, do you mean that the onset of melting is advanced or that the melting season is shorter and more intense?

P2 L24-39 Provide more information regarding the climate scenarios these vegetation increases related to (i.e. how many degrees increase in temperature) –this will provide better context for your own study

P3 L4 coupled "with" not "to"

P3 L9-12 Please provide a reference(s) to support/illustrate this

P3 L31 "Also based on dendroecological observations in northern Scandinavia" is misleading as the study discussed just previously (Myers-Smith 2015) was based on data from across the circumpolar region, not just northern Scandinavia.

P4 L 6-7 This spatial resolution cannot resolve "fine scale features of vegetation change", which will be occurring on much smaller scales than the model grid size

P4 L12-18 This paragraph could be clearer in its presentation of the main aims of the study. In the methodology you state that differences between seasons are of particular

interest, but this is not mentioned in this paragraph. You should list the hypotheses that are tested in the manuscript here (this will also help provide a structure to your discussion section, relating yours results to the research questions).

Methodology

P4 L22-24 This needs to be explained more clearly, i.e. how do the vegetation change simulations differ from one another

P4 L36 Requires references to said studies

P6 L10-11 "Alterations in the atmosphere results from the biophysical changes related to the applied vegetation perturbations alone" –does this mean that in simulations where the vegetation is prescribed based on a 1k increase in summer temperature, you do not adjust the forcing met data to reflect this?

P6 L19-23 This is not clear

P6 L19 The authors define vegetation categories according to "empirically derived climatic vegetation zones" –they cite Bakkestuen et al 2008 who develop a model for vegetation variation in Norway. It is not clear during the methodology section whether the "empirically derived climatic vegetation zones" are from Bakkestuen et al. 2008 or are derived by the authors.

P7 L4 Should be "e.g. see"

P7 L23 Specify the increase in JJA 2 m temperatures applied

P7 L34 I'm not sure what you mean here

P7 L36-38 Shouldn't the forcing met data reflect the temperature increase in the Veg1K simulation?

Results

P8 L2-3 This should be explained properly in the methodology

P8 L8-10 Refer to table 2 here

P8 L25 Remove "also"

P8 L27 Refer to specific plots, e.g. using lettered plots

P8 L35 Reference to Fig. 7 not Fig. 6

P9 L4 Reference to Fig. 4 not Fig. 3

P9 L14 "These areas" –do you mean areas with low alpine shrub expansion?

P9 L22-23 "... the small albedo decrease associated with the low-alpine shrub increase. The areas with taller shrubs and trees on the other hand, are characterised by a decrease in snow cover throughout the spring and summer seasons due to a stronger albedo decrease (Fig. S4)" –I can't see how this figure specifically shows the different albedo effects associated with these vegetation types (i.e. low-alpine shrub vs. tall shrub and tree)

P9 L27 Add parenthesis

P9 L29-30 "The increased SH mainly acts to heat the lower atmosphere within the boundary layer, while the LH is also released above the PBL height" –it's not clear where this result comes from

P9 L37-39 The figures you refer to do not show the results you present in the manuscript text here (net not incoming SW and LW)

P10 L3 You refer to "increased shrub cover", do you also mean increased tree cover here too? Later on in this paragraph you refer to "vegetation changes" and on L38 you talk about "increased shrub and tree cover" –are you using these three phrases interchangeably or do you mean something different in these instances? It is not clear. Also, please specify which simulation these results are from (I presume Veg0K – RefVeg).

P10 L 5-6 It is not clear from just looking at Fig. 7 that the low cloud cover increase is

predominantly occurring over areas of vegetation change –you should at least refer to the bottom panel in Fig. 1 that shows this or include it again in the empty plot of Fig. 7 for easy reference.

P10 L8 Provide p value

P10 L13-14 "This indicates". If the only thing that you changed between the two simulations was the vegetation cover, then surely then all the precipitation change must be attributable to this? What else would have caused it if all other variables were kept constant?

P10 L15-18 What do you mean by "summarized"? Looking at Table 2, the 2.2% increase is the averaged change over all areas with vegetation changes and over both cold and warm summer seasons. You only provide the warm spring increase in precipitation (1.1%), why not the change averaged over both warm and cold seasons in this case? This is not consistent with your presentation of the summer result. The 1.4% increase in snow and ice appears to be from the cold summer season (looking at Table 2), why did you select this specific value? Per Table 2 this is not a statistically significant finding, yet the p value you present in this paragraph is (p = 3.19 x 10-9).

P10 L20-25 Be clear that you are discussing the RefVeg simulations here

P10 L 22 "and a 3.1K warmer 2 m temperature, on average." Looking at Table 2, the difference seems to be 2.97K ?

P10 L27-28 Be clear that you are discussing the RefVeg simulations here

P10 L28 Be more specific –how many days earlier?

P10 L39 Spatial pattern of snow cover shown by Fig. S3 not the spatial pattern of snow depth

P11 L11 Please indicate the subplots of interest here, e.g. Fig. 10 c & d

P11 L21-26 The temperature values presented here do not match those given in Table

2. I presume this is because the Table values are means over areas with vegetation change only whereas in this paragraph you are presenting values for the entire model domain –is this correct? It is not clear why you are not consistent here, particularly as for precipitation you refer to Table 2 rather than continue with giving whole-domain values.

P11 L 36-38 If snow conditions are important then surely you need to take into account that temperature increases during the winter months are projected to be much greater than 1K in your Veg1K simulations.

P12 L2-4 This would benefit from a figure illustrating this change in the vegetation, i.e. such as in the bottom row of Fig. 1

P12 L11-12 This is confusing as both vegetation simulations (Veg0K and Veg1K) represent a "future" scenario? Or at least, Veg0K does not reflect the present-day vegetation distribution but the distribution that one would expect given our present-day climate.

P12 L14-16 The second sentence ("Therefore average spring season heating is therefore strongest in areas with the tallest vegetation") does not logically follow on from the sentence preceding it. Also, over-use of "therefore".

P12 L16-18 But what about the spread of temperatures. So, the highest temperature is found in Summer but in Fig 12 it looks as though this is very localised and that during the Spring more of the domain experiences higher temperatures.

P12 L20 Why is this not included in the supplementary material?

P12 L28 Start your discussion off with a summary of your major findings in the order they were presented as hypotheses in the methods section.

P13 L 17-19 Needs re-wording

P13 L21 The authors don't change the greenness factor of each grid cell between the simulations, this seems strange as you would expect a 'greening' effect with the 1K

increase?

P13 L30 "…areas with increased tall vegetation"

P13 L31 "…related to increased low shrub cover…"

P13 L32 "enhanced" rather than "added". What is the balance between these two factors during the spring season?

P13 L 36 What do you mean by "despite the snow masking effect in winter"? Also, you did not analyse winter months in this study. The final part of this sentence doesn't make sense: "the deciduous nature of the northward expanding shrubs and trees in this study, which is based on what is observed in the study region"

P13 L38-40 They haven't allowed for expansion of needle leaved trees –is this reasonable?

P14 L18 Clarify that you are talking about changes in SW and 2 m temperature, i.e. a reduction in early summer

P14 L21 Remove "also"

P14 L26 "we note that they observed a substantially larger response in soils temperatures than was shown in our results"

P14 L36-37 Can you provide the equivalent percentage shrub increases for your simulations to aid comparison with the studies discussed here?

P15 L1 "The response of shrub expansion" –this doesn't make sense; the response of what to shrub expansion?

P15 L2-3 More moderate than what?

P15 L6 Remove "were related to" and change to "are occupied by"

P15 L18 Change 0.04 to 0.05 as per Table 2

P15 L14 "we find" This is the first time a time lag is mentioned. What sort of time-lag –provide an estimate.

P15 L33 Remove "have"

P15 L34 Change "in evaluating" to "to evaluate"

P15 L35 Remove "have"

P16 L12 You stated in your results that this increased snow cover was due to "increased snow fall in the cold season and possibly the increased shading effect of the shrubs" whereas here you are implying it is entirely down to shrub expansion –this seems to be a contradiction.

P16 L35-36 Repetition of P15 L31-33

Figures

Figure 1. The following changes would improve the clarity of this figure:

Labelling sub-plots (e.g. a, b, c, d) would be beneficial for clearer linking between the text and the figure (this comment applies for further figures also).

The temperature limits shown by the contour lines are unclear.

The axis labelling of the fourth subplot is slightly inconsistent with the others.

Why does the vegetation classification change in the final subplot? For instance in the top panels, Tall shrubs are 0.5-2m whereas shrubs of this height are classed as Low alpine shrubs in the bottom panel.

It would be helpful to see a subplot like the one at the bottom of this figure for the other vegetation perturbation, i.e. Veg1K – RefVeg. Why have you not included this?

Figure 2. You have shown temperatures in degrees Celsius here but throughout the manuscript you refer to temperature changes in Kelvin, this should be kept consistent.

Figure 3. If you re-ordered your figures so that figures 2 and 3 precede Figure 1, this would make Figure 1 clearer as it reflects the order that you present the related concepts in the manuscript text.

Figure 5. Refer to Fig. 4 not Fig. 3

Figure 6. This Figure is too small and trying to show too much, which makes it unclear and difficult to extract the key information.

Why did you use that particular cross section?

It is hard to see the stippled lines

What is the difference between the inset scale and the main figure scale? Why are they not consistent?

Figure 7. The middle row of figures would be easier to interpret if the scales were the same.

Not clear from the figure caption which simulation we are looking at (i.e. Veg0k – RefVeg)

Figure 8. On the right plot, label the two seasons as 'warm' and 'cold' or 'RefVegwarm' and 'RefVegcold' instead of the year, as this is how you refer to them elsewhere in the manuscript.

The title of the left plot is too long and so is hard to read where it overlaps the left axis labels.

Figure 11. In the all other figures showing anomaly distributions across the domain, you state that the figure is "only showing significant results at the 95% confidence level" –why do you not do that here?

Figure 12. Inconsistent use of "2 M" and "2m" throughout figures in the manuscript

Specify that these are inter-seasonal means

Table 2. Responses are presented inconsistently: Why are precipitation, snowfall and low cloud coverage changes shown as percentages whereas the actual values are given for the other variables? Why is the mean value for RefVeg MAM Low cloud coverage 0.31 when the mean values for the warm and cold years are 0.29 and 0.29? (this applies to other values presented in the table) Why are you only averaging over areas with vegetation changes? Cloud cover and precipitation effects might not be limited to the atmosphere directly above the vegetation change for instance.

Reference list

P20L1-12 Duplication of reference: Myers-Smith et al. 2015a appears to be the same paper as Myers-Smith et al. 2015b

Supplementary material

Figure S1. Consistent scales would be better

Figure S5. In the text preceding this figure, it is not clear why "(Veg0K-RefVeg)" is included at the end.

---

## Referee Comment (RC2) · Anonymous Referee #2 · 7 Dec 2016

This paper extends prior research into the potential climatic effects of a hypothetical increase in shrub and tree cover (in this case, shrub and tree height) in a region in northern Fennoscandia. My overall impression is that this is a solid paper that provides some interesting new results on the topic of vegetation feedbacks onto climate in the Arctic region. This work does not represent a giant leap forward, but the study is sound and, in my opinion, the paper is worthy of publication.

The paper can be divided into two main parts. The first part focuses on the potential changes in vegetation distribution under a 1K temperature change. The second part focuses on an assessment of the impact of such a shift in vegetation on temperature.

From my perspective the first part is fine. One could quibble with aspects of the method

and argue whether or not the projected changes in vegetation distribution are completely realistic or not, but I don't feel that the realism is really the point. The main goal is to generate vegetation distribution changes that are at least quasi-realistic and that can then be applied in the subsequent vegetation change experiment. Perhaps the only recommendation that I would have here is for the authors to be a bit more explicit about this with a statement to the effect that predicting vegetation change based primarily on a climatic envelope should just be treated as a first-order assessment of potential vegetation distribution change.

For the second part, my main recommendation to the authors is that they work to put this study into better context. Prior studies, cited in the paper have looked at the impacts of shrub and tree area expansion in models and concluded, mainly, that these vegetation changes can lead to warming. So, the authors need to clearly establish what is new from this study. I see two main areas where this is new. The first is that this is being done within a regional climate model, which allows a more detailed assessment of the response. The second new result relates to the variability in the impact across high and low snowfall years and warm and cool summers. The authors should strive to emphasize these points.

Minor points:

1. For the summer feedback, the authors note that the impact of shrubs on summer temperatures is less sensitive to the mean summer temperature (warm or cold summers) than spring temps are to high versus low snow years. That is not surprising. Nonetheless, it would be good to explain why one would think that the summer temperature sensitivity could be related to mean summer temps. 2. p. 12, line 26. 0.16K versus 0.15K is essentially the same. Shouldn't say that one season has a slightly larger response when they are effectively identical. 3. Figure 11 and other figures. It would be clearer to be specific that you are talking about warm summer seasons and cool summer seasons. Just writing cold seasons and warm seasons can lead to ambiguity about whether referring to different seasons (spring versus summer, for

example).

---

## Author Comment (AC1) · 11 Jan 2017

Firstly, we would like to express our sincere gratitude towards the two anonymous referees for taking the time to review our study, and for the constructive suggestions and comments to improve the manuscript. See point-by-point replies to the comment below (some technical comments are grouped together), answers are given below each cited comment.

Anonymous Referee #1

"Rydsaa et al. present an interesting investigation into changes to the near-surface atmosphere resulting from vegetation change, with a particular focus on near-surface

temperatures. They use the WRF model to compare simulations where the vegetation extent is (i) based on present day climate distributions, and (ii) relating to a 1K increase in summer temperatures, relative to the current vegetation distribution. The authors also considered the sensitivity of this response to inter-annual variability. Based on the results presented in this manuscript, the authors suggest that tall shrubs are key to a summer warming feedback but that the main impact of shrub expansion is on advancing the onset of snowmelt in the spring, thus inducing a positive feedback to spring temperatures. In terms of inter-annual differences, the authors propose that their results show that the warm summer-tall shrubs feedback is consistent across warm and cold seasons. They finish by proposing that their findings show "a clear potential for a so-called vegetation-feedback tipping point"."

We are glad that the reviewer finds the study interesting. The reviewer has clearly taken the time and effort to get to know our study very well, and we truly appreciate this and the following thorough review of the manuscript.

"In order for this paper to be acceptable for publication in Biogeosciences, the following major revisions are required:

1. Ideas are not always introduced in a logical manner, and the text is frequently hard to follow. Moreover the writing style needs to be more concise to improve clarity and flow of ideas (See detailed line-by-line comments for examples). It would also be nice to see some hypothesis or specific research questions clearly stated in the introduction and these used to structure the subsequent results and discussion. This would greatly facilitate the overall readability and coherence of the work, but will require some major structural changes to the manuscript to be achieved."

As the reviewer suggests, the manuscript will undergo a thorough revision with emphasis on improving the writing style and structure for improved readability. Based on the comments from both reviewers, we realize that the purpose and ideas behind this study should be more clearly presented for the readers. Although as the reviewer

points out, some large structural changes are necessary to achieve this, we agree that these changes to the manuscript are vital to put our study in better context. The reviewer kindly proposes a more clearly stated hypothesis or list of research questions in the introduction, which is a great idea and will be added. A clarification of the primary research questions motivating this study and the corresponding experimental setup is presented below;

1. The warmer climate is causing more shrubs and low trees to grow in the northern Fennoscandia area, how does this feed back to the atmosphere in the region? More specifically: a. How will the feedback be influenced by varying shrub areal extent and shrub heights? b. Which season will be more affected and experience the strongest feedback; spring or summer? c. How sensitive is the feedback to different conditions, such as snow cover or temperatures?

In order to answer these questions, we run a fine scale atmospheric model (WRF) with prescribed and manually altered perturbations to the vegetation cover, and compare the atmospheric response. The perturbations reflect a spatial increase in shrub and tree cover in addition to an increase in shrub heights in some areas. To respond to the research questions in point 1a, the existing shrub cover as described in the model, was split into three sub-categories in order to distinguish the atmospheric sensitivity to varying shrub heights. In two different experiments shrubs and low trees are re-distributed in accordance with some "simplified bioclimatic envelopes" that were derived for this study. Based on an extensive literature review, mean summer temperatures were selected as the key environmental criteria used to guide the perturbations applied to the shrub and tree cover. In order to take into account some of the uncertainties inherit in the shrubs' response to summer temperatures, one bioclimatic envelope is based on present day summer temperatures, and one based on a 1K increase in mean summer temperatures.

In order to answer the research question in point 1b, we chose to focus on spring and summer. This is based mainly on the findings of previous studies, showing that these

two seasons were the ones experiencing the largest atmospheric feedbacks resulting from shrub and tree increase. Furthermore, as the atmospheric response may vary under different climatic conditions (e.g. warm vs. cold, snow rich vs. snow poor, present vs. future), we chose to run each set of vegetation distributions for two contrasting years, spanning the natural variability across a 10-year period with respect to temperature and snow cover in the study region. This setup allows us to investigate how particular conditions influence the vegetation feedbacks (research question 1c), and based on this, to make some careful assumptions regarding what may be expected under future versus present climate conditions.

"2. Central to this work is the assumption that published climate envelopes for vegetation types in Norway are sufficient to predict changes in future vegetation when mean summer temperatures are 1K warmer. However, the authors show significant discrepancies between the present-day vegetation distribution and that expected from the climate envelopes based on present day climate. This suggests that either (i) the climate envelopes are inappropriate, or (ii) the present-day vegetation is out of equilibrium with the present climate – perhaps due to warming that has already occurred. The former case presents obvious difficulties for the use of these climate envelopes. In the second case, it seems somewhat unrealistic that the vegetation will have had time to adjust to the scenario indicated by the climatic envelopes under 1K warming, given the timeframe over which 1K warmer summer temperatures will be achieved. As it stands, this study is limited in how much can be drawn from the 'future' distribution based on the climatic envelopes associated with a 1K increase. There needs to be significantly more discussion of the limitations of using climatic envelopes and justification of their use."

As pointed out by the reviewer, we realize that our use of climate envelopes in this work has not been properly introduced and explained, and the background for each of the vegetation distributions not well enough distinguished. As mentioned in the manuscript (but not adequately emphasized), the vegetation distribution in the reference simulations are based on the MODIS land use dataset (Broxton, 2014), which represents present day vegetation distribution (which one can only speculate to which degree is in total equilibrium with present day climate or not, as several other factors are also influencing the shrub cover, as explained in the introduction). As a satellite based dataset, it is not specifically linked to summer temperatures or bioclimatic envelopes and cannot be expected to be in complete equilibrium to the 10 year mean summer temperature based bioclimatic zones developed for the purpose of this study. Although this vegetation distribution in our opinion serves well as a reference point for our vegetation perturbations, one original limitation to this dataset with respect to our study, is that it has only one shrub vegetation category representing the tundra area of interest in our study (with shrub heights of 0.5m). To study the effects of different shrub heights on the atmospheric response (research question 1.a), we found it necessary to split this shrub class into three sub-classes with different heights; sub alpine, low alpine and mid alpine shrubs. In order to distribute these across the shrub covered area as defined by the MODIS dataset, we applied the simplified "bioclimatic envelopes" based on the key criteria of mean summer temperatures, in combination with empirically derived temperature-vegetation relationships from the region (as further explained in the Methodology section).

We realize that the distinction between the use of the MODIS dataset (which is not based on bioclimatic envelopes) and the vegetation perturbations applied to it (which are based on a key bioclimatic criteria; summer temperatures which represents a simplified bioclimatic envelope) was not properly introduced in the manuscript, and appeared confusing. This will be properly explained and clarified in the revised manuscript. We hope the reviewers agree that given this distinction, there is no discrepancy in the use of bioclimatic envelopes between the reference simulation and the perturbed simulations, rather bioclimatic envelopes are used as a refinement on the original, satellite-based dataset. However, following the reviewer's suggestion, this side of the study and its potential limitations will be discussed further in the revised manuscript.

With respect to the 1K "future" scenario, we do agree with the reviewer in that results from this experiment should be interpreted with care (as highlighted in the Discussion section). The manuscript will be rewritten to emphasize this, and to more clearly highlight the actual purpose of this experiment; The vegetation perturbation produced from the 1K perturbation to the mean summer temperatures were applied as a sensitivity experiment (adding to the aforementioned 0K bioclimatic perturbations), rather than to represent a pseudo-realistic future scenario for vegetation distribution in the area. As such, we have made no assumptions of a specific timeframe for this vegetation distribution. However, as a temperature increase is a very likely part of future climate conditions in this area, the loose term "future scenario" was used to describe it, yet the aim was rather to take into account an assumption of uncertainty in the shrubs' response to summer temperatures as a key environmental condition. The description of this experiment will be clarified and the discussion reframed in this context to avoid confusion.

"3. It is stated that only the vegetation distributions are altered and that the simulations are identical in all other respects. This suggests that your model is being forced with present day climate for both the reference and 1K runs, i.e. you are measuring the strength of the vegetation feedback under today's climate. However, of interest is the strength of the vegetation feedback in a 1K warmer climate. To assess this, surely the driving meteorological data need to reflect this 1K warming (and associated changes in winter climate and snow cover)? I suggest that you run a new 1K reference simulation with the present-day vegetation distribution and met data reflecting the 1K warming. This would still allow you to isolate the vegetation feedback, under a more meaningful scenario."

We agree with the reviewer's opinion that the strength of the atmospheric feedback might vary under different climatic conditions. This was exactly our reason for choosing to simulate our reference and perturbed vegetation distribution under varying climatic conditions, by choosing two contrasting years with respect to snow cover and summer

temperatures. The contrasting response between the two years will give an indication of how sensitive the feedback strength is to variations in the climatic conditions, whether they arise as part of the natural present climate variability, or represent some future mean state or condition. However, one distinction must be made clear; although the perturbations to the vegetation distribution in the two sensitivity experiments are based on a shift in the bioclimatic envelopes corresponding to a 1K difference in mean summer temperatures, the goal of investigating feedbacks under varying climatic conditions is not particularly linked to a 1K general warming of the climate. The temperature difference between the two contrasting years chosen here more than covers the 1K temperature difference chosen to perturb the vegetation cover (as shown in the results section). Rather than representing a particular shift in temperature, it represents the natural variability in temperature in this area over a 10 year period. However, our motivation is that the difference in response particularly across the warmer versus the colder year, may serve as an indicator of potential differences in feedback mechanisms that dominate in a warmer versus a colder climate.

The reviewer suggests an approach to investigate this aspect of the feedback sensitivity, by making another reference simulation with met data reflecting a 1K warming. It is not clear whether the suggestion inherits a perturbation of met data (which could prove problematic in comparison with the other simulations in principle, as this would force the regional model with "unrealistic" meteorological conditions), or if the reviewer in fact suggests a similar approach as the one we have chosen yet apparently failed to properly communicate. We acknowledge that this side of the setup/approach has not been clearly introduced in the manuscript, and that more emphasis should be put on this aspect of the setup in both the introduction, results section and in the discussion. The different response across the two contrasting years could be more emphasized and put in closer context with the two different vegetation distributions. We hope the reviewer agrees that this will be a sound approach to answer this research question. The revised manuscript will be altered accordingly following the reviewer's comments.

[Figure]

"4. The introduction highlights the importance of soil properties in determining the distribution of shrubs and their response to changes in climate, in particular drawing attention to soil moisture content, in addition to mean summer temperatures, as key drivers of shrub expansion. This study explores the effects of increased summer temperatures (by 1K). However, it is not clear that the influence of soil properties is taken into account regarding the perturbed vegetation simulations, although this will surely be an important constraint on future distributions. "

As referred to in the introduction, both soil moisture content and summer temperatures have been shown to be important factors in regulating shrub expansion. In this study we have chosen to use only mean summer temperatures to derive our "simplified bioclimatic envelopes" used to perturb the vegetation distribution. This will be clarified in the revised manuscript. The effect of soil moisture in the system is taken into account in the model simulations, and feedback to the soil moisture is briefly shown in the results. However, in the "simplified bioclimatic envelopes" as defined and used here, the soil moisture content is not a determining factor. We completely agree with the reviewer (and cited literature) that in a more realistic future scenario it would certainly be an interesting factor to include. Also, in the study region of interest here, other factors such as herbivory by reindeers could possibly play and equally important role, however a further investigation into these factors is not the aim of the present study (as also noted in point 1 by Reviewer#2).

Detailed comments

Abstract "P1L11 Specify that you are evaluating the sensitivity of near surface atmosphere / temperatures P1L12 Specify that these are model experiments P1L21 Shortwave radiation instead of SW would be clearer P1L22 "shrub and tree heights, which lower the surface albedo" P1L28 Be more specific –a role in what?"

We agree and will alter the abstract according to the reviewer's suggestions.

Introduction "P1 L34-6 Writing style needs to be more concise"

A thorough review of the language will be conducted throughout the manuscript.

"P2 L6 You mention biochemical effects here but nowhere else. Can you provide some references for this opening sentence? P2 L7-8 The increase in radiation absorption is due to the decrease in surface albedo –please structure the sentence to reflect this. P2 L12 "influence the melt and sublimation" this sentence is quite vague –can you specify what this influence is (i.e. does it enhance or reduce melting/sublimation) P2L15-16 Provide a reference for this. P2 L16 Missing punctuation. P2 L21 "speed the melting season" –this is unclear, do you mean that the onset of melting is advanced or that the melting season is shorter and more intense? P2 L24-39 Provide more information regarding the climate scenarios these vegetation increases related to (i.e. how many degrees increase in temperature) –this will provide better context for your own study P3 L4 coupled "with" not "to" P3 L9-12 Please provide a reference(s) to support/illustrate this"

We appreciate the reviewer's suggestions and will make alterations to the manuscript accordingly.

"P3 L31 "Also based on dendroecological observations in northern Scandinavia" is misleading as the study discussed just previously (Myers-Smith 2015) was based on data from across the circumpolar region, not just northern Scandinavia."

This is a good point, and we acknowledge that this was not made clear. It will be adjusted as suggested.

"P4 L 6-7 This spatial resolution cannot resolve "fine scale features of vegetation change", which will be occurring on much smaller scales than the model grid size"

This is of course a valid objection, and this statement was written in comparison with previous, coarser scale studies in mind. However, the sentence will be rewritten for clarity and the role of resolution more appropriately addressed.

"P4 L12-18 This paragraph could be clearer in its presentation of the main aims of the

study. In the methodology you state that differences between seasons are of particular interest, but this is not mentioned in this paragraph. You should list the hypotheses that are tested in the manuscript here (this will also help provide a structure to your discussion section, relating yours results to the research questions)."

We agree with the reviewer that a more clear presentation in this paragraph will greatly benefit the total readability and re-structuring of the manuscript. As mentioned in point 1. above, a more clear structure in the presentation of both the purpose and corresponding setup will be added to the revised manuscript, and followed up throughout the presentation and discussion of the results, following the reviewer's suggestions.

Methodology "P4 L22-24 This needs to be explained more clearly, i.e. how do the vegetation change simulations differ from one another P4 L36 Requires references to said studies"

Adjustments will be made accordingly.

"P6 L10-11 "Alterations in the atmosphere results from the biophysical changes related to the applied vegetation perturbations alone" –does this mean that in simulations where the vegetation is prescribed based on a 1k increase in summer temperature, you do not adjust the forcing met data to reflect this?"

As discussed in the points above, yes, the reference and perturbed simulations are run with the same meteorological forcing to isolate the effect of vegetation perturbations on the atmosphere. However, the sensitivity of the feedbacks related to variations in climatic conditions such as the temperature are investigated by choosing two contrasting years (2003 and 2008) with respect to temperature and snow cover.

"P6 L19-23 This is not clear"

Will be rewritten for improved clarity.

"P6 L19 The authors define vegetation categories according to "empirically derived climatic vegetation zones" –they cite Bakkestuen et al 2008 who develop a model for

vegetation variation in Norway. It is not clear during the methodology section whether the "empirically derived climatic vegetation zones" are from Bakkestuen et al. 2008 or are derived by the authors."

It will be clarified in the revised manuscript that we have followed a comparable approach as Bakkestuen et al, in deriving our simplified bioclimatic zones. However, whereas Bakkestuen et al. derived their bioclimatic zones from a multitude of climatic and other variables (temperature, precipitation, snow cover, geology, topography etc), the presented zones were purely based on summer temperature.

"P7 L4 Should be "e.g. see" P7 L23 Specify the increase in JJA 2 m temperatures applied"

Adjustments will be made accordingly.

"P7 L34 I'm not sure what you mean here P7 L36-38 Shouldn't the forcing met data reflect the temperature increase in the Veg1K simulation?"

As explained above, the feedback sensitivity to temperature is investigated using the two contrasting years of met forcing data.

Results "P8 L2-3 This should be explained properly in the methodology P8 L8-10 Refer to table 2 here P8 L25 Remove "also" P8 L27 Refer to specific plots, e.g. using lettered plots P8 L35 Reference to Fig. 7 not Fig. 6 P9 L4 Reference to Fig. 4 not Fig. 3"

These are all good points, and corresponding adjustments will be made to the revised manuscript.

"P9 L14 "These areas" –do you mean areas with low alpine shrub expansion?" Yes, this will be clarified in the revised manuscript.

"P9 L22-23 ". . . the small albedo decrease associated with the low-alpine shrub increase. The areas with taller shrubs and trees on the other hand, are characterized by a decrease in snow cover throughout the spring and summer seasons due to a

stronger albedo decrease (Fig. S4)" –I can't see how this figure specifically shows the different albedo effects associated with these vegetation types (i.e. low-alpine shrub vs. tall shrub and tree)"

The spatial differences in mean seasonal albedo changes are shown in the bottom panels. Each area with vegetation changes can be recognized by comparing with e.g. Fig. 1 (bottom panel); however, we acknowledge that this might be unclear. The figure can be amended by adding numbers or a bar plot for clarity, which will separate each area with vegetation change from each other.

"P9 L27 Add parenthesis"

Will be added.

"P9 L29-30 "The increased SH mainly acts to heat the lower atmosphere within the boundary layer, while the LH is also released above the PBL height" –it's not clear where this result comes from"

This is not specifically shown here, but more a reference to general meteorological processes. This will be rewritten for clarity.

"P9 L37-39 The figures you refer to do not show the results you present in the manuscript text here (net not incoming SW and LW)"

This is not specifically shown here, but provided as additional information to explain the results. This will be clarified in the revised manuscript.

"P10 L3 You refer to "increased shrub cover", do you also mean increased tree cover here too? Later on in this paragraph you refer to "vegetation changes" and on L38 you talk about "increased shrub and tree cover" –are you using these three phrases interchangeably or do you mean something different in these instances? It is not clear. Also, please specify which simulation these results are from (I presume Veg0K – RefVeg)."

Yes, the three phrases are, somewhat confusingly used interchangeably, as the reviewer points out. We will adjust the manuscript towards a more coherent presentation of the vegetation changes applied.

"P10 L 5-6 It is not clear from just looking at Fig. 7 that the low cloud cover increase is predominantly occurring over areas of vegetation change –you should at least refer to the bottom panel in Fig. 1 that shows this or include it again in the empty plot of Fig. 7 for easy reference. P10 L8 Provide p value"

Suggested changes will be added.

"P10 L13-14 "This indicates". If the only thing that you changed between the two simulations was the vegetation cover, then surely then all the precipitation change must be attributable to this? What else would have caused it if all other variables were kept constant?"

We acknowledge that this is vague, as the referee correctly points out, the changes are caused by vegetation changes and feedback mechanisms related to these. The sentence will be rewritten to clarify this.

"P10 L15-18 What do you mean by "summarized"? Looking at Table 2, the 2.2% increase is the averaged change over all areas with vegetation changes and over both cold and warm summer seasons. You only provide the warm spring increase in precipitation (1.1%), why not the change averaged over both warm and cold seasons in this case? This is not consistent with your presentation of the summer result. The 1.4% increase in snow and ice appears to be from the cold summer season (looking at Table 2), why did you select this specific value? Per Table 2 this is not a statistically significant finding, yet the p value you present in this paragraph is (p = 3.19 x 10-9)."

The presentation of these results will be revised for clarity.

"P10 L20-25 Be clear that you are discussing the RefVeg simulations here P10 L 22 "and a 3.1K warmer 2 m temperature, on average." Looking at Table 2, the difference seems to be 2.97K ? P10 L27-28 Be clear that you are discussing the RefVeg simulations here P10 L28 Be more specific –how many days earlier? P10 L39 Spatial pattern of snow cover shown by Fig. S3 not the spatial pattern of snow depth P11 L11 Please indicate the subplots of interest here, e.g. Fig. 10 c & d"

Adjustments to the manuscript according to these suggestions will be applied.

"P11 L21-26 The temperature values presented here do not match those given in Table 2. I presume this is because the Table values are means over areas with vegetation change only whereas in this paragraph you are presenting values for the entire model domain –is this correct? It is not clear why you are not consistent here, particularly as for precipitation you refer to Table 2 rather than continue with giving whole-domain values."

The numbers presented here are not particularly related to the experiments or the areas with vegetation changes, as they are given as a summary of the mean differences in the climatic conditions between the two contrasting years in the area as a whole. Therefore, they are given as domain averages (only land areas) and not related to areas with vegetation changes specifically. We acknowledge that this should be specified more clearly in the revised manuscript.

"P11 L 36-38 If snow conditions are important then surely you need to take into account that temperature increases during the winter months are projected to be much greater than 1K in your Veg1K simulations."

This paragraph will be restructured to put more emphasis on the differences between the two contrasting years reflecting different snow cover conditions, simulated here. The temperature difference in temperature feedback resulting from the vegetation distribution perturbations across the two years will give a good indication of the sensitivity of the system in this respect and will be discussed in more depth.

"P12 L2-4 This would benefit from a figure illustrating this change in the vegetation, i.e. such as in the bottom row of Fig. 1 P12 L11-12 This is confusing as both vegetation

simulations (Veg0K and Veg1K) represent a "future" scenario? Or at least, Veg0K does not reflect the present-day vegetation distribution but the distribution that one would expect given our present-day climate."

Good point. See discussion in the points above. Clarification of this will be added to the revised manuscript.

"P12 L14-16 The second sentence ("Therefore average spring season heating is therefore strongest in areas with the tallest vegetation") does not logically follow on from the sentence preceding it. Also, over-use of "therefore". P12 L16-18 But what about the spread of temperatures. So, the highest temperature is found in Summer but in Fig 12 it looks as though this is very localized and that during the Spring more of the domain experiences higher temperatures."

This is a good observation and a more detailed discussion will be added to the revised manuscript.

"P12 L20 Why is this not included in the supplementary material? P12 L28 Start your discussion off with a summary of your major findings in the order they were presented as hypotheses in the methods section."

We appreciate the reviewer's suggestion and will rewrite the discussion in accordance with the revised introduction as suggested.

"P13 L 17-19 Needs re-wording P13 L21 The authors don't change the greenness factor of each grid cell between the simulations, this seems strange as you would expect a 'greening' effect with the 1K increase?"

The reviewer makes a good point, and we acknowledge that there are many possible ways to make perturbations to the vegetation cover within this model framework. Here, our choice of only altering the type of vegetation is that this will indirectly lead to a modelled alteration of vegetation properties related to greenness (i.e. LAI, height, shading factor etc.), all properties that we through a thorough review of the literature

have sound scientific basis for changing. Also, the perturbations made here are sufficient to give answers to the research questions in focus here. The "greenness factor" variable in this model setup is related to the overall density of vegetation in a grid cell, i.e. is based on an entirely different satellite dataset and also has a monthly varying distribution, and is used to scale other vegetation-type specific variables in time. We found it difficult to base any alterations to this distribution on empirical or theoretical arguments. Although, as the reviewer points out, one could expect a change in what this variable represents, it is hard to estimate to which the degree this would occur, and whether it would be in addition to, or instead of, the changes already taken into account. Also, another reason for making the perturbations as simple and concise as possible is the interpretability of the results, and as such, we found that changing the vegetation type (and corresponding properties) was more beneficial. However, the results are carefully interpreted with the model setup and its possible limitations to this in mind. A short discussion of this issue will be included in the manuscript, with reference to up-dated publications.

"P13 L30 ". . .areas with increased tall vegetation" P13 L31 ". . .related to increased low shrub cover. . ." P13 L32 "enhanced" rather than "added". What is the balance between these two factors during the spring season?"

We appreciate the reviewer's suggestions and corresponding adjustments to the manuscript will be added.

"P13 L 36 What do you mean by "despite the snow masking effect in winter"? Also, you did not analyze winter months in this study. The final part of this sentence doesn't make sense: "the deciduous nature of the northward expanding shrubs and trees in this study, which is based on what is observed in the study region""

This will be rephrased for clarity.

"P13 L38-40 They haven't allowed for expansion of needle leaved trees –is this reasonable?"

In this study the aim was to look at vegetation changes related to the expansion of shrubs (area and height) and low trees in the tundra region specifically. Also, the northern Fennoscandia region is dominated by mountain birch forest, which is deciduous. Whether the simultaneous northward migration of evergreen needle leaved trees would give a more realistic vegetation distribution given our climatic conditions would certainly be interesting to look at, but is besides the aim of this study. This choice of limitation will be mentioned briefly in the revised manuscript.

"P14 L18 Clarify that you are talking about changes in SW and 2 m temperature, i.e. a reduction in early summer P14 L21 Remove "also" P14 L26 "we note that they observed a substantially larger response in soils temperatures than was shown in our results""

Suggestions are appreciated and adjustments to the manuscript will be added.

"P14 L36-37 Can you provide the equivalent percentage shrub increases for your simulations to aid comparison with the studies discussed here?"

We could add an area extent or a percentage number for comparison as the reviewer suggests, however we are reluctant to do so for the following reason; The papers cited here have looked at the entire circumpolar area, and made adjustments to the vegetation cover that differ substantially in nature from the ones applied in our study. A percentage number relating the areal extent of changes to the entire study domains would as such aid little in comparing the studies, and on the contrary give rise to an unfounded expectation of similarities in the results. As the studies share a similar aim (i.e. to study the feedback effects of high latitude vegetation changes on the atmosphere), an overall comparison of the atmospheric effects are defendable. However, a more close and qualitative comparison of the applied changes in vegetation distribution between the studies could potentially be more misleading than beneficial, in our opinion.

"P15 L1 "The response of shrub expansion" –this doesn't make sense; the response of what to shrub expansion? P15 L2-3 More moderate than what? P15 L6 Remove "were

related to" and change to "are occupied by""

Will be rephrased for clarity.

"P15 L18 Change 0.04 to 0.05 as per Table 2 P15 L14 "we find" This is the first time a time lag is mentioned. What sort of time-lag –provide an estimate. P15 L33 Remove "have" P15 L34 Change "in evaluating" to "to evaluate" P15 L35 Remove "have" P16 L12 You stated in your results that this increased snow cover was due to "increased snow fall in the cold season and possibly the increased shading effect of the shrubs" whereas here you are implying it is entirely down to shrub expansion –this seems to be a contradiction."

Will add an explanation of time-lags, and rephrase the sections according to the reviewer's suggestions.

"P16 L35-36 Repetition of P15 L31-33"

Figures "Figure 1. The following changes would improve the clarity of this figure: Labelling sub-plots (e.g. a, b, c, d) would be beneficial for clearer linking between the text and the figure (this comment applies for further figures also). The temperature limits shown by the contour lines are unclear. The axis labelling of the fourth subplot is slightly inconsistent with the others. Why does the vegetation classification change in the final subplot? For instance in the top panels, Tall shrubs are 0.5-2m whereas shrubs of this height are classed as Low alpine shrubs in the bottom panel. It would be helpful to see a subplot like the one at the bottom of this figure for the other vegetation perturbation, i.e. Veg1K – RefVeg. Why have you not included this?

Figure 2. You have shown temperatures in degrees Celsius here but throughout the manuscript you refer to temperature changes in Kelvin, this should be kept consistent.

Figure 3. If you re-ordered your figures so that figures 2 and 3 precede Figure 1, this would make Figure 1 clearer as it reflects the order that you present the related concepts in the manuscript text.

Figure 5. Refer to Fig. 4 not Fig. 3

Figure 6. This Figure is too small and trying to show too much, which makes it unclear and difficult to extract the key information. Why did you use that particular cross section? It is hard to see the stippled lines What is the difference between the inset scale and the main figure scale? Why are they not consistent?

Figure 7. The middle row of figures would be easier to interpret if the scales were the same. Not clear from the figure caption which simulation we are looking at (i.e. Veg0k – RefVeg)

Figure 8. On the right plot, label the two seasons as 'warm' and 'cold' or 'RefVegwarm' and 'RefVegcold' instead of the year, as this is how you refer to them elsewhere in the manuscript. The title of the left plot is too long and so is hard to read where it overlaps the left axis labels.

Figure 11. In the all other figures showing anomaly distributions across the domain, you state that the figure is "only showing significant results at the 95% confidence level" –why do you not do that here?

Figure 12. Inconsistent use of "2 M" and "2m" throughout figures in the manuscript Specify that these are inter-seasonal means

Table 2. Responses are presented inconsistently: Why are precipitation, snowfall and low cloud coverage changes shown as percentages whereas the actual values are given for the other variables? Why is the mean value for RefVeg MAM Low cloud coverage 0.31 when the mean values for the warm and cold years are 0.29 and 0.29? (this applies to other values presented in the table) Why are you only averaging over areas with vegetation changes? Cloud cover and precipitation effects might not be limited to the atmosphere directly above the vegetation change for instance."

We appreciate this thorough review of the figures and constrictive suggestions for improving them. We realize some adjustments are needed and will be made accordingly.

Reference list "P20L1-12 Duplication of reference: Myers-Smith et al. 2015a appears to be the same paper as Myers-Smith et al. 2015b"

Will be corrected.

Supplementary material "Figure S1. Consistent scales would be better Figure S5. In the text preceding this figure, it is not clear why "(Veg0K-RefVeg)" is included at the end. "

Will be corrected.

---

## Author Response (AR1)

Firstly, we would like to express our sincere gratitude towards the two anonymous referees for taking the time to review our study, and for the constructive suggestions and comments to improve the manuscript. See point-by-point replies to the comment below, answers are given below each cited comment in italics.

**Answers for anonymous Referee #1**

Received and published: 28 November 2016

10

50

5

"Rydsaa et al. present an interesting investigation into changes to the near-surface atmosphere resulting from vegetation change, with a particular focus on near-surface temperatures. They use the WRF model to compare simulations where the vegetation extent is (i) based on present day climate distributions, and (ii) relating to a 1K increase

- 15 in summer temperatures, relative to the current vegetation distribution. The authors also considered the sensitivity of this response to inter-annual variability. Based on the results presented in this manuscript, the authors suggest that tall shrubs are key to a summer warming feedback but that the main impact of shrub expansion is on advancing the onset of snowmelt in the spring, thus inducing a positive feedback to
- 20 spring temperatures. In terms of inter-annual differences, the authors propose that their results show that the warm summer-tall shrubs feedback is consistent across warm and cold seasons. They finish by proposing that their findings show "a clear potential for a so-called vegetation-feedback tipping point"."
- 25 We are glad that the reviewer finds the study interesting. The reviewer has clearly taken the time and effort to get to know our study very well, and we truly appreciate this and the following thorough review of the manuscript.

"In order for this paper to be acceptable for publication in Biogeosciences, the following major revisions are required:

1. Ideas are not always introduced in a logical manner, and the text is frequently hard to follow. Moreover the writing style needs to be more concise to improve clarity and flow of ideas (See detailed line-by-line comments for examples). It would also be nice

- 35 to see some hypothesis or specific research questions clearly stated in the introduction and these used to structure the subsequent results and discussion. This would greatly facilitate the overall readability and coherence of the work, but will require some major structural changes to the manuscript to be achieved."
- 40 The text has now undergone a thorough revision for the purpose of increased readability. We have also adjusted and re-organized the manuscript to further clarify the focus and ideas behind the study, by 1. rewriting the introduction to better introduce the main research questions, and 2. change the headers in the methods section and re-organize the text, 3. changed order of figures in methods section and 4. change the flowchart in what is now
- 45 Figure 2. Furthermore, we have re-arranged parts of the results presentation and discussion to follow up on the structure presented in the research questions and methods section.

The language has been examined and clarified throughout the manuscript, and certain sections have been substantially shortened in the process. We sincerely hope the reviewer agrees that the revised manuscript improved and more easily accessible.

"2. Central to this work is the assumption that published climate envelopes for vegetation types in Norway are sufficient to predict changes in future vegetation when

- 5 mean summer temperatures are 1K warmer. However, the authors show significant discrepancies between the present-day vegetation distribution and that expected from the climate envelopes based on present day climate. This suggests that either (i) the climate envelopes are inappropriate, or (ii) the present-day vegetation is out of equilibrium with the present climate – perhaps due to warming that has already occurred. The
- 10 former case presents obvious difficulties for the use of these climate envelopes. In the second case, it seems somewhat unrealistic that the vegetation will have had time to adjust to the scenario indicated by the climatic envelopes under 1K warming, given the timeframe over which 1K warmer summer temperatures will be achieved. As it stands, this study is limited in how much can be drawn from the 'future' distribution based on
- 15 the climatic envelopes associated with a 1K increase. There needs to be significantly more discussion of the limitations of using climatic envelopes and justification of their use."
- As pointed out by the reviewer, we realize that our use of bioclimatic envelopes in this work has not been properly introduced and explained, and the background for each of the vegetation distributions not well enough distinguished. As mentioned in the original manuscript (but not adequately emphasized), the vegetation distribution in the reference simulations are based on the MODIS land use dataset (Broxton, 2014), which represents present day vegetation distribution (which one can only speculate to which degree is in total
- 25 equilibrium with present day climate or not, as several other factors are also influencing the shrub cover, as explained in the introduction). As a satellite based dataset, it is not specifically linked to summer temperatures or bioclimatic envelopes and cannot be expected to be in complete equilibrium to the 10 year mean summer temperature based bioclimatic zones developed for the purpose of this study. Although this vegetation distribution in our
- 30 opinion serves well as a reference point for our vegetation perturbations, one original limitation to this dataset with respect to our study, is that it has only one shrub vegetation category representing the tundra area of interest here (with shrub heights of <0.5m). To study the effects of different shrub heights on the atmospheric response (research question 1.a), we found it necessary to split this shrub class into three sub-classes with different heights; sub
- 35 alpine, low alpine and mid alpine shrubs. In order to distribute these across the shrub covered area as defined by the MODIS dataset, we applied the simplified "bioclimatic envelopes" based on the key criteria of mean summer temperatures, in combination with empirically derived temperature-vegetation relationships from the region (as further explained in the Methodology section).
- 40

We realize that the distinction between the use of the MODIS dataset (which is not based on bioclimatic envelopes) and the vegetation perturbations applied to it (which are based on a key bioclimatic criteria; summer temperatures which represents a simplified bioclimatic envelope) was not properly introduced in the manuscript, and appeared confusing.

45

This has been clarified in the revised manuscript, both in the new "Study design" section (2.1) and by adding a further explanation of this in the "Land use and re-distribution" section 2.3. Furthermore, it is illustrated in the flowchart figure (now Figure 2.)

We hope the reviewers agree that given this distinction, there is no discrepancy in the use of bioclimatic envelopes between the reference simulation and the perturbed simulations, rather bioclimatic envelopes are used as a refinement on the original, satellite-based dataset. However, following the reviewer's suggestion, this side of the study and its potential limitations has been discussed further in the revised manuscript.

With respect to the original 1K "future" scenario, we do agree with the reviewer in that results from this experiment should be interpreted with care (as highlighted in the Discussion section).

10

25

50

5

The manuscript has been rewritten to emphasize this, and to more clearly highlight the actual purpose of this experiment; The vegetation perturbation produced from the 1K perturbation to the mean summer temperatures were applied as a sensitivity experiment, rather than to represent a pseudo-realistic future scenario for vegetation distribution in the area. As such,

15 we have made no assumptions of a specific timeframe for this vegetation distribution. However, as a temperature increase is a very likely part of future climate conditions in this area, the loose term "future scenario" was used to describe it, yet the aim was rather to take into account an assumption of uncertainty in the shrubs' response to summer temperatures as a key environmental condition. However, we argue that valuable information about potential future feedbacks can be inferred by interpreting the results from the two different experiments.

The description of this experiment in the introduction and in the study design sections has been be clarified and rewritten to better reflect this purpose, rather than emphasizing the "future" aspect. In addition, the results and the discussion sections have been rewritten (both headers and text) to better present the experiment in the proper context.

"3. It is stated that only the vegetation distributions are altered and that the simulations are identical in all other respects. This suggests that your model is being forced with
present day climate for both the reference and 1K runs, i.e. you are measuring the strength of the vegetation feedback under today's climate. However, of interest is the strength of the vegetation feedback in a 1K warmer climate. To assess this, surely the driving meteorological data need to reflect this 1K warming (and associated changes in winter climate and snow cover)? I suggest that you run a new 1K reference simulation

35 with the present-day vegetation distribution and met data reflecting the 1K warming. This would still allow you to isolate the vegetation feedback, under a more meaningful scenario."

We agree with the reviewer's opinion that the strength of the atmospheric feedback might
vary under different climatic conditions. This was exactly our reason for choosing to simulate
our reference and perturbed vegetation distribution under varying climatic conditions, by
choosing two contrasting years with respect to snow cover and summer temperatures. The
contrasting response between the two years will give an indication of how sensitive the
feedback strength is to variations in the climatic conditions, whether they arise as part of the
natural present climate variability, or represent some future mean state or condition.

Although the perturbations to the vegetation distribution in the two sensitivity experiments are based on a shift in the bioclimatic envelopes corresponding to a 1K difference in mean summer temperatures, the goal of investigating feedbacks under varying climatic conditions is not particularly linked to a 1K general warming of the climate. The temperature difference between the two contrasting years chosen here more than covers the 1K temperature difference chosen to perturb the vegetation cover (as shown in the results section). Rather than representing a particular shift in temperature, it represents the natural variability in temperature in this area over a 10 year period. However, our motivation is that the difference in response particularly across the warmer versus the colder year, may serve as an indicator

5 in response particularly across the warmer versus the colder year, may serve as an indicator of potential differences in feedback mechanisms that dominate in a warmer versus a colder climate.

10

The reviewer suggests an approach to investigate this aspect of the feedback sensitivity, by making another reference simulation with met data reflecting a 1K warming. It is not clear whether the suggestion inherits a perturbation of met data (which could prove problematic in comparison with the other simulations in principle, as this would force the regional model with "unrealistic" meteorological conditions), or if the reviewer in fact suggests a similar approach as the one we have chosen yet apparently failed to properly communicate.

15

20

We acknowledge that this side of the setup/approach was not introduced clearly enough in the original manuscript, and in the revised manuscript, more emphasis has been put on this aspect of the setup in both the introduction, study design section, results section and in the discussion. The different response across the two contrasting years has been more emphasized, and the context/meaning of this investigation has been highlighted. We hope the reviewer agrees that this is a sound approach to answer this research question.

"4. The introduction highlights the importance of soil properties in determining the distribution of shrubs and their response to changes in climate, in particular drawing attention to soil moisture content, in addition to mean summer temperatures, as key drivers of shrub expansion. This study explores the effects of increased summer temperatures (by 1K). However, it is not clear that the influence of soil properties is taken into account regarding the perturbed vegetation simulations, although this will surely be an

30 important constraint on future distributions. "

As referred to in the introduction, both soil moisture content and summer temperatures have been shown to be important factors in regulating shrub expansion. In this study we have chosen to use only mean summer temperatures to derive our "simplified bioclimatic envelopes" used to perturb the vegetation distribution. This has been clarified in the revised manuscript, particularly under the section "study design". The effect of soil moisture in the system is taken into account in the model simulations, and feedback to the soil moisture is briefly shown in the results. However, in the "simplified bioclimatic envelopes" as defined and used here, the soil moisture content is not a determining factor. We completely agree with the reviewer (and cited literature) that in a more realistic future scenario it would certainly be an interesting factor to include. Also, in the study region of interest here, other factors such as herbivory by reindeers could possibly play and equally important role, however a further investigation into these factors is not the aim of the present study (as also noted in point 1 by Reviewer#2).

45

**Detailed comments**

Abstract

We agree and the abstract has been rewritten in accordance with the reviewer's suggestions. The abstract has also been revised to better fit with the restructuring of the rest of the manuscript.

5 "P1L11 Specify that you are evaluating the sensitivity of near surface atmosphere / Temperatures

Corrected

10 P1L12 Specify that these are model experiments *Corrected.*

P1L21 Shortwave radiation instead of SW would be clearer *Corrected*

15

P1L28 Be more specific –a role in what?"

20 We agree that this was vague and have adjusted the abstract accordingly. Added; "...role in land-atmosphere feedback processes..."

**Introduction**

25 "P1 L34-6 Writing style needs to be more concise"

The section has been revised, and a thorough review of the language has been conducted throughout the manuscript.

30 "P2 L6 You mention biochemical effects here but nowhere else. Can you provide some references for this opening sentence?

As the biochemical effects are not addressed in this manuscript, the phrase has been removed to avoid confsion.

35

P2 L7-8 The increase in radiation absorption is due to the decrease in surface albedo –please structure the sentence to reflect this. *Corrected*

40 P2 L12 "influence the melt and sublimation" this sentence is quite vague –can you specify what this influence is (i.e. does it enhance or reduce melting/sublimation)

Re-phrased for clarity

45 P2L15-16 Provide a reference for this. *The paragraph is rewritten for clarity, showing that this statement is referring to the findings in the Sturm paper cited.*

P2 L16 Missing punctuation.

50 *Corrected*

P1L22 "shrub and tree heights, which lower the surface albedo" *Revised in accordance with the reviewer's suggestion*

P2 L21 "speed the melting season" –this is unclear, do you mean that the onset of melting is advanced or that the melting season is shorter and more intense?

5 Both the onset and rate of snowmelt is meant, and clarified in the revised manuscript.

P2 L24-39 Provide more information regarding the climate scenarios these vegetation increases related to (i.e. how many degrees increase in temperature) –this will provide better context for your own study

10

15

The mentioned modelling studies do not put their shrub expansion experiments in relation to a specified climate scenario or temperature increase, rather that the shrub expansion is a result of some hypothetical future climatic state (the year 2100 is mentioned in one of the studies). In this respect, their approach is similar to ours, and therefore, in our opinion, these studies serve as excellent studies for comparison to ours.

P3 L4 coupled "with" not "to" *Corrected*

20 P3 L9-12 Please provide a reference(s) to support/illustrate this"

References inserted

"P3 L31 "Also based on dendroecological observations in northern Scandinavia" is
misleading as the study discussed just previously (Myers-Smith 2015) was based on data from across the circumpolar region, not just northern Scandinavia."

This is a good point, and we acknowledge that this was not made clear. It has been adjusted as suggested.

30

40

"P4 L 6-7 This spatial resolution cannot resolve "fine scale features of vegetation change", which will be occurring on much smaller scales than the model grid size"

This is of course a valid objection, and this statement was written in comparison with previous, coarser scale studies in mind. However, the sentence has been rewritten for clarity and the role of resolution more appropriately addressed to reflect the comparison.

"P4 L12-18 This paragraph could be clearer in its presentation of the main aims of the study. In the methodology you state that differences between seasons are of particular interest, but this is not mentioned in this paragraph. You should list the hypotheses that are tested in the manuscript here (this will also help provide a structure to your discussion section, relating yours results to the research questions)."

We agree with the reviewer that a more clear presentation in this paragraph greatly benefitted the total readability and re-structuring of the manuscript. As mentioned in point 1. above, the more clear structure in the presentation of both the purpose and corresponding setup presented above has been added to the revised manuscript, and followed up throughout the methodology section and in the presentation and discussion of the results.

50

**Methodology**

"P4 L22-24 This needs to be explained more clearly, i.e. how do the vegetation change simulations differ from one another

5 The difference between vegetation distributions have been more clearly presented and ideas behind them more appropriately introduced in both the introduction and in the study design section.

P4 L36 Requires references to said studies"

10

15

P.4, Line 38: References inserted.

"P6 L10-11 "Alterations in the atmosphere results from the biophysical changes related to the applied vegetation perturbations alone" –does this mean that in simulations where the vegetation is prescribed based on a 1k increase in summer temperature, you

do not adjust the forcing met data to reflect this?"

As discussed in the points above, yes, the reference and perturbed simulations are run with the same meteorological forcing to isolate the effect of vegetation perturbations on the atmosphere. However, the sensitivity of the feedbacks related to variations in climatic conditions such as the temperature are investigated by choosing two contrasting years (2003 and 2008) with respect to temperature and snow cover.

"P6 L19-23 This is not clear"

25

P6, line 35-38: Rewritten for improved clarity.

"P6 L19 The authors define vegetation categories according to "empirically derived climatic vegetation zones" –they cite Bakkestuen et al 2008 who develop a model for

- 30 vegetation variation in Norway. It is not clear during the methodology section whether the "empirically derived climatic vegetation zones" are from Bakkestuen et al. 2008 or are derived by the authors."
- It will be clarified in the revised manuscript that we have followed a comparable approach as Bakkestuen et al, in deriving our simplified bioclimatic zones. However, whereas Bakkestuen et al. derived their bioclimatic zones from a multitude of climatic and other variables (temperature, precipitation, snow cover, geology, topography etc.), the presented zones were purely based on summer temperature.
- 40 "P7 L4 Should be "e.g. see" Corrected

P7 L23 Specify the increase in JJA 2 m temperatures applied"

45 *Clarification added*,

"P7 L34 I'm not sure what you mean here

The sentence was unclear and unnecessary and has been removed from the revised manuscript

50

P7 L36-38 Shouldn't the forcing met data reflect the temperature increase in the Veg1K simulation?"

As explained above, the feedback sensitivity to temperature is investigated using the two contrasting years of met forcing data.

**Results**

Corrected

Corrected

"P8 L2-3 This should be explained properly in the methodology *Section has been moved up to methods section.*

Lettered plots incerted, and references updated accordingly

10

15

5

P8 L8-10 Refer to table 2 here *The section has been rewritten for clarity, and reference to Table2 inserted below.*

P8 L25 Remove "also" Corrected

P8 L27 Refer to specific plots, e.g. using lettered plots

20 P8 L35 Reference to Fig. 7 not Fig. 6

P9 L4 Reference to Fig. 4 not Fig. 3"

25

"P9 L14 "These areas" –do you mean areas with low alpine shrub expansion?" *Yes, and clarification has been added, in addition to other adjustments to the section.*

"P9 L22-23 "... the small albedo decrease associated with the low-alpine shrub increase.
The areas with taller shrubs and trees on the other hand, are characterized by a decrease in snow cover throughout the spring and summer seasons due to a stronger albedo decrease (Fig. S4)" –I can't see how this figure specifically shows the different albedo effects associated with these vegetation types (i.e. low-alpine shrub vs. tall shrub and tree)"

35

45

*The spatial distribution in mean seasonal albedo changes are shown in the bottom panels. Each area with vegetation changes can be recognized by comparing with e.g. Fig. 3.*

"P9 L27 Add parenthesis"

40 Corrected

"P9 L29-30 "The increased SH mainly acts to heat the lower atmosphere within the boundary layer, while the LH is also released above the PBL height" –it's not clear where this result comes from"

This is not specifically shown here, but more a reference to general meteorological processes. The sentence has been rewritten for clarity: *P. 9, lines 38-40 "The increased SH mainly acts to heat the planetary boundary layer (PBL), while the LH is mainly released above the PBL height. The LH therefore does not affect the 2 m temperature to the same degree as the SH, as the heat is released as the water condenses, which may well be higher up in the atmosphere."*

5 "P9 L37-39 The figures you refer to do not show the results you present in the manuscript text here (net not incoming SW and LW)"

*P.10, Lines 10-12: This is not specifically shown here, but provided as additional information to explain the results. This is clarified in the revised manuscript.*

10

"P10 L3 You refer to "increased shrub cover", do you also mean increased tree cover here too? Later on in this paragraph you refer to "vegetation changes" and on L38 you talk about "increased shrub and tree cover" –are you using these three phrases interchangeably or do you mean something different in these instances? It is not clear. Also, please specify which simulation these results are from (I presume VegOK – RefVeg)."

Yes, the three phrases are, somewhat confusingly used interchangeably, as the reviewer points out. The revised manuscript has been rewritten towards a more coherent presentation of the vegetation changes applied.

**20**

15

"P10 L 5-6 It is not clear from just looking at Fig. 7 that the low cloud cover increase is predominantly occurring over areas of vegetation change –you should at least refer to the bottom panel in Fig. 1 that shows this or include it again in the empty plot of Fig. 7 for easy reference.

**25**

35

**A reference to the maps in Fig. 3 has been inserted in the revised manuscript.**

"P10 L13-14 "This indicates". If the only thing that you changed between the two simulations was the vegetation cover, then surely then all the precipitation change must

30 be attributable to this? What else would have caused it if all other variables were kept constant?"

We acknowledge that this is vague, as the referee correctly points out, the changes are caused by vegetation changes and feedback mechanisms related to these. The sentence has been omitted from the revised manuscript as it was misleading.

"P10 L15-18 What do you mean by "summarized"? Looking at Table 2, the 2.2% increase is the averaged change over all areas with vegetation changes and over both cold and warm summer seasons.

- 40 By "summarized", we refer to the fact that rather than averaging over the areas with vegetation changes, as with the other variables, it was more informative to summarize over these areas for this particular, accumulated variable. The sentence has been rewritten for clarity in the revised manuscript.
- 45 You only provide the warm spring increase in precipitation (1.1%), why not the change averaged over both warm and cold seasons in this case? This is not consistent with your presentation of the summer result.

The number has been changed for the mean value, as suggested for consistency.

50

The 1.4% increase in snow and ice appears to be from the cold summer season (looking

at Table 2), why did you select this specific value? Per Table 2 this is not a statistically significant finding, yet the p value you present in this paragraph is  $(p = 3.19 \times 10^{-9})$ ."

The 1.4% was referring to the increase in spring average snowfall, however, Table 2 said 5 1.3%, due to a round-off error, which is now corrected. P-values for some of these numbers have been omitted in the revised manuscript, for improved readability, as they were not necessary.

"P10 L20-25 Be clear that you are discussing the RefVeg simulations here

10

15

Clarification in the form of simulation abbreviations have been added.

P10 L 22 "and a 3.1K warmer 2 m temperature, on average." Looking at Table 2, the difference seems to be 2.97K ?

These are numbers that are based on the domain as a whole (as they are not linked to the experiments as such) which has been clarified in the revised manuscript.

P10 L27-28 Be clear that you are discussing the RefVeg simulations here

20

Clarifying references to simulation names have been inserted for clarity.

P10 L28 Be more specific -how many days earlier? We refer the reader to the figure for more exact numbers.

25

P10 L39 Spatial pattern of snow cover shown by Fig. S3 not the spatial pattern of snow Depth

This is an excellent point, we have revised the manuscript to account for this.

30

P11 L11 Please indicate the subplots of interest here, e.g. Fig. 10 c & d"

Figure and references have been adjusted in accordance with the reviewer's suggestion.

35 "P11 L21-26 The temperature values presented here do not match those given in Table 2. I presume this is because the Table values are means over areas with vegetation change only whereas in this paragraph you are presenting values for the entire model domain -is this correct? It is not clear why you are not consistent here, particularly as for precipitation you refer to Table 2 rather than continue with giving whole-domain values." 40

45

The numbers presented here are not particularly related to the experiments or the areas with vegetation changes, as they are given as a summary of the mean differences in the climatic conditions between the two contrasting years in the area as a whole. Therefore, they are given as domain averages (only land areas) and not related to areas with vegetation changes specifically, as the case in the previous section. We have clarified this in the revised

"P11 L 36-38 If snow conditions are important then surely you need to take into account 50

manuscript, following the reviewer's constructive comment.

that temperature increases during the winter months are projected to be much greater than 1K in your Veg1K simulations."

The differing snow conditions are a direct result of the differing temperatures between the
warm and cold years selected here. The simulations span the whole snow accumulation
season (from November), even though only spring and summer are in focus here, and is as
such taken into account. The span in temperatures between the two spring seasons are
greater than 1K, as explained in the first paragraph in Section .3.2. This sentence is only
meant to summarize the findings in 3.2 and 3.3 by highlighting which meteorological
condition that seems to have the greater effect on the atmospheric response and is further
explained in the revised manuscript (P.12, lines 1-5).

"P12 L2-4 This would benefit from a figure illustrating this change in the vegetation, i.e. such as in the bottom row of Fig. 1

15

An additional panel is added to Fig 1(which is now Fig.3) to more clearly show the areas referred to in this section. A reference to this figure is also added.

P12 L11-12 This is confusing as both vegetation simulations (Veg0K and Veg1K) represent
a "future" scenario? Or at least, Veg0K does not reflect the present-day vegetation
distribution but the distribution that one would expect given our present-day climate."

We acknowledge that this side of the study design was not properly explained in the original manuscript, and a thorough revision has been made to better explain this in the revised manuscript (see also above, points 1 and 3).

"P12 L14-16 The second sentence ("Therefore average spring season heating is therefore strongest in areas with the tallest vegetation") does not logically follow on from the sentence preceding it. Also, over-use of "therefore".

30

35

25

The sentence has been rewritten for clarity. (P.12, lines 20-22)

P12 L16-18 But what about the spread of temperatures. So, the highest temperature is found in Summer but in Fig 12 it looks as though this is very localized and that during the Spring more of the domain experiences higher temperatures."

We completely agree with the reviewers point, and have clarified this in the revised manuscript. (P.12, lines 18-25)

40 "P12 L20 Why is this not included in the supplementary material? We found that the figure was very similar to the one with the net LW, and as such contributed little to the manuscript.

P12 L28 Start your discussion off with a summary of your major findings in the order they were presented as hypotheses in the methods section."

We appreciate the reviewer's suggestion and have re-organized and rewritten the entire discussion in accordance with the revised introduction, as suggested.

50 "P13 L 17-19 Needs re-wording

**Corrected**

P13 L21 The authors don't change the greenness factor of each grid cell between the simulations, this seems strange as you would expect a 'greening' effect with the 1K increase?"

5

40

50

The reviewer makes a good point, and we acknowledge that there are many possible ways to make perturbations to the vegetation cover within this model framework. Here, our choice of only altering the type of vegetation is that this will indirectly lead to a modelled alteration of

- vegetation properties related to greenness (i.e. LAI, height, shading factor etc.), all properties 10 that we through a thorough review of the literature have sound scientific basis for changing. Also, the perturbations made here are sufficient to give answers to the research questions in focus here. The "greenness factor" variable in this model setup is related to the overall density of vegetation in a grid cell, i.e. is based on an entirely different satellite dataset and
- 15 also has a monthly varying distribution, and is used to scale other vegetation-type specific variables in time. We found it difficult to base any alterations to this distribution on empirical or theoretical arguments. Although, as the reviewer points out, one could expect a change in what this variable represents, it is hard to estimate to which the degree this would occur, and whether it would be in addition to, or instead of, the changes already taken into account.
- 20 Also, another reason for making the perturbations as simple and concise as possible is the interpretability of the results, and as such, we found that changing the vegetation type (and corresponding properties) was more beneficial. However, the results are carefully interpreted with the model setup and its possible limitations to this in mind. A short discussion of this issue has been included in the revised manuscript, and a consideration of both greening and the opposite; a browning is inserted. 25

"P13 L30"... areas with increased tall vegetation" Corrected

P13 L31 "... related to increased low shrub cover..." 30 Corrected

P13 L32 "enhanced" rather than "added". What is the balance between these two factors during the spring season?'

35 Corrected, and a clarification and further explanation of the balance is added and lifted up to the first paragraph in the Discussion section.

"P13 L 36 What do you mean by "despite the snow masking effect in winter"? Also, you did not analyze winter months in this study. The final part of this sentence doesn't make sense: "the deciduous nature of the northward expanding shrubs and trees in this study, which is based on what is observed in the study region""

This paragraph has been rephrased for clarity. See also point below.

"P13 L38-40 They haven't allowed for expansion of needle leaved trees -is this reasonable?" 45

In this study the aim was to look at vegetation changes related to the expansion of shrubs (area and height) and low trees in the tundra region specifically. Also, the northern Fennoscandia region is dominated by mountain birch forest, which is deciduous. Whether the simultaneous northward migration of evergreen needle leaved trees would give a more

realistic vegetation distribution given our climatic conditions would certainly be interesting to look at, but is besides the aim of this study. This choice of limitation has been added to the discussion in the revised manuscript. (P 14, lines 15-25)

5 "P14 L18 Clarify that you are talking about changes in SW and 2 m temperature, i.e. a reduction in early summer

Clarified in the revised manuscript.

10 P14 L21 Remove "also" Corrected

P14 L26 "we note that they observed a substantially larger response in soils temperatures than was shown in our results""

Suggestions are appreciated and adjustments to the manuscript added.

"P14 L36-37 Can you provide the equivalent percentage shrub increases for your simulations to aid comparison with the studies discussed here?"

20

15

We could add an area extent or a percentage number for comparison as the reviewer suggests, however we are reluctant to do so for the following reason; The papers cited here have looked at the entire circumpolar area, and made adjustments to the vegetation cover that differ substantially in nature from the ones applied in our study. A percentage number

- 25 relating the areal extent of changes to the entire study domains would as such aid little in comparing the studies, and on the contrary give rise to an unfounded expectation of similarities in the results. As the studies share a similar aim (i.e. to study the feedback effects of high latitude vegetation changes on the atmosphere), an overall comparison of the atmospheric effects are defendable. However, a more close and qualitative comparison of the
- 30 *applied changes in vegetation distribution between the studies could potentially be more misleading than beneficial, in our opinion.*

"P15 L1 "The response of shrub expansion" –this doesn't make sense; the response of what to shrub expansion?

P. 13, line 32: Added: "The atmospheric response to shrub cover increase"

P15 L2-3 More moderate than what? *The paragraph has been rewritten for clarity (P. 13, Lines 32-37)*

40

35

P15 L6 Remove "were related to" and change to "are occupied by"" *Rephrased for clarity.*

"P15 L18 Change 0.04 to 0.05 as per Table 2

45 Corrected

P15 L14 "we find" This is the first time a time lag is mentioned. What sort of time-lag –provide an estimate.

The sentence has been omitted during rewriting of the Discussion section for the revised manuscript.

P15 L33 Remove "have"

5 Corrected

P15 L34 Change "in evaluating" to "to evaluate" *Corrected*

10 P15 L35 Remove "have" Corrected

P16 L12 You stated in your results that this increased snow cover was due to "increased snow fall in the cold season and possibly the increased shading effect of the shrubs" whereas here you are implying it is entirely down to shrub expansion –this seems to be

15 whereas here you are implying it is entirely down to shrub expansion –this seems to a contradiction."

This is rewritten for clarity and coherence, mainly in the results section, where the speculation with regard to the shrubs shading is omitted.

20

"P16 L35-36 Repetition of P15 L31-33" Corrected

**Figures**

- 25 "Figure 1. The following changes would improve the clarity of this figure: Labelling sub-plots (e.g. a, b, c, d) would be beneficial for clearer linking between the text and the figure (this comment applies for further figures also). The temperature limits shown by the contour lines are unclear. The axis labelling of the fourth subplot is slightly inconsistent with the others.
- Why does the vegetation classification change in the final subplot? For instance in the top panels, Tall shrubs are 0.5-2m whereas shrubs of this height are classed as Low alpine shrubs in the bottom panel.
   It would be helpful to see a subplot like the one at the bottom of this figure for the other vegetation perturbation, i.e. Veg1K RefVeg. Why have you not included this?

35

The figure has been revised in accordance with the reviewer's suggestions.

Figure 2. You have shown temperatures in degrees Celsius here but throughout the manuscript you refer to temperature changes in Kelvin, this should be kept consistent.

- 40 Although the Kelvin formulation is used in the text, where temperature differences are presented (Si units are encouraged in most journals), the figures have temperatures in Celsius for better interpretability.
- Figure 3. If you re-ordered your figures so that figures 2 and 3 precede Figure 1,
- 45 this would make Figure 1 clearer as it reflects the order that you present the related concepts in the manuscript text.
  The figure figure for the figure for t

The order of the figures has been altered in accordance with the reviewer's suggestion.

Figure 5. Refer to Fig. 4 not Fig. 3

50 Corrected

Figure 6. This Figure is too small and trying to show too much, which makes it unclear and difficult to extract the key information.

Why did you use that particular cross section?

5 It is hard to see the stippled lines

What is the difference between the inset scale and the main figure scale? Why are they not consistent?

We agree that this figure shows a lot, which was also our intention with it. As the vertical profile of the atmospheric response in various variables has not been the main focus here, we found it beneficial to include only one figure of this, and rather condensed. The particular cross section is chosen so that it intersects areas with all types of vegetation changes, and areas with no changes. The main figure scale shows temperature differences in a vertical cross section, while the inset shows temperatures in the bottom model layer, therefore the scales differ.

We would like to keep the figure as we think it shows several of the important processes related to shrub cover increase. However, we acknowledge that the figure is not of pivotal importance in presenting the main results of the article, we have chosen to move it to the supplementary material, along with some of the corresponding text.

Figure 7. The middle row of figures would be easier to interpret if the scales were the same.

Although we do acknowledge the reviewer's point of interpretability, we find that due to the large differences in values in the two seasons, the figures are more informative and show the results more clearly with colors on different scales. We have strived to keep scales similar for several other figures, but find that an exception is necessary for this particular figure.

Not clear from the figure caption which simulation we are looking at (i.e. Veg0k -

30 RefVeg) Corrected

20

Corrected

Figure 8. On the right plot, label the two seasons as 'warm' and 'cold' or 'RefVegwarm' and 'RefVegcold' instead of the year, as this is how you refer to them elsewhere in the manuscript.

35 manuscript Corrected

The title of the left plot is too long and so is hard to read where it overlaps the left axis labels.

40 Corrected

Figure 11. In the all other figures showing anomaly distributions across the domain, you state that the figure is "only showing significant results at the 95% confidence level" –why do you not do that here?

45 Just forgot, it has been corrected in the revised manuscript.

Figure 12. Inconsistent use of "2 M" and "2m" throughout figures in the manuscript Specify that these are inter-seasonal means *Corrected*

50

Table 2. Responses are presented inconsistently: Why are precipitation, snowfall and low cloud coverage changes shown as percentages whereas the actual values are given for the other variables? Why is the mean value for RefVeg MAM Low cloud coverage 0.31 when the mean values for the warm and cold years are 0.29 and 0.29?

5 (this applies to other values presented in the table) Why are you only averaging over areas with vegetation changes? Cloud cover and precipitation effects might not be limited to the atmosphere directly above the vegetation change for instance."

For precipitation and snowfall we found that accumulated values, and relative changes to these were more informative and gave a better impression than actual values as averaged over the areas, as presented for eh other variables. This is in part because of the temporally and spatially scattered pattern in these values, as opposed to the other variables presented. The low cloud fraction is a diagnostic variable more than a direct physical value, and as such, we found it more informative to also present changes in relative numbers. Although

- 15 these specific variables (precipitation, cloud cover) affect the atmosphere over an area greater than the ones directly covered by shrub cover increase, this is not the case for all variables. To provide informative values of responses to shrub cover increase, and to keep as much consistency as possible, we have chosen to use these specific areas for averaging ad accumulation throughout. However, to show the spatial extent of the atmospheric response to the different variables, maps are presented in the figures as additional information to the
- values in this table. We appreciate this thorough review of the table and constrictive suggestions for improving it. We have made corrections where needed.

**Reference list**

25 "P20L1-12 Duplication of reference: Myers-Smith et al. 2015a appears to be the same paper as Myers-Smith et al. 2015b"

Corrected.

**30 Supplementary material**

"Figure S1. Consistent scales would be better Figure S5. In the text preceding this figure, it is not clear why "(Veg0K-RefVeg)" is included at the end. "

Figures and captions have been revised and corrected where appropriate.

**35**

**Answers for anonymous Referee #2**

Received and published: 7 December 2016

"This paper extends prior research into the potential climatic effects of a hypothetical increase in shrub and tree cover (in this case, shrub and tree height) in a region in northern Fennoscandia. My overall impression is that this is a solid paper that provides some interesting new results on the topic of vegetation feedbacks onto climate in the Arctic region. This work does not represent a giant leap forward, but the study is sound and, in my opinion, the paper is worthy of publication.

45

The paper can be divided into two main parts. The first part focuses on the potential changes in vegetation distribution under a 1K temperature change. The second part focuses on an assessment of the impact of such a shift in vegetation on temperature. From my perspective the first part is fine. One could quibble with aspects of the method

50 and argue whether or not the projected changes in vegetation distribution are completely realistic or not, but I don't feel that the realism is really the point. The main

goal is to generate vegetation distribution changes that are at least quasi-realistic and that can then be applied in the subsequent vegetation change experiment. Perhaps the only recommendation that I would have here is for the authors to be a bit more explicit about this with a statement to the effect that predicting vegetation change based primerily on a climatic equation and into the treated on a first order assessment of

5 primarily on a climatic envelope should just be treated as a first-order assessment of potential vegetation distribution change. "

Firstly, we appreciate the reviewer's positive comments. Regarding the use of the bioclimatic envelopes in this study, the reviewer's assessment is correct in that the aim of the redistribution of vegetation applied here is mainly aimed at providing a pseudo-realistic experiment for investigating the atmospheric feedback mechanisms, rather than providing an accurate and realistic map of present/future vegetation. We agree with the reviewer's suggestion of highlighting this and communicating this aim more explicitly in the manuscript (also see our response to comments from Reviewer #1). In the revised manuscript this side of the study has been more appropriately introduced and the ideas and purpose of the use of

15 the study has been more appropriately introduced and the ideas and purpose of the use bioclimatic envelopes has been better explained.

"For the second part, my main recommendation to the authors is that they work to put this study into better context. Prior studies, cited in the paper have looked at the impacts of shrub and tree area expansion in models and concluded, mainly, that these vegetation changes can lead to warming. So, the authors need to clearly establish what is new from this study. I see two main areas where this is new. The first is that this is being done within a regional climate model, which allows a more detailed assessment of the response. The second new result relates to the variability in the impact across

25 high and low snowfall years and warm and cool summers. The authors should strive to emphasize these points."

We have followed the reviewer's constructive suggestion and reorganized the manuscript to better emphasize these aspects of the study design in the revised manuscript.

30

35

10

With regard to the variability of the impacts across seasons, part of our goal has been to investigate the sensitivity of the atmospheric response to varying conditions represented here by choosing two contrasting years with respect to temperature and snow cover (as further explained in the answer to point 1 from Reviewer #1). In line with this comment, and with similar comments from Reviewer #1, this aspect of the study has been emphasized and presented more clearly in the revised manuscript.

**Minor points:**

40 "1. For the summer feedback, the authors note that the impact of shrubs on summer temperatures is less sensitive to the mean summer temperature (warm or cold summers) than spring temps are to high versus low snow years. That is not surprising. Nonetheless, it would be good to explain why one would think that the summer temperature sensitivity could be related to mean summer temps. "

45

50

This is a good point, and we see that this side of the expected feedbacks has not been properly introduced. As further explained in the revised manuscript, several other similar studies have found the strongest impacts in summer as resulting from increased shrub cover. Furthermore, and as highlighted and cited in the revised introduction, such feedbacks are largely driven by increased surface fluxes of heat and water resulting from shrub expansion, besides albedo changes and the SW/LW fluxes. As such, the mean summer temperature can influence all these variables, and therefore potentially the strength of the feedback as averaged across the season. We hope the reviewer finds this aspect of the study better introduced and discussed in the revised manuscript.

5

10

15

"2. p. 12, line 26. 0.16K versus 0.15K is essentially the same. Shouldn't say that one season has a slightly larger response when they are effectively identical. "

We acknowledge that the sentence is placing too much emphasis on the second decimal number here, and have adjusted this statement in accordance with the reviewer's comment.

"3. Figure 11 and other figures. It would be clearer to be specific that you are talking about warm summer seasons and cool summer seasons. Just writing cold seasons and warm seasons can lead to ambiguity about whether referring to different seasons (spring versus summer, for example)."

This has been amended in the revised manuscript, and figures adjusted where needed for clarity.

20

**Effects of shrub and tree cover increase on the near surface atmosphere in northern Fennoscandia 25**

Johanne H. Rydsaa1, Frode Stordal1, Anders Bryn, 2, Lena M. Tallaksen1

[1] Department of Geosciences, University of Oslo, Oslo, Norway

[2] Natural History Museum, University of Oslo, Oslo, Norway

30

Correspondence to: Johanne H. Rydsaa (j.h.rydsaa@geo.uio.no)

climate change. Extensive evidence has shown that shrub expansion that can lead to positive feedbacks to the regional climate. In this study we evaluate the sensitivity of the near surface atmosphere of the near surface atmosphere to a potential expansion-increase in shrub and tree cover in the northern Fennoscandia region. We have applied the Weather Research and Forecasting model (WRF) with the Noah-UA land surface module in evaluating biophysical effects of increased shrub cover on the near surface atmosphere on a fine resolution (5.4 km x 5.4 km). Using a state of the art atmospheric model, two perturbation Perturbation experiments are performed in which we prescribe a gradual increase of vegetation height in the alpine shrub and tree cover according to empirically established bioclimatic zones within the study region. The first experime

Abstract. Shrub expansionIncreased shrub and tree cover in high latitudes is a widely observed response to

40

35

present day climate, and the second is based on a future 1 K increase in temperature. We focus on the spring and summer atmospheric response. To evaluate the sensitivity of the atmospheric response to inter-annual variations variability in climateic conditions, simulations were conducted for two different contrasting years, one with

**Comment [LMT1]:**

warmer and one with cold-yearer, spring and summer conditions. We have applied the Weather Research and easting model (WRF) with the Noah UA land surface module in evaluating biophysical effects shrub cover on the near surface atmosphere on a fine resolution (5.4 km x 5.4 km). We find that shrub and tree cover increase leads to a general increase in near surface temperatures with the peak-highest influence seen occurring during the snow melting season. It has the largest effect in spring, by advancing the onset of the melting season, and has a more moderate effect on during summer temperatures. We find that the net short wave radiation absorbed by the surfacewarming effect is sensitive to anthe enhancement of shrub and tree heights, which lower-decreases the surface albedo, resulting in -C-taller vegetation having a stronger influence on both spring and summer temperatures. Counteracting effects include increased evapotranspiration increased snow cover and enhanced evapotranspiration which can causinglead to increased cloud cover, and precipitation and snow cover. We find that the strength of the atmospheric feedback effects resulting from increased shrub cover is more sensitive to snow cover variations-, and to a lesser extent to than-summer temperatures.-. Taller vegetation has a stronger influence on both spring and summer temperatures. However, OoOur results show that the positive feedback to high latitudes warming induced by increased shrub and tree cover is a robust feature across inter-annual differences in meteorological conditions, and will likely-play an important role in land-atmosphere feedback processes in land atmosphere feedback processes land atmosphere feedback processes in the future. an you be more specific here? \_\_potentially delete this part}

Keywords. Climate change, Arctic amplification, -Vvegetation perturbations Shrubs, shrub expansion, Arctic
 greening, Fennoscandia, WRF, land-atmosphere feedback

**1 Introduction**

5

10

15

25

30

35

Arctic warming is occurring at about twice the rate as the global mean warming (IPCC, 2013;Pithan and Mauritsen, 2014).5 This is partly owing to partly due toland-atmosphere feedback mechanisms in thein -high latitude ecosystems (Beringer et al., 2001;Chapin et al., 2005;Serreze and Barry, 2011;Pearson et al., 2013). such as - Some of these feedbacks are related to the observed aArctic greening (Myneni et al., 1997;Piao et al., 2011;Snyder, 2013). This Arctic greeningterm refers to the observed increase in high latitude biomass mainly resulting mainly from increased temperatures (Walker et al., 2006;Forbes et al., 2010;Elmendorf et al., 2012). While present across arctic ecosystems, tThe observed increase in biomass is largely related toincludes the extensive expansion increase in shrub and tree cover in areas previously <del>tundra</del> covered by tundra areas (Tape et al., 2006;Sturm et al., 2001;Forbes et al., 2010) and northward migrating tree lines (Soja et al., 2007;Tommervik et al., 2009;Hofgaard et al., 2013;Chapin et al., 2005).

Increased tree and shrub cover alters the biophysical as well as the biochemical properties of the surface, leading toinducing\_This changeschanges in\_land-atmosphere feedbacks\_(e.g. Bonan, 2008). With increasing canopy height and complexity, the overall surface albedo is-decreasesd, and more incoming radiation is absorbed, and the overall surface albedo is decreased. This is known as the albedo effect of increased vegetation. Sturm et al. (2005a) observed the importance impact of shrub cover on wintertime albedo in snow covered regions and its implications for the winter surface energy balance. They concluded that increased shrub cover due to higher temperatures-caused a positive feedback to warming through lowered surface albedo. The absorbed radiation heated the canopy itself and increased the sensible heat flux to the atmosphere. They also found that an increase of shrub canopies protruding the snow covercover also-shaded the snow beneath the canopy from radiation. This further led to - and decreaseddecreased -melt and sublimations<del>, as increased higher shrub and tree cover enhanced the winter snow cover beneath the shrubs and increased the soil temperatures in winter. Other</del>

5 studies have shown that more shrubs act to speed both the onset and advance of the melting season through its effect on surface albedo (McFadden et al., 2001;Sturm et al., 2001a).

Increase Enhanced in leaf area index (LAI) associated with an increase in shrub and tree cover can eause enhanced morelead to higher evapotranspiration (ET). This subsequently leads to more latent heat (LH) being transferred into the atmosphere, and which may also acts to increase air the temperatures (Chapin et al., 2005). The enhanced increase ind LH mayight also lead to enhanced more cloudiness and precipitation (Bonfils et al., 2012;Liess et al., 2011). Increased cloud cover may in turn act to limit the effect of anthe albedo decrease through lowering of the short wave (SW) radiation reaching the surface. In addition, increased shrub and tree eover has been observed to increase the soil temperatures in winter, enhance the winter snow eover and speed the melting season through its effect on surface albedo (MeFadden et al., 2001;Sturm et al., 2001a).

20

25

40

15

10

The height of the shrubs and treesheight is also important forinfluences the strength of the atmospheric responseland-atmosphere feedbacks, and this wasas studied specifically by Bonfils et al. (2012). By modelling an increase in the shrub cover by 20% in areas north of 60°N, they found that a higher increase in the regional temperature increased more for taller shrubs as compared to lower ones. They explained the temperature increase by the additional lowering of albedo and increase in ET-LH corresponding to taller and more complex canopies.

In summer, 4increased shrub cover may also act to shade the soil beneath the shrubs, thereby lowering the temperature of the soil and . Thisthus, acted to decrease summer permafrost thaw as observed by Blok et al. (2010). This effect was also modelled in a study by Lawrence and Swenson (2011) who applied an increase in shrub cover by ~20% in the Arctic region. Their findings suggest, y, however, found that increased temperatures due to albedo decrease more than offset the cooling of the soil by the shading effect, resulting in a net increase in soil temperatures. In both these

30 The studies of In both of the mentionedBonfils et al. (2012) and Lawrence and Swenson (2011) both prescribe modelling studies, a 20% increase in shrub-cover was prescribed, based onby a procedure of expanding existing shrub cover into areas of tundra or bare ground. Based on circumpolar dendroecological data and several future emission scenarios, Pearson et al. (2013) concluded that the warming effects of increased shrub cover found in these two studies by Bonfils et al. (2012) and Lawrence and Swenson (2011)-were realistic, however, ,-but that athe applied shrub expansion of a-20% increase in land cover mayight be substantially underestimated. They predicted by applying various climate thatscenarios, that about half of the regions defined as tundra could be covered by shrubs by 2050, by applying various climate scenarios.

The actual extent of shrub expansion into tundra regions\_<del>, as well asa-nd</del> the predicted increase in shrub <del>density</del> <del>and</del> height in coming decades<del>, are is-highly uncertain, and</del> determined by numerous and complex mechanisms

and environmental forcers. As highlighted by Myers-Smith et al. (2011), climatic forcers (e.g. air temperature, incoming solar radiation, precipitation), and soil properties (e.g. soil moisture, soil temperature and active layer depth), coupled withter biochemical factors such as the availability of soil nutrients and atmospheric CO2 concentrations, all influence the rate of shrub growth. In addition, disturbances, such as fires, heavy snow pack and biotic interactions including herbivory, makes accurate estimates of future shrub distribution challenging (Milbau et al., 2013). Tape et al. (2012) <del>also</del>-highlighteds the importance of soil properties in estimating likely areas of shrub expansion and shrub-climate sensitivity, and argued that. Tiphis factor increases the geographic heterogeneity of shrub expansion. In addition-to these determining mechanisms, increased shrub cover has also been suggested to trigger feedback loops that further induce shrub growth by e.g., shrub-snow interactions (Sturm et al., 2005a;Sturm et al., 2001a;Sturm et al., 2005b). Positive feedbacks include lowering of winter spring albedo which-causinges earlier snowmelt, longer growing seasons and increased soil temperatures, all favorable for growth. Also, thicker wintertime snow packs in shrub areas acts to insulate the ground during winter and increase the soil temperatures (Sturm et al., 2001a).

5

10

- Several of the controlling factors regulating shrub growth and expansion have been further investigated using dynamic vegetation models. Miller and Smith (2012) found-simulated an increase in shrub cover caused by mainly warmer temperatures and longer growing seasons. They explained found that the shrub cover increase was in part enhanced by shrub-atmosphere feedbacks, particularly related to a reduction in the albedo effect of covering tundra with shrubs, related to leading to with an increase in canopiesy heights protruding the snow cover. In agreement with observations, several other modelling studies have also reported found increased biomass production and LAI-related to shrub invasion and replacement of low shrubs by taller shrubs and trees (e.g. Zhang et al., 2013;Miller and Smith, 2012;Wolf et al., 2008).
- Several recent studies have aimed at isolating a few of the dominating environmental drivers of shrub expansion. 25 Myers-Smith et al. (2015a) investigated climate-shrub growth relationships and found that mean summer temperatures and soil moisture content are particularly important forcers. By examining circumpolar dendroecological data from Arctic and alpine sites, they demonstrated that the sensitivity of shrub growth to increased summer temperatures was higher at European than American sites. Furthermore, they found that thea 30 higher sensitivity to climate forcing was higher for taller shrubs at the upper or northern edges of their present domain and at sites with greater higher soil moisture. Also bB ased on dendroecological observations-in northern Seandinavia, Hallinger et al. (2010) concluded that the mean summer temperatures and winter snow cover are the main climatic drivers correlated with shrub growth in sub-alpine areas in northern Scandinavia. Based on tundra vegetation surveys covering 30 years in 158 plant communities spread across 46 high latitude locations, 35 Elmendorf et al. (2012)\_also-demonstrated a biome-wide link between high latitude vegetation increase and local summer warming, based on tundra vegetation surveys covering 30 years in 158 plant communities spread acr 46 high latitude locations.
- 40 associated withto increased shrub cover in tundra areas areis more moderate\_compared for example to an

**Comment [JHR2]:** Dette avsnittet kan muligens kuttes.

expansion of forest ecosystems, -and -as compared to changes resulting from e.g. a comparable expansion of -a comparatively morerather modest effect on the overlying atmosphere is to be expected (Beringer et al., 2005; Chapin et al., 2005; Rydsaa et al., 2015). Still, afore-mentioned observational and modelling studies have shown considerabledemonstrated notable feedbacks to the regional climate. Considerable

5 However, large uncertainties still exist concerning within the estimated of both the extent of shrub and tree advance in relation response to elimatic warmingforcing, and to the corresponding feedback to climate resulting of in response to these ecosystem changes (Myers-Smith et al., 2015a; Pearson et al., 2013).

-In this study we investigate the regional atmospheric response related to biophysical changes resulting 4 10 from enhanced vegetation cover in high latitudes. Our investigations are carried out on a domain covering northern Fennoscandia and north-west Russia. This is a sensitive region for shrub expansion in response to climate forcing-according to (Myers-Smith et al., 2015a). Extensive increase in the shrub covered area, in addition to as well as shifts in the tree lines towards higher latitudes and altitudes, haves been observed in this region over the past decades (Tommervik et al., 2004;Hallinger et al., 2010;Tommervik et al., 2009;Rannow, 15 2013). We The study focus addresses the atmospheric response to on both an expansion of the areal extension increase in the aread covered by of shrubs and low deciduous trees in northern Fennoscandia, area and and the sensitivity effects of to their increased height when it comes to feedback strengthheight of shrubs and trees. For simulations are conducted for a limited region on high temporal and spatial resolution (5.4 km x 5.4 km), using a state of the art regional atmospheric model. This enables us to investigate fine scale features 20 corresponding atmospheric response study domain by applying three types of shrub and tree classes according to their climatic envelope These are derived from empirically determined vegetation climate relationships for the region. The \_ primary research questions motivatingin this study, and the corresponding experimental setup is presented below are:

25

| <del>]</del> | The warme   | r elimate  | is causing | more shru    | bs and  | low  | trees  | to grow  | in the | northern | -Fennoseandia | area.   |
|--------------|-------------|------------|------------|--------------|---------|------|--------|----------|--------|----------|---------------|---------|
| how does     | this feed h | ack to the | atmosphe   | re in the re | gion? N | Aore | specif | fically: |        |          |               |  |

- How will the feedback be influenced by increased shrub and tree cover and heights? Which season will be more affected and experience the strongest feedback?
- 30

35

How sensitive is the feedback to varying climatic conditions, such as snow cover and or temperatures? How sensitive are the atmospheric feedbacks to the amount of shrub and tree increase?

The spring season has been identified as the season with the strongest feedback to temperatures from increa shrub-cover in previous studies -(!!! INVALID CITATION !!! (Bonfils et al., 2012;Lawrence and Swenson, 2011)). This is due to the effects on snowmelt and corresponding surface albedo changes. As the mean summer temperature has been identified as one of the main environmental drivers for future shrub expansion by several studies, a large potential for growth feedbacks lies with the warming response of the atmosphere during this season. For these reasons we have chosen to focus on the atmospheric response during spring and summer seasons.

Formatted: Justified, Line spacing: 1.5 lines, Numbered + Level: 1 + Numbering Style: a, b, c, ... + Start at: 1 + Alignment: Left + Aligned at: 0,63 cm + Indent at: 1,27 cm, Adjust space between Latin and Asian text, Adjust space between Asian text and numbers

| ļ | Formatted: Font: 10 pt, Not Italic |  |
|---|------------------------------------|--|
| ١ | Formatted: Font: 10 pt, Not Italic |  |

DThe details of the methodology, and the experimental design, is presented in section 2.1, the model used in 2.2 and and the development of bioclimatic envelopes for re-distributing shrubs and trees across the study domain in accordance with empirical vegetation climate relationships for the region is are presented in Section 2.3. The results for the atmospheric response for spring and summer is are presented in Section 3 (Results).4, and including

5 the differencesing in response under various climatic conditions and for varying amountsdegree of shrub and tree cover-are presented in Sections 3.2-3.4. Finally, followed by a discussion and conclusions follow in Sections 4 and Section 5.

We focus on the response of the atmosphere during the spring and summer seasons. In addition, the sensitivity to inter-annual variation in mean environmental conditions is investigated. Finally, to explore the potential future atmospheric feedbacks to increased shrub and tree cover on high latitudes, an additional experiment representing a simplified future scenario is conducted. In this experiment, we apply a re-distribution of the shrub and tree cover corresponding to a theoretical 1 K increase in summer temperatures. The resulting feedbacks to the atmosphere are assessed and compared to the effect of vegetation changes corresponding to present day climatic envelopes.

**15 2 Methodology and modelstudy design**

**2.1 MethodologyStudy design**

SModel simulations were conducted on a limited region on with a state-of-the-art high temporal and spatial resolution (5.4 km x 5.4 km), using an atmospheric model. This enableds us to investigate finer scale features of vegetation changes as compared to modelling studies applying coarser grids, and the corresponding finer scale 20 atmospheric responses. To answer research question 1a and investigate the effects of increased shrub and tree cover (referring to both areal expansion and increased height) (Research question a)), we have-conducted six simulations; reference simulations for two seasons (Research questions b) and two climatically contrasting years, (Research question c), and for each year, two separate eorrespondingaecompanying sets of simulations in which the vegetation cover has been was manually altered to represent increased shrub and tree cover (using two 25 different vegetation redistributions) (Research questions d). By comparing the reference and perturbed simulations, we can isolate the effect of shrub and tree cover changes on the overlying atmosphere and evaluate the feedback sensitivity to the degree of shrub and tree increase, (Research question d)assince the simulations are otherwise identical-(i.e. they are driven by the same meteorological forcing and input datasets). Mean summer temperatures were selected as the key environmental criteria used to guide the perturbations applied to the shrub 30 and tree cover.

In order to answer the research question in point 1b, we chose to focus on spring and summer. The spring season has been identified as the season with the strongest feedback to temperatures from increased shrub cover in previous studies due to -surface albedo changes (Bonfils et al., 2012;Lawrence and Swenson, 2011). This is due to the effects on snowmelt and corresponding surface albedo changes. As the mean summer temperature has been identified as one of the main environmental drivers for future shrub expansion by several studies. Furthermore, a large potential for growth feedbacks lies with the warming response of the atmosphere

10

35

during this summereason. For these reasons we have chosen to focus on the atmospheric response during spring and summer seasons.

As the atmospheric response may vary under different climatic conditions (e.g. warm vs. cold, snow rich vs.
 snow poor, present vs. future), and-we chose to run each seasonexperiments for two contrasting years, spanning the -natural variability across a 10-year period with respect to temperature and snow cover in the study region. By averaging the response across two climatically contrasting years, we achieve a robust result representing a wide range in the meteorological variability across this period, without having to simulatinge many years. Secondly, weby investigatinge the contrasting response between the two years, this setup provides us with valuable information of how the contrasting climatic conditions influence the atmospheric feedbacks (Research question c).

The two contrasting-years were selected based on a ten-year (2001-2010) long simulation by Rydsaa et al., (2015), who performedwhich is a dynamical downscaling of ERA Interim, using the Weather Research and
 Forecasting (WRF) model. The reason for using this dataset, instead of a global dataset, was the ability to search through relevant variables to identify suitable years. , and theFurthermore, it provideds consistency in modelling toolmodel setup and boundary conditions with this study. The year of-2003 was chosen as it representeds a low snow cover spring season and a warm summer season in this region in thisthe study region (hereafter referred to as the warm spring and summer season). The year 2008 representeds a snow-rich spring season and a cold summer season in this region (hereafter referred to as the cold spring and summer season).

Two different vegetation redistributions were applied In order to take into account for some of the uncertainties inherit-inherited in the shrubs' response to summer temperatures, one bioelimatic envelope is based on a 10 year average of JJA temperatures, and one based on a 1-K increase in the same JJA temperatures, yielding a newtwo different vegetation redistributions have been introduced. They are based on the concept of bioclimatic zones (ref., -as explained in-Section 2.3), and the two distributions allows: The difference in atmospheric response to these two cases of shrub cover increase provides a measure of the sensitivity of the atmospheric feedback to the variability in shrub cover increase in response to mean JJA temperatures change to be assessed, and is briefly examined. The more drastic vegetation change (i.e. the one based on a 1 K temperature increase) may represent a scenario in which the response of the shrub cover to warmer conditions is faster, or alternatively represent some future distribution of shrubs.

35

30

25

StudyingCombining findings of the atmospheric response in two different vegetation distributions, and contrasting years (- and considering the differing response in the warm versusas welled the cold) years, further some careful assumptions about the potential changes in atmospheric response under varying climatic conditions and vegetation distributions can be made. Based on this, we can make some careful assumptions regarding whatallow 
[revised manuscript text omitted]

**2.2.1 Land cover and re-distribution**

20

25

30

35

or tThe land cover input data is based on, we use the newly available 20 class MODIS 15 sec resolution dataset 2014). In this dataset most of the Arctic and alpine part of our study area is covered by the category of "open shrubland", consisting of low shrubs of less than 0.5 m height. This gory was further used tohas a basis to implement empirically based adjustments aiming to distinguish the atmospheric sensitivity to shrubs and low deciduous trees of various heights. The study domain was divided into ed on mean JJA temperatures and re-distributed shrubs and low trees following the approach of Bakkestuen et al. (2008). The shrub and tree vegetation was re-distributed across the study domain applying bioclimatic envelopes, which were derived from empirically determined vegetation climate relationships for the region. For this purpose we divided our study domain into elimatic vegetation zones based emperatures, and we defined shrub and low tree categories according to their empirically derived getation zones (Bakkestuen et al., 2008). In order to prevent shrubs from being distributed in areas insuitable. despite favorable elimatic conditions, the extent of areaextents with of other vegetation way, the vegetation was adjusted in accordance with the derived In this keeping the heterogeneity in the while vegetation distribution acrosswas kept erved in the original data

The results for the dominant vegetation categories as distributed across the domain for the different simulations, along with mean JJA 2 m temperature limits used to determine the elimatic envelopes, are shown in Fig. 1.

In order to distribute the elimatic vegetation zones geographically across the study domain, we utilized were derived using some general features of vegetation distribution among the alpine zones that have been determined for this area. (The various alpine zones are defined as altitudinal dependent belts of vegetation above the forest Formatted: Highlight

linewhere (Gottfried et al., 2012). Although the altitudinal extent of each alpine zone is determined by the local mean decline in temperature with elevation (i.e. local meantemperature lapse rate2) (Bakkestuen et al., 2008) in addition to various geographical and elimatic features, the relative\_feeded?] altitudinal extent of eachalpine zones remains rather constant throughout across the domain in focus here. This altitudinal extent of each alpine zone to determine temperature based elimatic envelopes for each alpine zone. The altitudinal extent of each alpine zone used in this study is based on Moen et al. (1999), but also confirmed by a new dataset from the region (Bjørklund et al., 2015).

10

15

5

As illustrated in Fig. 2, and fFollowing the vegetation categorization of Moen et al. (1999) and Bakkestuen et al. (2008), we defined tall shrubs and boreal deciduous trees reported to be characterized bywith a height from 2 to 5 m (Aune et al., 2011) as to\_belonging to the sub-alpine zone,. We defined tall shrubs with height from 0.5-2 m as to\_belonging to the low-alpine zone, and low shrubs with height up to 0.5 meters to\_belonging to the mid-alpine zone (Fig. 1). The high alpine zone contains no shrubs and is characterized by barren ground, boulder fields or seattered vegetation (Moen et al., 1999). High mountain tops were regarded as high-alpine (and largely fit in agreement withwith the defined elimatic limits), and vegetation cover in these areas were adjusted accordingly (see e.g. Karlsen et al. (2005)).

The climatic forest line was found and appliedused to separate the boreal forest from the sub-alpine region characterized by scattered mountain birch (Aas and Faarlund, 2000). The last mountain birches stretching towards higher elevations are approximately 2 m tall, and define the so-called boreal tundra or tree line ecotone (Hofgaard, 1997;Bryn et al., 2013;de Wit et al., 2014). This ecotone was determined here to be above the line where the fraction of boreal tree cover exceeds 25% in each grid cell. This line furthermore defines the base line temperature used to derive the alpine vegetation zones at higher elevations. The climatic boreal forest line was found to correspond well with the mean summer 12 °C isotherm (in our domain) (Fig. 1), which is slightly higher than what is found in southern parts of mountainous Scandinavia (Aas and Faarlund, 2000;Bryn, 2008). The sub-alpine zone was then determined based on an average altitudinal extent of 100 m (Aas and Faarlund, 2000), the low-alpine and mid-alpine zones were both estimated to be on average 300 m in altitudinal extenteach, and vegetation cover at higher elevations defined as high alpine zone (Moen et al., 1999). This constitutes the reference vegetation distribution (RefVeg) as illustrated in (Fig. 2).

30 Based on temperatures from previous simulations covering the area by\_(Rydsaa et al., (2015), the mean tropospheric JJA lapse rate for the area was found to be 6.0 K km-1. This value was used together with the average zone-heights to find the summer temperature ranges for each elimatic vegetation zone. The interpolated mean JJA 2 m temperature was then used to distribute each shrub category across the domain in accordance with their climatic envelope. isIn order to simulate a shrub expansion and re distribution of tall vs. low shrubs in accordance with a theoretical future increase in summer temperatures, rRevised climatic zones with the same relative altitudinal extent were calculated based on increased JJA 2 m temperatures, and vegetation categories were re distributed accordingly. is (Fig. 2)For simplicity, the mean lapse rate is assumed to be the same in the new temperatures. The procedure is illustrated in Fig. 3T.

the two\_To represent each alpine vegetation type in the model, we chose suitable vegetation categories (and ) from the ones already defined within the thus model system. The categories based on categories types already present in the domain., with Special emphasis was ongiven to decreasing LAI and canopy height for vegetation applied distributed towards higher altitudes and latitudes, and The choices were based on a recent mapping of vegetation types in the region (Bjørklund et al., categories and their corresponding parameter values is presented in Table S1, 2015) supplementary material. With two exceptions (see supplementary material, Table S1, bold), parameter values keep consistency between and within each vegetation category. The original usage in the given along with the minor alterations. The only changealteration between the reference simulations ("RefVeg") and perturbed simulations ("Veg0K", "Veg1K") is the land cover. Any other differences in other variable values result from the land cover changes, as simulations are otherwise identical with respect to setup and meteorological forcing data. This methodology aims to isolate the signal in the atmospherie response to land cover changes alone. The difference between VegOK and RefVeg shows the effects of an increase in shrub and tree cover as having reached the present day climatic potential (as de-mean rfined here). The difference between Veg1K and RefVeg on the other hand, shows the enhanced effect of a potential vegetation shift corresponding to a 1 K shift in surface temperatures and is presented afterwards.

15

20

25

10

5

**3 Results**

The only change between the reference simulations ("RefVeg") and perturbed simulations ("Veg0K", "Veg1K") is the land cover. Any other differences in other variable values result from the land cover changes, as simulations are otherwise identical with respect to setup and meteorological forcing data. This methodology aims to isolate the signal in the atmospheric response to land cover changes alone. The results are therefore presented largely as anomalies between reference and perturbed simulations when the effects of land cover are of main interest. The difference between Veg0K and RefVeg shows the effects of an increase in shrub and tree cover as having reached the present day elimatic potential (as defined here). The difference between Veg1K and RefVeg on the other hand, shows the enhanced effect of a potential vegetation shift corresponding to a 1 K shift in surface temperatures and is presented afterwards.

30

35

In the following sSections- 3.1 --3.-3 present the the mean-seasonal effects on the overlying atmosphere of increased shrub and tree coverever (Veg0K-RefVeg) for each mean\_season (MAM and JJA) are presented...; specifically. The most moderate vegetation re distribution (Veg0K) is focused.-Results are presented as mean anomalies between the reference and perturbed simulations (Veg0K-RefVeg), as averaged over the warm and the cold year. Special emphasis is on Special emphasis is on how the increased shrub and tree cover alters the feedback to atmospheric near surface temperatures. Changes in other variables are presented largely to explain the variations in temperature. We start presenting the results In Section 3.1 Results are first given as averages over the two spring seasons and two summer seasons. This gives an mean-estimate of the mean response of the atmosphere across a wide range in meteorological conditions and thus represents a robust estimate of shrub induced effects across inter-annual variations. To show the sensitivity in the atmospheric response to differing meteorological conditions, rResults Next we comparingson of eing the anomalies response between in the warm

and versus the cold spring and summer seasons are presented next in Sections 3.2 and 3.3, as it will indicate the sensitivity of the atmospheric response to specific meteorological variations. Section 3.2 focuses on the effect of variation We focus in Section 3.2primarily on Here these effects of are limited to variations inin spring snow cover between the two years, primarily (in Section 3.1.1) and Section 3.3 on the effect of variation in summer near surface temperatures (in Section 3.21.2). Finally, in order to account for the sensitivity of the shrub and tree cover to JJA temperatures, the atmospheric results response to-of the morest extensive vegetation re-distribution (Veg1K-RefVeg) is are presented in Section 3.43. in Section 3.2 a brief analysis of the future scenario with distributed according to a 1 K increase in temperature is presented. Special emphasis is on how the shrub and tree cover alters the feedback to atmospheric temperature. response as averaged over all areas with vegetation changes for variables, are presented in Table 2. <del>ce atmo</del>

10

15

5

**3.1 Atmospheric effects of shrub and tree cover increase**

Mean values for the reference simulations (mean of RefVeg\_old and RefVeg\_warm), along with mean rResponses in surface fluxes and near surface atmospheric variables (as averaged over all areas with vegetation changes and across the warm and cold years (Veg0K-RefVeg), and acrossfor bothcach years (-in parentheses-)years, are presented in Table 2. - for surface fluxes and near surface atmospheric variables, are presented in Table 2. the values for the spring seasons are given in. Effects of shrub and tree cover increase as averaged over the two spring seasons (Veg0K-RefVeg)by both expansion into areas with lower vegetation, and by increased height <del>of existing shrubs (Veg0K RefVeg) as averaged over the two spring seasons\_, a</del>re presented in Fig. 4.<del>.</del>. Top panels Fig. 4a shows the spatial distribution change in 2 m temperature anomalies averaged over the spring <del>seasons (</del>left) and mean values for each <del>separate bioclimatic zone area with vegetation changes for each</del> vegetation type(corresponding areas are indicated in Fig.14, botttom panel) in the bar plot (right)(right).

25

30

20

In spring, an overall increase in near surface temperatures is seen for all areas whereith shrub and tree cover increases (Fig. 4a). The higher anomaly values are seen in areas with increase in taller vegetation shrubs and trees (as also indicated in the bar plots). Average increase in 2 m temperature over the spring season is 0.1 K (Table 2); however, there are large spatial differences appear (Fig. 4a, bar plot). Values close to The mean response reaches up to 0.6 K are in seen 
[revised manuscript text omitted]
 eovered areas of the total study simulation domain). There are small differences in Total precipitation is similar for among \_ the two years, although the rain-to-snow ratio is larger in the warm season spring due to higher temperatures. The difference in mean spring snow cover between the warm and cold spring seasons is presented in Fig. 78.

**10**

15

5

It is clear from Fig. 7 that t-The snowmelt lso starts earlier in the warm spring season (RefVegwarm) (more than two weeks) and a fasteronset and speed of the melting rate of snowmelt is seen as -seen for the warm season differ between the two as compared to the cold spring season (RefVegcold)years, with and the main the largest difference in snow cover is in May (Fig. 7). The melting starts more than two weeks earlier in the warm spring season. It is worth noting that the The dominatingmost pronounced effects of increased shrub cover differ between snow covered and snow free conditions and the differences in the shrubs' influence-on the atmosphere are largest-during the melting season, i.e. -in-May-June.

20

The difference in 2 m temperature response to increased shrub cover(AT2m =Veg0K RefVeg) between the warm and cold spring seasons (AT2m warm AT2m cold), is shown in Fig. 80. The warm spring season experiences up to 0.38 K higher increases in 2 m temperature in response to shrub and tree cover increase as compared to the cold one (Fig. 8) spring season. As seen in the right panel of Fig. 89, the anomaly distribution is shifted towards overall higher values in general in the warm season. The shrubs act to enhance warming more in the warm spring season more than in the cold spring season. This represents a positive feedback to warm conditions and early snowmelt.

25

30

35

The increased shrub and tree cover decreases-leads to a reduction in the snow depth in the spring season-as averaged over all areas with vegetation changes, as seen in Fig.  $940_{25}$  top panel (the spatial pattern distribution of snow cover is shown in Fig.  $S3_{35}$  in the -supplementary material). An exception is seen in the late spring (and early cold season-summer, not shown). This is owing-related to the above mentioned-late spring and early summer increase in snow cover found in areas with low-alpine shrub increase. These areas experience. In these areas the mean increase is due tod an increase ind snow fall in the cold summer season and subsequently, and possibly the increase in late spring/early summer snow cover leads to a a shortening of the snow free season in these areas (a grid cell is considered snow freehere defined as if the fraction of ground covered by snow is less than 0.1) (Fig. 9b). In the cold season the shortening is only about half a day averaged over the areas with vegetation changes. The warming effect of shrub cover in the warm season on the other hand, acts to prolong the snow free season by just over one day, but-however, it speeds the onset of melting by several days.

**Formatted: Subscript Formatted: Subscript**

Also, increased shrub and tree cover acts to enhance the soil temperatures (Fig. 9c-), with maximum impact in the upper layers of the soil (not shown). The increased precipitation throughoutduring both spring and summer seasons also influences the soil moisture. SThe soil moisture (Fig. 9d) is increased in areas with increased shrub and tree cover throughout the warm spring. and dDue to the shrubs' ability to decrease surface runoff during the

5 main snowmelt. A notable increase in the mid May-soil moisture, and corresponding decrease in surface runoff, is particularly increased seen in mid-May at the time of maximum snow melt. Increased shrub and tree cover also influences the soil moisture and temperature, as shown in Fig. 10. The shrubs act to decrease surface runoff (Fig 9e), during the peak of for both the cold and warm melting seasons. However, during the warm spring season before the main snowmelt starts, runoff is slightly increased higher due to shrubs-during the warm spring season, because of their effect on the increased snow melt earlier in spring for areas with increased shrub and tree cover. The soil moisture is increased in areas with increased shrub cover throughout the warm spring, and due to the shrubs' ability to decrease surface runoff during the main snowmelt, the mid May soil moisture is particularly increased. Also, increased shrub and tree cover acts to enhance the soil temperatures (Fig. 10, third panel), with maximum impact in the upper layers of the soil (not shown). The increased precipitation throughout both spring and summer seasons also influences the soil moisture.

**3.1.23.3 Sensitivity to summer temperatures**

20

25

The two-warm and cold\_summer seasons represent-encompass\_a large range in inter-annual temperature variability. For the With-reference vegetation (RefVeg), Tthe mean JJA 2 m temperature-temperature (averaged over land areas in the domain) for the warm summer season (RefVegwarm) was 11.7 °C, while the cold summer (RefVegcold) represents a lower than usual mean temperature of 9.7 °C, for the warm and the cold year in the two cases. Comparing the two summers, tIn he temperature difference over land is on average 2 °C, and in some places areas the average-difference reacheds 3.3 °C. The corresponding increase in atmospheric absolute humidity at 2 m is 6.9%. The warm summer also represents drier conditions with lower less precipitation (Table 2).

The difference in atmospheric temperature response to increased shrub and tree cover between the two summers  $(\Delta T2m_{warm} \Delta T2m_{eold})$ , is shown in Fig. 104.

The response of the atmosphere to increased shrub cover (Veg0K-RefVeg) shows more similarity across the two warm and cold summer seasons as compared to the two-warm and cold spring seasons. For the summer seasons, the mean difference in near surface2 m temperature response between the warm and cold season-is smaller and more-rather evenly distributed around zero (Fig. 104, right panel). PThe-positive values over areas with low\_ alpine shrub expansion largely indicate less cooling in the warm summer season as compared with the cold summer\_season, during which these areas were partially covered by snow. The tall vegetation changes contributes to similar warming in the summer seasons. The temperature response in the warm season is slightly shifted towards warmer anomalies (Fig 104, right panel), indicating a slightly positive-larger vegetation feedback to warmer summer temperatures in the warm summer season when compared with the cold.

TAs the difference in atmospheric temperature response is larger between the warm and cold spring season when compared with than between the warm and cold summer season.37 Thus, it seems that the strength of the temperature feedback to increased shrub cover feedback is more sensitive to <del>snow meteorological</del> conditions in spring than summer temperatures. This is likely due to the fact that to the temperature feedback isbeing closely linked to albedo changes, which <del>again</del> are heavily dependent on snow cover. Therefore, the strength of

the feedback is more sensitive to temperature in the melting season.

3.23.4 Sensitivity to a 1 Kthe degree of shift in vegetation vegetation changes cover: "future scenario"

The shift in shrub and tree distribution according to the theoretical 1 K increase in summer temperature (Veg1K vegetation distribution) results largely in a northward shift in -migration of distribution of the boreal tree line ecotone, replacing low\_-alpine shrubs with small trees across most of the shrub covered areas, as compared to the Veg0K distribution. It also acts to increase the low\_-alpine shrub cover in higher latitudes and altitudes (Fig. 32). The increased cover of trees rather thanat the expense of shrubs, with corresponding strong decrease in albedo and increase in LAI, enhances the net SW absorbed by the surface. This is balanced by strong increases in SH and LH (Table 2, and Fig. S5 in the -supplementary material). In addition, the vegetation changes result in increasing precipitation and cloud cover (Table 2).

The mean seasonal response in 2 m temperature caused by this vegetation shiftto the increase in shrub and tree cover corresponding to the\_1 K increasein the most substantial vegetation change case (Veg1K-RefVeg) is shown in Figure 121.

[revised manuscript text omitted]

**6 Tables and figures**

**Table 1: Key parameterizations used in the model setup.**

| Parameterization scheme                     | Reference               |
|---------------------------------------------|-------------------------|
| Mellor-Yamada-Janjić planetary boundary     | (Janjic, 1994)          |
| Morrison two moment microphysics            | (Morrison et al., 2009) |
| RRTMG short- and longwave radiation options | (Iacono et al., 2008)   |
| Noah-UA land surface model                  | (Wang et al., 2010)     |

|                                            | RefVeg m        | ean value      | Δ Veg0            | K-RefVeg        | Δ Veg1         | K-RefVeg       |
|--------------------------------------------|-----------------|----------------|-------------------|-----------------|----------------|----------------|
|                                            | MAM             | JJA            | MAM               | JJA             | MAM            | JJA            |
|                                            | (Warm,Cold)     | (Warm,Cold)    | (Warm,Cold)       | (Warm,Cold)     | (Warm,Cold)    | (Warm Cold)    |
| Near surface
temperature [K]            | -5.77           | 10.02          | 0.10              | 0.05            | 0.23           | 0.16           |
| ····· F · · · · · · · · · · · · · · · ·    | (-4.28 , -7.25) | (11.0, 9.06)   | (0.13,0.07)       | (0.06, 0.03)    | (0.28, 0.18)   | (0.16, 0.15)   |
| Upward sensible                            | 0.3             | 52.3           | 0.8               | 1.8             | 1.9            | 4.2            |
| heat flux
[W m -2 ]          | (0.1, 0.5)      | (59.2, 45.5)   | (1.1, 0.6)        | (2.2, 1.5)      | (2.4, 1.3)     | (4.5, 3.8)     |
| Upward latent                              | 6.1             | 33.7           | 2.3               | 2.5             | 3.7            | 3.8            |
| [W m -2 ]                       | (7.7, 4.5`)     | (34.7, 32.7)   | (2.3, 2,3)        | (2.8, 2.2)      | (3.7, 3.7)     | (4.2, 3.5)     |
| Net short wave                             | 54.2            | 153.2          | 2.45              | 3.6             | 4.93           | 7.22           |
| down [W m 2 ]                   | (60.2, 48.3)    | (165.4, 141.0) | (3.18, 1.73)      | (4.26, 2.99)    | (5.98, 3.88)   | (7.86, 6.58)   |
| Net Long wave                              | -38.0           | -55.45         | 0.35              | 0.64            | 0.60           | 0.47           |
| down [W m 2 ]                   | (-40.3, -35.7)  | (-60.8, -50.1) | (0.09, 0.60)      | (0.59, 0.69)    | (0.16, 1.04)   | (0.53, 0.42)   |
| Precipitation*                             | 5865            | 8446           | 1.07%             | 2.2%            | 2.5%           | 4.3%           |
| [mm day * ]                     | (6496, 5234)    | (8090, 8801)   | (1.1%,1.01%)      | (2.4%, 2.06%)   | (2.7%, 1.6%)   | (5.0, 3.7)%    |
| Snowfall*                                  | 4477            | 274            | 1. 4 3%    | 2.3 %**         | 2.8%           | 3.0%**         |
| [mm day * ]                     | (4289, 4666)    | (328, 220)     | (1.5%, 1.3%)      | (3.04%, 1.4%)** | (3.0%, 2.4%)   | (3.5%, 1.2%)** |
| Lowcloud                                   | 0.31            | 0.16           | 1.92%             | 0.81%           | 3.2%           | 0.71%          |
| coverage (<3km)
[fraction] † | (0.29,0.29)     | (0.14, 0.19)   | (2.06%,
1.85%) | (1.0%, 0.7%)    | (3.3%, 3.4%)   | (1.0%, 0.5%)   |
| Vegetation                                 | 0.87            | 0.01           | -0.42             | -               | -0.52          | -              |
| buried by snow
[fraction]               | (0.78, 0.95)    | (0.00, 0.02)   | (-0.43, -0.42)    |                 | (-0.49, -0.55) |                |

Table 2. Mean response in surface fluxes and near surface atmospheric variables as averaged over all areas with vegetation changes.

\*accumulated values over areas with vegetation changes, \*\*not statistically significant,  $^{\dagger}$  average fraction over model layers below 3km

---

## Author Response (AR2)

**Point-by-point reply, Editor Review**

We sincerely thank the editor for taking the time to review our manuscript and provide such constructive comments and suggestions for improvements. Here is a point-by-point list with the editors comments, our replies (in italics), and corresponding changes made to the manuscript. We would also like to thank the anonymous reviewer #2 for reviewing our manuscript again.

**Major comments:**
1. As a non-expert, I find it non-trivial that increases vegetation height (rather than associated changes in stem area index, leaf area index, snow masking or whatever) reduces albedo. Please provide a brief explanation as to why this is in both abstract and introduction.

*This is a good point, and we agree that it should be addressed more precisely in the manuscript. We acknowledge that in the beginning of the manuscript we have not been clear about the cause of the albedo decrease, linking it to much to the height alone, rather than stressing that the increased height is only the most prominent one of several changes to the vegetation, such as LAI, snow masking etc. As the editor points out, it is the taller, more complex canopies that reflect less sunlight back up that causes the decrease in surface albedo. As the height of vegetation increases, most of the other properties will in change as well, and this is the case also in our simulations, as more thoroughly explained in the methods and results sections (specific changes to all parameter values are given in Table S1 in the supplementary material). This is clarified in the revised manuscript in both abstract and introduction, as the editor requested.*

*Abstract: We find that the warming effect is stronger in taller vegetation types, with more complex canopies leading to decreases in the surface albedo.*

*P2, lines 5-8: Increased tree and shrub cover alters the biophysical properties of the surface, inducing land-atmosphere feedbacks (e.g. Bonan, 2008). With increasing canopy height and complexity (including associated variables as leaf area, shade area etc.), the overall surface albedo decreases as more of the incoming radiation is absorbed.*

2. Page 2 15ff to then end of this page, possibly beyond. I think this introduction is too long here, providing too much detail on the specifics of particular studies. Please shorten here to the essential ideas and findings and mention these specifics where appropriate in the discussion section.

*As the editor suggests, the introduction is shortened and some specifics removed. One entire paragraph is moved to the discussion (P.14). We would however, like to keep the remaining text, although we acknowledge it being somewhat long, both for the purpose of properly placing our work in relation to previous research on this topic, and in order to meet the requirements made by the reviewers in the previous round of revision.*

3. Paragraph page 5 L20 could be moved to Section 2.2, but should at least point to Section 2.2, not 2.3? The sentence starting in L23 would be easier to follow in Section 2.2.

*As the editor suggests, the paragraph is moved to section 2.2, and rewritten for clarity.*

*P. 5, lines 33-P6.line 3: moved ad rewritten for clarity.*

4. Figure 2 does not help to understand the paragraph in page 6 L33ff. I also find this text hard to follow. Please try to clarify, and potentially revise/remove Figure 2.

*As the editor points out, the reference to Figure 2 was not accurate and has been removed in the revised manuscript. The surrounding text has been rewritten slightly to try and clarify our methods. In this respect, we do feel that Figure 2 can be very useful, although a proper introduction to that figure has been added to the text.*

*P7, lines 8-20: rewritten for clarity.*

5. P8 L18ff. In my view, this paragraph is not necessary. If you feel the need to give an overview on the results obtained, writing an introductory paragraph to the discussion may be more helpful. I leave this to you to decide.

*As several comments in the previous round of review were related to confusion around the setup of experiments and the presentation of the results, we find that a thorough explanation of how the results are presented and in which order, can be of great help to the readers. We also find that it may help present the results in the right context. As the experiment setup and corresponding analysis are quite complex, we would like to keep this paragraph although it may not be strictly necessary for all readers.*

**Minor comments:**

1. Page 5 L 11. I think the sentence starting with "The two years..." would be best placed after "in the study region." in L7. This may require some additional editing of the rest of the paragraph.
*As the editor suggests, the two sentences are switched and the paragraph has been adjusted.*

*P.5, lines 10-21: Rewritten for clarity.*

2. Integrate the content of Table 1 into the text, this table is not necessary.

*The table is removed as suggested, and the information is incorporated in the text*

*P8, lines 10-12: Key physical schemes applied include the Mellor–Yamada–Janjić planetary boundary scheme (Janjic, 1994), the Morrison two moment microphysics scheme (Morrison et al., 2009), and RRTMG short- and longwave radiation (Iacono et al., 2008).*

3. Figure 1 can be SI material.

*As the setup of the simulations and the derivation of the bioclimatic zones applied here have been subject to confusion in the previous round of review, we are somewhat reluctant to remove this figure from the main article, as we feel it is essential in illustrating very central aspects of our methodology and the link between our simulations and local bioclimatic features. We have edited the text somewhat to better include the figure, and sincerely hope the editor can agree that this figure illustrates central features important in the methodology of the paper and allows for us to keep the figure in the paper.*

4. P6 L 22: Is it "scattered mountain birch" or "last mountain birch"?

*In the sub-alpine region a typical feature is scattered mountain birch. The upper ones of these scattered birches, which grow at the upper limit of this zone, are about 2 meters tall. As the editor points out, the text was somewhat confusing, and has been adjusted to clarify.*

5. P7 L4 "confused by each alpine shrub type." I understand that you've used an alpine vegetation characterization, but you are still trying to simulate boreal vegetation types? Maybe it's helpful to list "each" type here.

*The paragraph has been altered somewhat to try and add clarity to the concept of deriving each alpine vegetation zone. Also, a reference to figure 1 has been added, as it illustrates each alpine vegetation zone and the relation to the mean JJA temperatures. We hope the editor agrees that the paragraph is improved and the method more clearly presented in the revised version.*

6. P7 L 32: How large is the small domain?

*The smaller domain is 330 x 130 grid cells of size 5.4 km x 5.4 km. This has been added to the text.*

7. P12 L 2: Is this so, or does it only seem to be so? Please use precise language.

*As the editor suggests, the language is changed to better reflect the results.*

8. P 16 L 19: are after exception

*Corrected.*

9. Please carefully check punctuation, and importantly the use of commata.

*The manuscript has been thoroughly revised and adjustments have been made according to the editor's suggestion.*

**Effects of shrub and tree cover increase on the near surface atmosphere in northern Fennoscandia**

Johanne H. Rydsaa[1], Frode Stordal[1], Anders Bryn, [2], Lena M. Tallaksen[1]

5    [1] Department of Geosciences, University of Oslo, Oslo, Norway
[2] Natural History Museum, University of Oslo, Oslo, Norway

*Correspondence to*: Johanne H. Rydsaa (j.h.rydsaa@geo.uio.no)

**Abstract.** Increased shrub and tree cover in high latitudes is a widely observed response to climate change that
10    can lead to positive feedbacks to the regional climate. In this study we evaluate the sensitivity of the near surface atmosphere to a potential increase in shrub and tree cover in the northern Fennoscandia region. We have applied the Weather Research and Forecasting model (WRF) with the Noah-UA land surface module in evaluating biophysical effects of increased shrub cover on the near surface atmosphere on a fine resolution (5.4 km x 5.4 km). Perturbation experiments are performed in which we prescribe a gradual increase of taller vegetation
15    in the alpine shrub and tree cover according to empirically established bioclimatic zones within the study region. We focus on the spring and summer atmospheric response. To evaluate the sensitivity of the atmospheric response to inter-annual variability in climate, simulations were conducted for two contrasting years, one warm and one cold. We find that shrub and tree cover increase leads to a general increase in near surface temperatures with the highest influence seen during the snow melting season, and a more moderate effect during summer. We
20    find that the warming effect is stronger in taller vegetation types, with more complex canopies leading to decreases in the surface albedo. Counteracting effects include increased evapotranspiration which can lead to increased cloud cover, precipitation and snow cover. We find that the strength of the atmospheric feedback is sensitive to snow cover variations, and to a lesser extent to summer
25    temperatures. Our results show that the positive feedback to high latitudes warming induced by increased shrub and tree cover is a robust feature across inter-annual differences in meteorological conditions, and will likely play an important role in land-atmosphere feedback processes in the future.

**Keywords.** Climate change, Arctic amplification, vegetation perturbations, Arctic greening, Fennoscandia,
30    WRF, land-atmosphere feedback

**1    Introduction**

Arctic warming is occurring at about twice the rate as the global mean warming (IPCC, 2013;Pithan and Mauritsen, 2014). This is partly owing to land-atmosphere feedback mechanisms in high latitude ecosystems (Beringer et al., 2001;Chapin et al., 2005;Serreze and Barry, 2011;Pearson et al., 2013), such as Arctic greening
35    (Myneni et al., 1997;Piao et al., 2011;Snyder, 2013). Arctic greening refers to the observed increase in high latitude biomass resulting mainly from increased temperature (Walker et al., 2006;Forbes et al., 2010;Elmendorf

et al., 2012). The observed increase in biomass includes extensive increase in shrub and tree cover in areas previously covered by tundra (Tape et al., 2006;Sturm et al., 2001b;Forbes et al., 2010) and northward migrating tree lines (Soja et al., 2007;Tommervik et al., 2009;Hofgaard et al., 2013;Chapin et al., 2005).

5    Increased tree and shrub cover alters the biophysical properties of the surface, inducing land-atmosphere feedbacks (e.g. Bonan, 2008). With increasing canopy height and complexity (including associated variables as leaf and steam area, shade area etc.), the overall surface albedo decreases as more of the 
[revised manuscript text omitted]
 different seasons (Research questions b)  in two climatically contrasting years (Research question c)For each year, two additional simulations  with manually perturbed vegetation cover  representing a gradual increase in shrub and tree cover (using two different vegetation redistributions) were conducted (Research questions d). By comparing the reference and perturbed simulations, we can isolate the effect of shrub and tree cover changes on the overlying atmosphere and evaluate the feedback sensitivity to the degree of shrub and tree increase, since the simulations are otherwise identical.

The spring season has been identified as the season with the strongest feedback to temperatures from increased shrub cover in previous studies due to surface albedo changes (Bonfils et al., 2012;Lawrence and Swenson, 2011). Furthermore, a large potential for growth feedbacks lies with the warming response of the atmosphere during summer. For these reasons we have chosen to focus on the atmospheric response during spring and

5    summer seasons.

As the atmospheric response may vary under different climatic conditions (e.g. warm vs. cold, snow rich vs. snow poor, present vs. future), we chose to run experiments for two contrasting years The two years span the natural variability across a 10-year period with respect to temperature and snow cover in the study region.

10    The two years were selected based on a ten-year (2001-2010) long simulation by Rydsaa et al., (2015), who performed a dynamical downscaling of ERA Interim using the Weather Research and Forecasting (WRF) model. This dataset provides the ability to search through relevant variables to identify suitable years   keeping consistency in model setup and boundary conditions with this study. By averaging the response across two climatically contrasting years, we achieve a robust result representing the

15    meteorological variability across this period, without simulating many years. Secondly, by investigating the contrasting response between the two years, this setup provides us with valuable information of how the contrasting climatic conditions influence the atmospheric feedbacks (Research question c).

20     The year 2003 was chosen as it represented a low snow cover spring season and a warm summer season in this region (hereafter referred to as the warm spring and summer season). The year 2008 represented a snow-rich spring season and a cold summer season in this region (hereafter referred to as the cold spring and summer season).

    ~~Two different vegetation redistributions were applied to account for some of the uncertainties inherited in the shrubs' response to summer temperatures. They are based on the concept of bioclimatic zones (ref. Section 2.3), and the two distributions allows the sensitivity of the atmospheric feedback to the variability in shrub cover change to be assessed. The more drastic vegetation change (i.e. the one based on a 1 K temperature increase)~~

30

**2.2   Land cover and re-distribution**

Two different vegetation redistributions were applied to account for some of the uncertainties inherited in the shrubs' response to summer temperatures. They are based on the concept of bioclimatic zonesBy applying two different distributions, one more moderate and one more drastic,  we account for

some of the  uncertainties related to  the atmosphere's influence on the shrub cover  growth. The more drastic vegetation change  may represent a scenario in which the response of the shrub cover to warmer conditions is faster, or alternatively represent some future distribution of shrubs. Furthermore, by

combining findings of the atmospheric response in two different vegetation distributions,  and the response in the two contrasting years (warm and cold),  allow us to identify potential responses  in  future climate conditions.

[revised manuscript text omitted]

Revised bioclimatic zones with a 1 K increase in JJA 2 m temperatures andwith the same zone-heightsrelative altitudinal extent, but with a 1 K increase in JJA 2 m temperatures, were calculated derived in the same way and vegetation categories re-distributed, resulting in a upward and northward shift in the distribution of shrub

20    categories across the domain. This distribution is referred to as Veg1K, and represents a more drastic change in shrub distribution compared to the reference simulation (Fig. 2). A schematic overview of the simulations and how they were derived from existing datasets is shown in Figure 2.

The reference vegetation distribution (RefVeg) and the two perturbed distributions (Veg0K and Veg1K) are

25    shown in Fig. 3.

To represent each alpine shrub type in the model, we chose suitable vegetation categories (and corresponding parameter values) from the ones already defined within the satellite dataset provided and thus tested within the framework of the model system. The categories were chosen based on vegetation types already present in the

30    domain. Special emphasis was given to decreasing LAI and canopy height for vegetation distributed towards higher altitudes and latitudes, and further based on a recent mapping of vegetation types in the region (Bjørklund et al., 2015). A list of the shrub categories and their corresponding parameter values is presented in Table S1, supplementary material. With two exceptions (see supplementary material, Table S1, bold), parameter values were left unaltered to keep consistency between and within each vegetation category.

The only alteration between the reference simulations (RefVeg) and perturbed simulations (Veg0K, Veg1K) is the land cover. Any differences in atmospheric and soil variable values result from the land cover changes, as simulations are otherwise identical with respect to setup and meteorological forcing. The difference between Veg0K and RefVeg shows the effects of an increase in shrub and tree cover where shrub heights are in

40    equilibrium with the climatic potential (as defined by the bioclimatic zones and 10-year mean JJA temperatures).

The difference between Veg1K and RefVeg, in comparison, shows the sensitivity to a potential vegetation shift derived from a 1 K increase in mean JJA temperatures.

**2.3  Model**

WRF V3.7.1 (Skamarock et al., 2008) is a non-hydrostatic weather prediction system with a wide variety of applications ranging from local scale domains of a few hundred meters in resolution to global simulations. With a range of physical parameterization schemes, the setup may be adjusted to simulate case-specific short-term weather events, or decadal long climate simulations. The current setup is based on available literature (ref. the NCAR choices for physical parametrizations for high latitude domains), and a consideration of the polar WRF setup and validation studies (Hines and Bromwich, 2008;Hines et al., 2011). A summary of kKey physical schemes applied is presented in Table 1.include the Mellor–Yamada–Janjić planetary boundary scheme (Janjic, 1994), the Morrison two moment microphysics scheme (Morrison et al., 2009), and RRTMG short- and longwave radiation (Iacono et al., 2008).

[revised manuscript text omitted]